# KERNEL NEURAL OPTIMAL TRANSPORT

**Alexander Korotin**
Skolkovo Institute of Science and Technology
Artificial Intelligence Research Institute
Moscow, Russia
`a.korotin@skoltech.ru`

**Daniil Selikhanovych**
Skolkovo Institute of Science and Technology
Moscow, Russia
`selikhanovychdaniil@gmail.com`

**Evgeny Burnaev**
Skolkovo Institute of Science and Technology
Artificial Intelligence Research Institute
Moscow, Russia
`e.burnaev@skoltech.ru`

## ABSTRACT

We study the Neural Optimal Transport (NOT) algorithm which uses the general optimal transport formulation and learns stochastic transport plans. We show that NOT with the weak quadratic cost may learn fake plans which are not optimal. To resolve this issue, we introduce kernel weak quadratic costs. We show that they provide improved theoretical guarantees and practical performance. We test NOT with kernel costs on the unpaired image-to-image translation task.

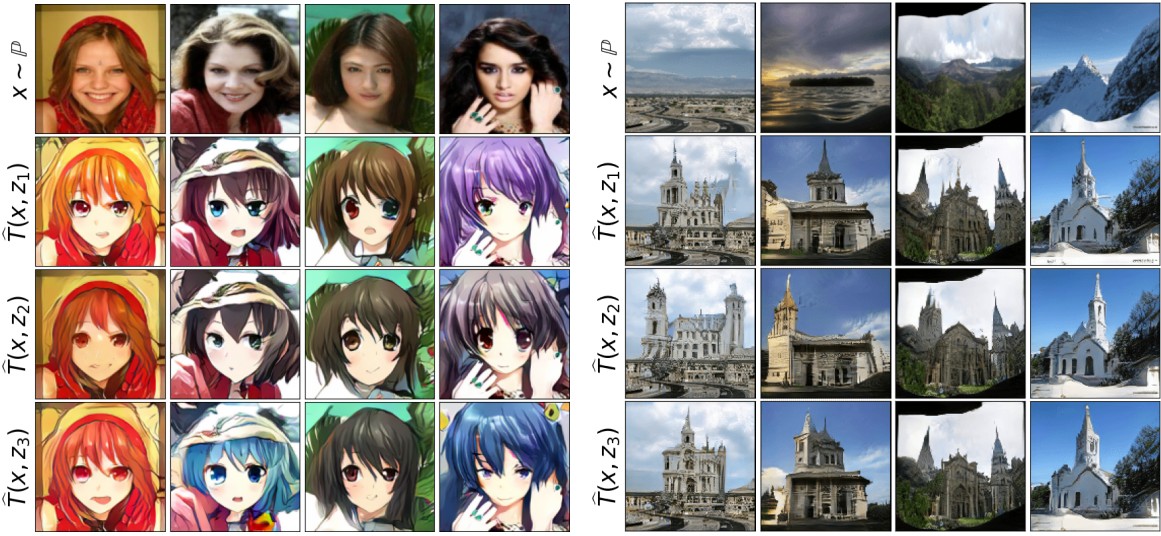

(a) Celeba (female) $\rightarrow$ anime, $128 \times 128$.    (b) Outdoor $\rightarrow$ church, $128 \times 128$.

Figure 1: Unpaired image-to-image translation (one-to-many) by Kernel Neural Optimal Transport.

## 1 INTRODUCTION

Neural methods have become widespread in Optimal Transport (OT) starting from the introduction of the large-scale OT (Genevay et al., 2016; Seguy et al., 2018) and the Wasserstein Generative Adversarial Networks (Arjovsky et al., 2017) (WGANs). Most existing methods employ the **OT cost** as the loss function to update the generator in GANs (Gulrajani et al., 2017; Sanjabi et al., 2018; Petzka et al., 2018). In contrast to these approaches, (Korotin et al., 2023; Rout et al., 2022; Daniels et al., 2021; Fan et al., 2022a; Korotin et al., 2023) have recently proposed scalable neural methods to compute the **OT plan** (or map) and use it directly as the generative model.

In this paper, we focus on the Neural Optimal Transport (NOT) algorithm (Korotin et al., 2023). It is capable of learning optimal deterministic (one-to-one) and stochastic (one-to-many) maps and plans for quite general **strong** and **weak** (Gozlan et al., 2017; Gozlan & Juillet, 2020; Backhoff-Veraguas et al., 2019) transport costs. In practice, the authors of NOT test it on the unpaired image-to-image translation task (Korotin et al., 2023, §5) with the *weak quadratic cost* (Alibert et al., 2019, §5.2).

**Contributions.** We conduct the theoretical and empirical analysis of the saddle point optimization problem of NOT algorithm for the weak quadratic cost. We show that it may have a lot of *fake solutions* which do not provide an OT plan. We show that NOT indeed might recover them (§3.1). We propose *weak kernel quadratic costs* and prove that they solve this issue (§3.2). Practically, we show how NOT with kernel costs performs on the *unpaired image-to-image translation* task (§5).

**Notations.** We use $\mathcal{X}, \mathcal{Y}, \mathcal{Z}$ to denote Polish spaces and $\mathcal{P}(\mathcal{X}), \mathcal{P}(\mathcal{Y}), \mathcal{P}(\mathcal{Z})$ to denote the respective sets of probability distributions on them. For a distribution $\mathbb{P}$, we denote its mean and covariance matrix by $m_{\mathbb{P}}$ and $\Sigma_{\mathbb{P}}$, respectively. We denote the set of probability distributions on $\mathcal{X} \times \mathcal{Y}$ with marginals $\mathbb{P}$ and $\mathbb{Q}$ by $\Pi(\mathbb{P}, \mathbb{Q})$. For a measurable map $T : \mathcal{X} \times \mathcal{Z} \to \mathcal{Y}$ (or $T_x : \mathcal{Z} \to \mathcal{Y}$), we denote the associated push-forward operator by $T\sharp$ (or $T_x\sharp$). We use $\mathcal{H}$ to denote a Hilbert space (feature space). Its inner product is $\langle \cdot, \cdot \rangle_{\mathcal{H}}$, and $\| \cdot \|_{\mathcal{H}}$ is the corresponding norm. For a function $u : \mathcal{Y} \to \mathcal{H}$ (feature map), we denote the respective positive definite symmetric (PDS) kernel by $k(y, y') \stackrel{def}{=} \langle u(y), u(y') \rangle_{\mathcal{H}}$. A PDS kernel $k : \mathcal{Y} \times \mathcal{Y} \to \mathbb{R}$ is called characteristic if the kernel mean embedding $\mathcal{P}(\mathcal{Y}) \ni \mu \mapsto u(\mu) \stackrel{def}{=} \int_{\mathcal{X}} u(y) d\mu(y) \in \mathcal{H}$ is a one-to-one mapping. For a function $\phi : \mathbb{R}^D \to \mathbb{R} \cup \{\infty\}$, we denote its convex conjugate by $\overline{\phi}(y) \stackrel{def}{=} \sup_{x \in \mathbb{R}^D} \{\langle x, y \rangle - \phi(x)\}$.

## 2 BACKGROUND ON OPTIMAL TRANSPORT

**Strong OT formulation**. For distributions $\mathbb{P} \in \mathcal{P}(\mathcal{X}), \mathbb{Q} \in \mathcal{P}(\mathcal{Y})$ and a cost function $c : \mathcal{X} \times \mathcal{Y} \to \mathbb{R}$, Kantorovich's (Villani, 2008) primal formulation of the optimal transport cost (Figure 2a) is

$$\text{Cost}(\mathbb{P}, \mathbb{Q}) \stackrel{def}{=} \inf_{\pi \in \Pi(\mathbb{P}, \mathbb{Q})} \int_{\mathcal{X} \times \mathcal{Y}} c(x, y) d\pi(x, y), \tag{1}$$

where the minimum is taken over all transport plans $\pi$, i.e., distributions on $\mathcal{X} \times \mathcal{Y}$ whose marginals are $\mathbb{P}$ and $\mathbb{Q}$. The optimal $\pi^* \in \Pi(\mathbb{P}, \mathbb{Q})$ is called the optimal transport *plan*. A popular example of an OT cost for $\mathcal{X} = \mathcal{Y} = \mathbb{R}^D$ is the Wasserstein-2 ($\mathbb{W}_2^2$), i.e., formulation (1) for $c(x, y) = \frac{1}{2}\|x - y\|^2$.

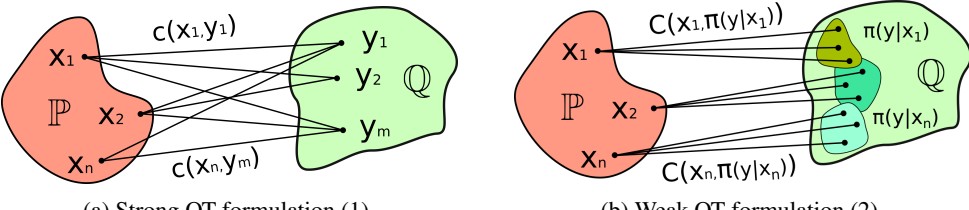

(a) Strong OT formulation (1).      (b) Weak OT formulation (2).

Figure 2: Strong (Kantorovich's) and weak (Gozlan et al., 2017) optimal transport formulations.

**Weak OT formulation**. Let $C : \mathcal{X} \times \mathcal{P}(\mathcal{Y}) \to \mathbb{R}$ be a *weak* cost (Gozlan et al., 2017), i.e., a function which takes a point $x \in \mathcal{X}$ and a distribution of $y \in \mathcal{Y}$ as inputs. The weak OT cost between $\mathbb{P}, \mathbb{Q}$ is

$$\text{Cost}(\mathbb{P}, \mathbb{Q}) \stackrel{def}{=} \inf_{\pi \in \Pi(\mathbb{P}, \mathbb{Q})} \int_{\mathcal{X}} C\big(x, \pi(\cdot|x)\big) d\pi(x), \tag{2}$$

where $\pi(\cdot|x)$ denotes the conditional distribution (Figure 2b). Weak OT (2) subsumes strong OT formulation (1) for $C(x, \mu) = \int_{\mathcal{Y}} c(x, y) d\mu(y)$. An example of a weak OT cost for $\mathcal{X} = \mathcal{Y} = \mathbb{R}^D$ is the $\gamma$-weak ($\gamma \geq 0$) Wasserstein-2 ($\mathcal{W}_{2,\gamma}$), i.e., formulation (2) with the $\gamma$-weak quadratic cost

$$C_{2,\gamma}(x, \mu) \stackrel{def}{=} \int_{\mathcal{Y}} \frac{1}{2}\|x - y\|^2 d\mu(y) - \frac{\gamma}{2}\text{Var}(\mu) = \int_{\mathcal{Y}} \frac{1}{2}\|x - \int_{\mathcal{Y}} y \, d\mu(y)\|^2 d\mu(y) + \frac{1 - \gamma}{2}\text{Var}(\mu), \tag{3}$$

where $\text{Var}(\mu)$ denotes the variance of $\mu$:

$$\text{Var}(\mu) \stackrel{def}{=} \int_{\mathcal{Y}} \|y - \int_{\mathcal{Y}} y' d\mu(y')\|^2 d\mu(y) = \frac{1}{2} \int_{\mathcal{Y} \times \mathcal{Y}} \|y - y'\|^2 d\mu(y) d\mu(y'). \tag{4}$$

For $\gamma = 0$, the transport cost (3) is strong, i.e., $\mathbb{W}_2 = \mathcal{W}_{2,0}$.

If the cost $C(x, \mu)$ is lower bounded, lower-semicontinuous and convex in the second argument, then we say that it is *appropriate*. For appropriate costs, the minimizer $\pi^*$ of (2) always exists (Backhoff-Veraguas et al., 2019, §1.3.1). Since $\text{Var}(\mu)$ is concave and non-negative, cost (3) is appropriate when $\gamma \in [0, 1]$. For appropriate costs the (2) admits the following dual formulation:

$$\text{Cost}(\mathbb{P}, \mathbb{Q}) = \sup_f \int_{\mathcal{X}} f^C(x) d\mathbb{P}(x) + \int_{\mathcal{Y}} f(y) d\mathbb{Q}(y), \tag{5}$$

where $f$ are upper-bounded, continuous and not rapidly decreasing functions, see (Backhoff-Veraguas et al., 2019, §1.3.2), and $f^C(x) \overset{\text{def}}{=} \inf_{\mu \in \mathcal{P}(\mathcal{Y})} \{C(x, \mu) - \int_{\mathcal{Y}} f(y) d\mu(y)\}$ is the weak $C$-transform.

**Neural Optimal Transport (NOT)**. In (Korotin et al., 2023), the authors propose an algorithm to *implicitly* learn an OT plan $\pi^*$ with neural nets (Figure 3). They introduce a (latent) atomless distribution $\mathbb{S} \in \mathcal{P}(\mathcal{Z})$, e.g., $\mathcal{Z} = \mathbb{R}^Z$ and $\mathbb{S} = \mathcal{N}(0, I_Z)$, and search for a function $T^* : \mathcal{X} \times \mathcal{Z} \to \mathcal{Y}$ (*stochastic OT map*) which satisfies $T_x^* \sharp \mathbb{S} = \pi^*(y|x)$ for some OT plan $\pi^*$. That is, given $x \in \mathcal{X}$, function $T^*$ pushes the distribution $\mathbb{S}$ to the *conditional distribution* $\pi^*(y|x)$ of an OT plan $\pi^*$. In particular, $T^*$ satisfies the distribution-preserving condition $T^* \sharp (\mathbb{P} \times \mathbb{S}) = \mathbb{Q}$. To get $T^*$, the authors use (5) to derive an equivalent dual form:

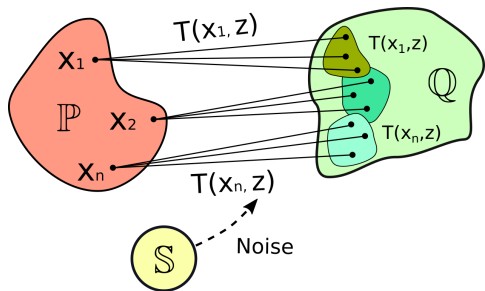

Figure 3: Implicit representation of a transport plan via function $T : \mathcal{X} \times \mathcal{Z} \to \mathcal{Y}$.

$$\text{Cost}(\mathbb{P}, \mathbb{Q}) = \sup_f \inf_T \int_{\mathcal{X}} \left( C(x, T_x \sharp \mathbb{S}) - \int_{\mathcal{Z}} f(T_x(z)) d\mathbb{S}(z) \right) d\mathbb{P}(x) + \int_{\mathcal{Y}} f(y) d\mathbb{Q}(y), \tag{6}$$

where the inf is taken over measurable functions $T : \mathcal{X} \times \mathcal{Z} \to \mathcal{Y}$. The functional under $\sup_f \inf_T$ is denoted by $\mathcal{L}(f, T)$. For every optimal potential $f^* \in \arg\sup_f \inf_T \mathcal{L}(f, T)$, it holds that

$$T^* \in \arg\inf_T \mathcal{L}(f^*, T), \tag{7}$$

see (Korotin et al., 2023, Lemma 4). Consequently, one may extract optimal maps $T^*$ from optimal saddle points $(f^*, T^*)$ of problem (6). In practice, saddle point problem (6) can be approached with neural nets $f_\omega, T_\theta$ and the stochastic gradient descent-ascent (Korotin et al., 2023, Algorithm 1).

The limitation (Korotin et al., 2023, §6) of NOT algorithm is that $\arg\inf_T$ set of $f^*$ in (7) may contain not only optimal transport maps but other functions as well. As a result, the function $T^*$ recovered from a saddle point $(f^*, T^*)$ may be not an optimal stochastic map. In this paper, we show that for the $\gamma$-weak quadratic cost (3) this may be *problematic*: the $\arg\inf_T$ sets might contain *fake* solutions $T^*$ (§3.1). To resolve the issue, we propose kernel $\gamma$-weak quadratic costs (§3.2).

**Convex order**. For two probability distributions $\mathbb{P}, \mathbb{Q}$ on $\mathbb{R}^D$, we write $\mathbb{P} \preceq \mathbb{Q}$ if for all convex functions $h : \mathbb{R}^D \to \mathbb{R}$ it holds $\int h(x) d\mathbb{P}(x) \leq \int h(x) d\mathbb{Q}(x)$. The relation "$\preceq$" is a partial order, i.e., not all $\mathbb{P}, \mathbb{Q}$ are comparable. If $\mathbb{P} \preceq \mathbb{Q}$, then $m_{\mathbb{P}} = m_{\mathbb{Q}}$ and $\Sigma_{\mathbb{P}} \preceq \Sigma_{\mathbb{Q}}$ (Scarsini, 1998, Lemma 3). If $\mathbb{P}, \mathbb{Q}$ are Gaussians, $\mathbb{P} \preceq \mathbb{Q}$ holds if and only if $m_{\mathbb{P}} = m_{\mathbb{Q}}$ and $\Sigma_{\mathbb{P}} \preceq \Sigma_{\mathbb{Q}}$ (Scarsini, 1998, Theorem 4). For $\mathbb{P}, \mathbb{Q}$, we define the *projection* of $\mathbb{P}$ onto the convex set of distributions which are $\preceq \mathbb{Q}$ as

$$\text{Proj}_{\preceq \mathbb{Q}}(\mathbb{P}) = \arg\inf_{\mathbb{P}' \preceq \mathbb{Q}} \mathbb{W}_2^2(\mathbb{P}, \mathbb{P}'). \tag{8}$$

The infimum is attained uniquely (Gozlan & Juillet, 2020, Proposition 1.1). There exists a 1-Lipschitz, continuously differentiable and convex function $\phi : \mathbb{R}^D \to \mathbb{R}$ satisfying $\text{Proj}_{\preceq \mathbb{Q}}(\mathbb{P}) = \nabla\phi\sharp\mathbb{P}$, see (Gozlan & Juillet, 2020, Theorem 1.2) for details.

**Weak OT with the quadratic cost** ($\gamma > 0$). For the $\gamma$-weak quadratic cost (3) on $\mathcal{X} = \mathcal{Y} = \mathbb{R}^D$, (Gozlan & Juillet, 2020, §5), (Alibert et al., 2019, §5.2) prove that there exists a continuously differentiable convex function $\phi^* : \mathbb{R}^D \to \mathbb{R}$ such that $\pi^* \in \Pi(\mathbb{P}, \mathbb{Q})$ is optimal if and only if $\int_{\mathcal{Y}} y \, d\pi^*(y|x) = \nabla\phi^*(x)$ holds true $\mathbb{P}$-almost surely. In general, $\pi^*$ is *not unique*. We say that $\phi^*$ is an *optimal restricted potential*. It may be not unique as a function $\mathbb{R}^D \to \mathbb{R}$, but $\nabla\phi^*$ is uniquely defined $\mathbb{P}$-almost everywhere. It holds true

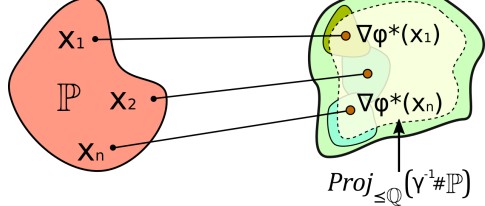

Figure 4: Every optimal restricted potential $\phi^*$ satisfies $\nabla\phi^*\sharp\mathbb{P} = \text{Proj}_{\preceq \mathbb{Q}}(\frac{1}{\gamma}\sharp\mathbb{P}) \preceq \mathbb{Q}$.

that $\nabla\phi^*$ is $\frac{1}{\gamma}$-Lipschitz; $\nabla\phi^*\sharp\mathbb{P} = \mathrm{Proj}_{\preceq\mathbb{Q}}(\frac{1}{\gamma}\sharp\mathbb{P})$; it is the OT map between $\mathbb{P}$ and $\mathrm{Proj}_{\preceq\mathbb{Q}}(\frac{1}{\gamma}\sharp\mathbb{P})$ for the strong quadratic cost (Figure 4). The function $\phi^*$ maximizes the dual form alternative to (5):

$$\mathcal{W}_{2,\gamma}^2(\mathbb{P},\mathbb{Q}) = \max_{\phi\in\mathrm{Cvx}(\frac{1}{\gamma})}\left[\int_{\mathcal{X}}\frac{\|x\|^2}{2}d\mathbb{P}(x)+\int_{\mathcal{Y}}\frac{\|y\|^2}{2}d\mathbb{Q}(x)-\left(\int_{\mathcal{X}}\phi(x)d\mathbb{P}(x)+\int_{\mathcal{Y}}\overline{\phi}(y)d\mathbb{Q}(y)\right)\right], \quad (9)$$

where $\mathrm{Cvx}(\frac{1}{\gamma})$ denotes the set of $\frac{1}{\gamma}$-smooth convex functions $\phi : \mathbb{R}^D \to \mathbb{R}$. Duality formula (9) appears in (Alibert et al., 2019, §5.2), (Gozlan & Juillet, 2020, Theorem 1.2 & §5) but with different parametrization. In Appendix F, for completeness of the exposition, we underline{derive} (9) from the results of (Gozlan & Juillet, 2020) by the change of variables.

## 3    SOLVING ISSUES OF NEURAL OPTIMAL TRANSPORT

In what follows, we consider $\mathcal{X} = \mathcal{Y} \subset \mathbb{R}^D$. In §3.1, we theoretically derive that $\arg\inf_T$ sets (7) for the $\gamma$-weak quadratic cost (3) may indeed contain functions which are not stochastic OT maps. In §3.2, we introduce kernel weak quadratic costs and prove that they do not suffer from this issue, i.e., all functions in sets $\arg\inf_T$ for all optimal $f^*$ are stochastic OT maps. In §3.3, we discuss the practical aspects of learning with kernels. We give the _proofs_ of all the statements in Appendix G.

### 3.1    FAKE SOLUTIONS FOR THE WEAK QUADRATIC COST

In this subsection, we consider $\mathcal{X} = \mathcal{Y} = \mathbb{R}^D$. We show that $\arg\inf_T \mathcal{L}(f^*, T)$ sets (7) of optimal potentials $f^*$ in (5) for the $\gamma$-weak quadratic cost (3), in general, may contain functions $T$ which are not stochastic OT maps. We call such $T$ _fake solutions_. To show why one should be concerned about fake solutions, we emphasize their key defect below.

**Lemma 1** (Fake solutions are not distribution-preserving). _Let $f \in \arg\sup_f\inf_T \mathcal{L}(f, T)$ and $T^\dagger \in \arg\inf_T \mathcal{L}(f^*, T)$ be a fake solution. Then it holds that $T^\dagger\sharp(\mathbb{P}\times\mathbb{S}) \neq \mathbb{Q}$._

Throughout the section, we assume that $\mathbb{P}, \mathbb{Q}$ have finite second moments. We analyse the potentials of the form $f^*(y) = \frac{1}{2}\|y\|^2 - \overline{\phi^*}(y)$, where $\phi^* : \mathbb{R}^D \to \mathbb{R}$ is an optimal restricted potential. To begin with, we show that such potentials are indeed optimal potentials for dual forms (5) and (6).

**Lemma 2** (Optimal restricted potentials provide dual form maximizers). _Let $\phi^* : \mathbb{R}^D \to \mathbb{R}$ be an optimal restricted potential. Assume that $\overline{\phi^*}$ takes only finite values. Then $f^*(y) \stackrel{def}{=} \frac{1}{2}\|y\|^2 - \overline{\phi^*}(y)$ maximizes dual formulations (5) and (6)._

For a convex $\psi : \mathbb{R}^D \to \mathbb{R}$, we denote the area around point $y \in \mathbb{R}^D$ in which $\psi$ is linear by

$$U_\psi(y) = \{y' \in \mathbb{R}^D \text{ such that } \forall x \in \partial_y\psi(y) \text{ it holds } \psi(y') = \psi(y) + \langle x, y' - y\rangle\} \supseteq \{y\}. \quad (10)$$

**Proposition 1** (Convexity of sets of local linearity of a convex function). _Set $U_\psi(y)$ is convex._

Our following theorem provides a full characterization of the $\arg\inf_T \mathcal{L}(f^*, T)$ sets in view.

**Theorem 1** (Characterization of saddle points with optimal restricted $f^*$). _Let $f^*(y) = \frac{\|y\|^2}{2} - \overline{\phi^*}(y)$, where $\phi^*$ is an optimal restricted potential. Assume that $\overline{\phi^*}$ takes only finite values. Then it holds true that $\arg\inf_T \mathcal{L}(f^*, T)$ is a **convex set** and_

$$T^\dagger \in \underset{T}{\arg\inf}\, \mathcal{L}(f^*, T) \Leftrightarrow \begin{cases} \int_{\mathcal{Z}} T_x^\dagger(z)d\mathbb{S}(z) = \nabla\phi^*(x) & \text{holds true } \mathbb{P}\text{-almost everywhere}; \\ T_x^\dagger(z) \in U_\psi\big(\nabla\phi^*(x)\big) & \text{holds true } \mathbb{P}\times\mathbb{S}\text{-almost everywhere}, \end{cases} \quad (11)$$

_where $\psi(y) \stackrel{def}{=} \overline{\phi^*}(y) - \frac{\gamma}{2}\|y\|^2$. Note that $\psi$ is convex since $\phi^*$ is $\frac{1}{\gamma}$-smooth (Kakade et al., 2009)._

We define the _optimal barycentric projection_ $\overline{T_x^*}(z) \stackrel{def}{=} \nabla\phi^*(x)$ for $(x, z) \in \mathcal{X}\times\mathcal{Z}$; it does not depend on $z \in \mathcal{Z}$. The function depends on the choice of optimal $\phi^*$; we are interested only in its values in the support of $\mathbb{P}$, where $\nabla\phi^*$ is unique (§2). From definition (10), we see that $\nabla\phi^*(x) \in U_\psi\big(\nabla\phi^*(x)\big)$ and $\overline{T^*}$ satisfies both conditions on the right side of (11). Thus, we have $\overline{T^*} \in \arg\inf_T \mathcal{L}(f^*, T)$.

**Lemma 3** (The barycentric projection is not always a stochastic OT map). _The following holds true_

$$\overline{T^*} \text{ is a stochastic OT map } \overset{(a)}{\Longleftrightarrow} \mathrm{Proj}_{\preceq\mathbb{Q}}\Big(\frac{1}{\gamma}\sharp\mathbb{P}\Big) = \mathbb{Q} \overset{(b)}{\Longrightarrow} \overline{T^*} \text{ is the unique stochastic OT map.}$$

We use the word _stochastic_ but $\overline{T^*}$ is actually deterministic since it does not depend on $z$. From our Lemma 3, we derive that if $\mathrm{Proj}_{\preceq\mathbb{Q}}(\frac{1}{\gamma}\sharp\mathbb{P}) \neq \mathbb{Q}$, it holds that $(f^*, \overline{T^*})$ is a fake saddle point.

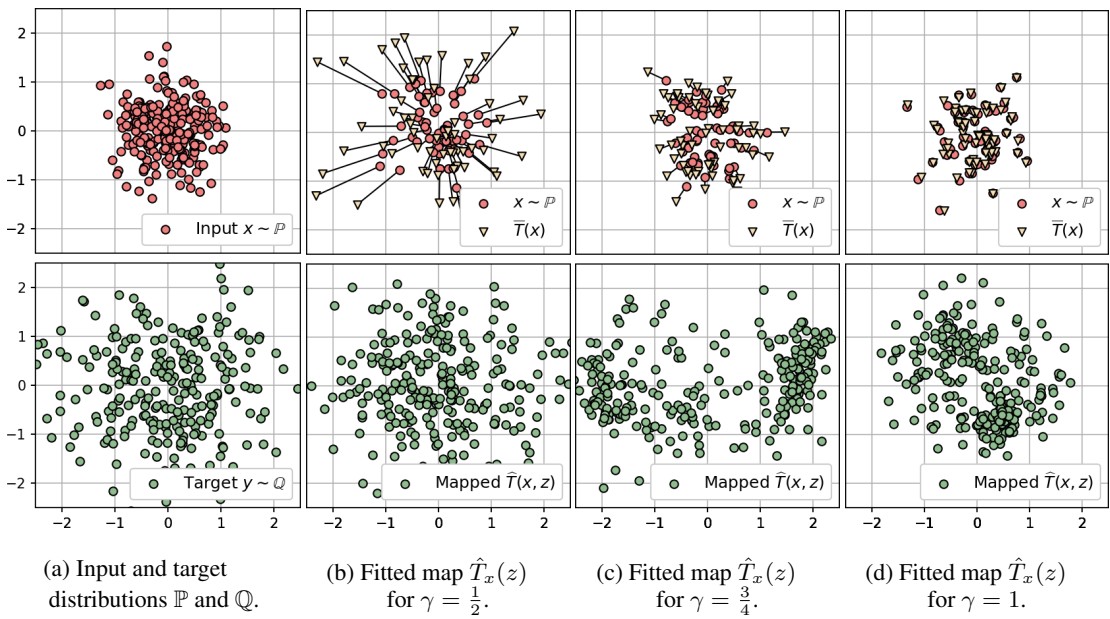

(a) Input and target distributions $\mathbb{P}$ and $\mathbb{Q}$.

(b) Fitted map $\hat{T}_x(z)$ for $\gamma = \frac{1}{2}$.

(c) Fitted map $\hat{T}_x(z)$ for $\gamma = \frac{3}{4}$.

(d) Fitted map $\hat{T}_x(z)$ for $\gamma = 1$.

Figure 5: Stochastic maps $\widehat{T}$ between $\mathbb{P}$ and $\mathbb{Q}$ fitted by NOT algorithm with costs $C_{2,\gamma}$ for various $\gamma$.

**Corollary 1** (Existence of fake saddle points). *Assume that* $\mathrm{Proj}_{\preceq\mathbb{Q}}(\frac{1}{\gamma}\sharp\mathbb{P}) \neq \mathbb{Q}$. *Then problem* (6) *has optimal saddle points* $(f^*, T^*)$ *in which* $T^*$ *is not a stochastic OT map.*

Beside $\overline{T^*}$, our Theorem 1 can be used to construct arbitrary many fake solutions which are not OT maps. Let $T^*$ be *any* stochastic OT map and $T^\dagger \in \arg\inf_T \mathcal{L}(f^*, T)$ satisfy $T^\dagger \neq T^*$ and $\mathrm{Var}\left(T^\dagger\sharp(\mathbb{P}\times\mathbb{S})\right) \leq \mathrm{Var}(\mathbb{Q})$. For example, $T^\dagger$ may be another stochastic OT map or the optimal barycentric projection $\overline{T^*}$. For any $\alpha \in (0,1)$ consider $T^\alpha = \alpha T^* + (1-\alpha)T^\dagger \in \arg\inf_T \mathcal{L}(f^*, T)$.

**Proposition 2** (Interpolant is not a stochastic OT map). *Assume that* $\mathrm{Proj}_{\preceq\mathbb{Q}}(\frac{1}{\gamma}\sharp\mathbb{P}) \neq \mathbb{Q}$. *Then* $T^\alpha\sharp(\mathbb{P}\times\mathbb{S}) \neq \mathbb{Q}$. *Consequently,* $T^\alpha$ *is not a stochastic OT map between* $\mathbb{P}$ *and* $\mathbb{Q}$.

It follows that a *necessary* condition for non-existence of fake saddle points is $\mathrm{Proj}_{\preceq\mathbb{Q}}(\frac{1}{\gamma}\sharp\mathbb{P}) = \mathbb{Q}$. This requirement is very restrictive and naturally prohibits using large values of $\gamma$. Also, due to our Lemma 3, the OT plan between $\mathbb{P}, \mathbb{Q}$ must be deterministic. From the practical point of view, this requirement means that there will be no diversity in samples $T^*_x(z)$ for a fixed $x$ and $z \sim \mathbb{S}$.

On the other hand, if $\mathrm{Proj}_{\preceq\mathbb{Q}}(\frac{1}{\gamma}\sharp\mathbb{P}) \neq \mathbb{Q}$, i.e., $\mathbb{P}$ is not $\gamma$-*times more disperse* than $\mathbb{Q}$, the optimization may indeed converge to fake solutions. To show this, we consider the following example.

**Toy 2D example**. We consider $\mathbb{P} = \mathcal{N}(0, [\frac{1}{2}]^2 I_2)$, $\mathbb{Q} = \mathcal{N}(0, I_2)$ (Figure 5a) and run NOT (Korotin et al., 2023, Algorihm 1) for $\gamma \in \{\frac{1}{2}, \frac{3}{4}, 1\}$-weak quadratic costs $C_{2,\gamma}$. We show the learned stochastic maps $\hat{T}_x(z)$ and their barycentric projections $\overline{T}(x) \stackrel{def}{=} \int_{\mathcal{Z}} \hat{T}_x(z)d\mathbb{S}(z)$ in Figures 5b, 5c, 5d.

**Good case**. When $\gamma \leq \frac{1}{2}$, we have $\frac{1}{\gamma}\sharp\mathbb{P} = \mathcal{N}(0, [\frac{1}{2\gamma}]^2 I_2)$ with $\frac{1}{2\gamma} \geq 1$. Since the distributions $\frac{1}{\gamma}\sharp\mathbb{P}$ and $\mathbb{Q}$ are Gaussians and $\Sigma_\mathbb{Q} \preceq \Sigma_{\frac{1}{\gamma}\sharp\mathbb{P}}$, we conclude that $\mathrm{Proj}_{\preceq\mathbb{Q}}(\frac{1}{\gamma}\sharp\mathbb{P}) = \mathbb{Q}$ (Gozlan & Juillet, 2020, Corollary 2.1). Next, we use our Lemma 3 and derive that the OT plan is unique, deterministic and equals the barycentric projection. The latter is the OT map between $\mathbb{P}$ and $\mathbb{Q}$ for the quadratic cost. It is given by $\nabla\phi^*(x) = 2x$ (Álvarez-Esteban et al., 2016, Theorem 2.3). In Figure 5b (when $\gamma = \frac{1}{2}$), we have $\widehat{T}_x(z) = \overline{T}(x) \approx 2x = \nabla\phi^*(x)$ and $\widehat{T}\sharp(\mathbb{P}\times\mathbb{S}) \approx \mathbb{Q}$. Thus, NOT correctly learns the (unique and deterministic) OT plan.

**Bad case**. When $\gamma > \frac{1}{2}$, we have $\frac{1}{\gamma}\sharp\mathbb{P} = \mathcal{N}(0, [\frac{1}{2\gamma}]^2 I_2)$ with $\frac{1}{2\gamma} < 1$. Since $\frac{1}{\gamma}\sharp\mathbb{P}$ and $\mathbb{Q}$ are Gaussians and $\Sigma_\mathbb{Q} \succeq \Sigma_{\frac{1}{\gamma}\sharp\mathbb{P}}$, we conclude that $\frac{1}{\gamma}\mathbb{P} \preceq \mathbb{Q}$ (recall §2). Thus, $\mathrm{Proj}_{\preceq\mathbb{Q}}(\frac{1}{\gamma}\sharp\mathbb{P}) = \frac{1}{\gamma}\sharp\mathbb{P} \neq \mathbb{Q}$ by definition of the projection (8). The optimal barycentric projection is the OT map between Gaussians $\mathbb{P}$ and $\frac{1}{\gamma}\sharp\mathbb{P}$ for the quadratic cost. It is given by $\nabla\phi^*(x) = \frac{1}{\gamma}x$ (Álvarez-Esteban et al., 2016, Theorem 2.3). In Figures 5c and 5d, we see that the learned $\overline{T}(x) \approx \frac{1}{\gamma}x$, i.e., $\widehat{T}$ captures the

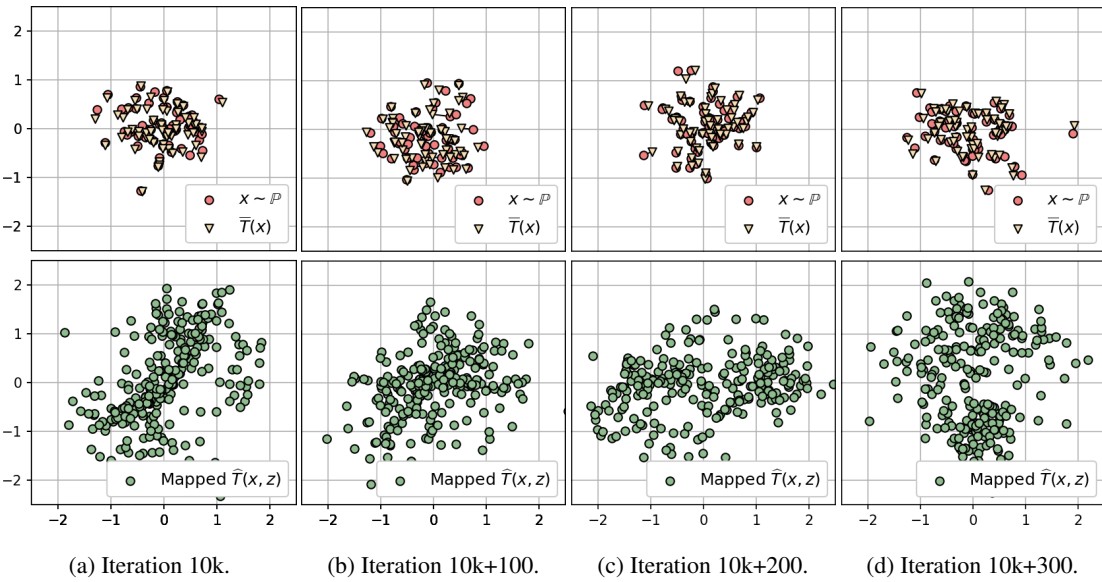

(a) Iteration 10k.   (b) Iteration 10k+100.   (c) Iteration 10k+200.   (d) Iteration 10k+300.

Figure 6: The evolution of the learned transport map $\widehat{T}$ during training on a toy 2D example ($\gamma = 1$). conditional expectation of $\pi^*(y|x)$ shared by all OT plans $\pi^*$ (§2). However, $\widehat{T}\sharp(\mathbb{P}\times\mathbb{S}) \neq \mathbb{Q}$ and NOT *fails to learn* an OT plan.

Importantly, we found that when $\gamma > \frac{1}{2}$ (Figures 5c, 5d), the transport map $\hat{T}$ extremely *fluctuates* during the optimization rather than converges to a solution. In Figure 6, we visualize the evolution of $\widehat{T}$ during training (for $\gamma = 1$). In all the cases, the barycentric projection $\overline{T}(x) \approx x = \nabla\phi^*(x)$ is almost correct. However, the "remaining" part of $\widehat{T}$ is literally random. To explain the behavior, we integrate $\nabla\phi^*(x) = x$ and get that $\phi^*(x) = \frac{1}{2}\|x\|^2$ is an optimal restricted potential. We derive $\psi(y) = \overline{\phi^*}(y) - \frac{1}{2}\|y\|^2 = \frac{1}{2}\|y\|^2 - \frac{1}{2}\|y\|^2 \equiv 0 \implies U_\psi(y) = U_0(y) \equiv \mathbb{R}^D$ for every $y \in \mathbb{R}^D$. From our Theorem 1 it follows that $T^\dagger \in \arg\inf_T \mathcal{L}(f^*, T) \Leftrightarrow \int_{\mathcal{Z}} T_x^\dagger(z)d\mathbb{S}(z) = \nabla\phi^*(x) = x$ holds $\mathbb{P}$-almost everywhere. Thus, a function $T^\dagger$ recovered from (6) may be literally any function which captures the first conditional moment of a plan $\pi^*(y|x)$. This agrees with our practical observations. In Appendix C, we give an additional toy example illustrating the issue with fake solutions.

Our results show that the $\gamma$-weak quadratic cost $C_{2,\gamma}$ may be not a good choice for NOT algorithm due to fake solutions. However, prior works on OT (Korotin et al., 2023; Rout et al., 2022; Fan et al., 2022a; Gazdieva et al., 2022; Korotin et al., 2022) use strong/weak quadratic costs and show promising practical performance. *Should we really care about solutions being fake?*

**Yes**. First, fake solutions $T^* \in \arg\inf_T \mathcal{L}(f^*, T)$ do not satisfy $T^*\sharp(\mathbb{P}\times\mathbb{S}) \neq \mathbb{Q}$, i.e., they are *not distribution preserving* (Lemma 1). Second, our analysis suggests that fake solutions might be one of the *causes* for the training instabilities reported in related works (Korotin et al., 2023, Appendix D), (Korotin et al., 2021b, §4): the map $\widehat{T}$ may fluctuate between fake solutions rather than converge.

### 3.2 KERNEL WEAK QUADRATIC COST REMOVES FAKE SADDLE POINTS

In this section, we introduce kernel weak quadratic costs which generalize weak quadratic cost (3). We prove that for characteristic kernels the costs completely resolve the ambiguity of $\arg\inf_T$ sets.

Henceforth, we assume that $\mathcal{X} = \mathcal{Y} \subset \mathbb{R}^D$ are compact sets. Let $\mathcal{H}$ be a Hilbert space (feature space). Let $u : \mathcal{X} \to \mathcal{H}$ be a function (feature map). We define the $\gamma$-weak quadratic cost between features:

$$C_{u,\gamma}(x, \mu) \stackrel{def}{=} \frac{1}{2}\int_{\mathcal{Y}} \|u(x) - u(y)\|_{\mathcal{H}}^2 d\mu(y) - \frac{\gamma}{2}\cdot\left[\frac{1}{2}\int_{\mathcal{Y}\times\mathcal{Y}} \|u(y) - u(y')\|_{\mathcal{H}}^2 d\mu(y)d\mu(y')\right]. \quad (12)$$

We denote the PDS kernel $k : \mathcal{Y} \times \mathcal{Y} \to \mathbb{R}$ with the feature map $u$ by $k(y, y') \stackrel{def}{=} \langle u(y), u(y')\rangle_{\mathcal{H}}$. Cost (12) can be computed without knowing the map $u$, i.e., it is enough to know the PDS kernel $k$. By using $\|u(y) - u(y')\|_{\mathcal{H}}^2 = k(y, y) - 2k(y, y') + k(y', y')$, we obtain the equivalent form of (12):

$$C_{k,\gamma}(x, \mu) \stackrel{def}{=} \frac{1}{2}k(x, x) + \frac{1-\gamma}{2}\int_{\mathcal{Y}} k(y, y)d\mu(y) - \int_{\mathcal{Y}} k(x, y)d\mu(y) + \frac{\gamma}{2}\int_{\mathcal{Y}\times\mathcal{Y}} k(y, y')d\mu(y)d\mu(y'). \quad (13)$$

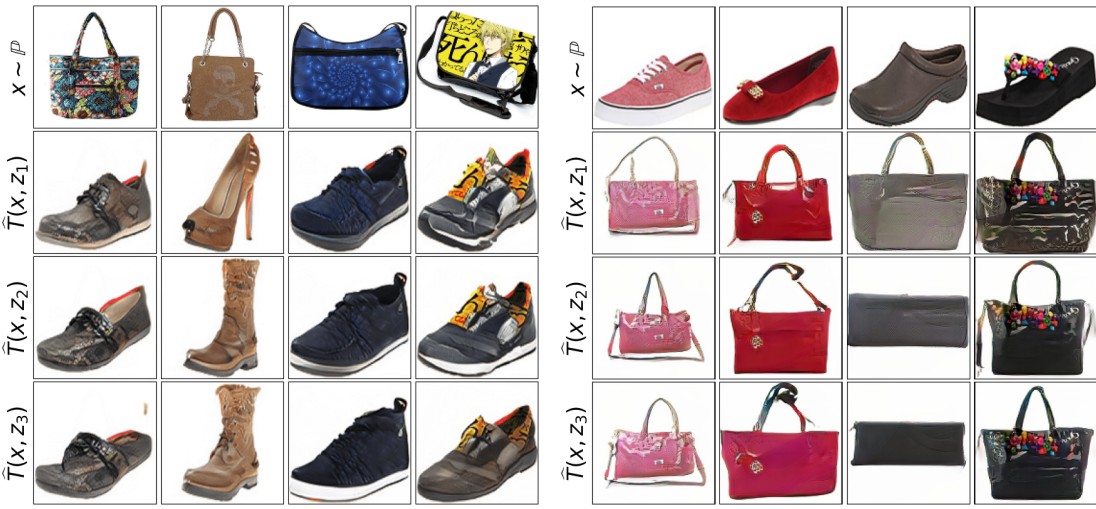

(a) Handbag → shoes, $128 \times 128$.    (b) Shoes → handbags, $128 \times 128$.

Figure 7: Unpaired one-to-many translation with kernel Neural Optimal Transport (NOT).

We call (12) and (13) the $\gamma$-*weak kernel cost*. The $\gamma$-weak quadratic cost (3) is its particular case for $\mathcal{H} = \mathbb{R}^D$ and $u(x) = x$. The respective kernel $k(x, y) = \langle u(x), u(y) \rangle = \langle x, y \rangle$ is bilinear.

**Lemma 4** (Weak kernel costs are appropriate). *Let $k$ be a continuous PDS kernel and $\gamma \in [0, 1]$. Then the cost $C_{k,\gamma}(x, \mu)$ is convex, lower semi-continuous and lower bounded in $\mu$.*

**Corollary 2** (Existence and duality for kernel costs). *Let $k$ be a continuous PDS kernel and $\gamma \in [0, 1]$. Then an OT plan $\pi^*$ for cost $C_{k,\gamma}(x, \mu)$ exists and duality formulas (5) and (6) hold true.*

We focus on characteristic kernels $k$ and show that they resolve the ambiguity in $\arg\inf_T$ sets.

**Lemma 5** (Uniqueness of the optimal plan for characteristic kernel costs). *Let $k$ be a characteristic PDS kernel and $\gamma \in (0, 1]$. Then the OT plan $\pi^*$ for cost $C_{k,\gamma}(x, \mu)$ is unique.*

**Theorem 2** (Optimality of stochastic functions in all optimal saddle points). *Let $k$ be a continuous characteristic PDS kernel and $\gamma \in (0, 1]$. Consider weak OT problem (2) with cost $C_{k,\gamma}$ and its dual problem (5). For any optimal potential $f^* \in \arg\sup_f \inf_T \mathcal{L}(f, T)$ it holds that*

$$T^* \in \arg\inf_T \mathcal{L}(f^*, T) \iff T_x^* \sharp \mathbb{S} = \pi^*(y|x) \text{ holds true } \mathbb{P}\text{-almost surely for all } x \in \mathcal{X}, \quad (14)$$

*i.e., every optimal saddle point $(f^*, T^*)$ provides a stochastic OT map $T^*$.*

Bilinear kernel $k(x, y) = \langle x, y \rangle$ is not characteristic and is not covered by our Theorem 2; its respective $\gamma$-weak quadratic cost $C_{2,\gamma}$ suffers from fake solutions (§3.1). In the next subsection, we give examples of practically interesting kernels $k(x, y)$ which are ideologically similar to the bilinear but are characteristic. Consequently, their respective costs $C_k$ do not have ambiguity in $\arg\inf_T$ sets.

### 3.3 PRACTICAL ASPECTS OF LEARNING WITH KERNEL COSTS

**Optimization.** To learn the stochastic OT map $T^*$ for kernel cost (13), we use NOT's training procedure (Korotin et al., 2023, Algorithm 1). It requires stochastic estimation of $C_{k,\gamma}(x, T_x \sharp \mathbb{S})$ to compute the corresponding term in (6). Similar to the $\gamma$-weak quadratic cost (Korotin et al., 2023, Equation 23), it is possible to derive the following *unbiased* Monte-Carlo estimator $\widehat{C}_{k,\gamma}$ for $x \in \mathcal{X}$ and a batch $Z \sim \mathbb{S}$ ($|Z| \geq 2$):

$$\widehat{C}_{k,\gamma}\big(x, T_x(Z)\big) = \frac{1}{2}k(x, x) + \frac{1-\gamma}{2|Z|} \sum_{z \in Z} k\big(T_x(z), T_x(z)\big) - $$

$$\frac{1}{|Z|} \sum_{z \in Z} k(x, T_x(z)) + \frac{\gamma}{2|Z|(|Z|-1)} \sum_{z \neq z'} k\big(T_x(z), T_x(z')\big) \approx C_{k,\gamma}(x, T_x \sharp \mathbb{S}). \quad (15)$$

The time complexity of estimator (15) is $O(|Z|^2)$ since it requires considering pairs $z, z'$ in batch to estimate the variance term. Specifically for the bilinear kernel $k(y, y') = \langle y, y' \rangle$, the variance can be

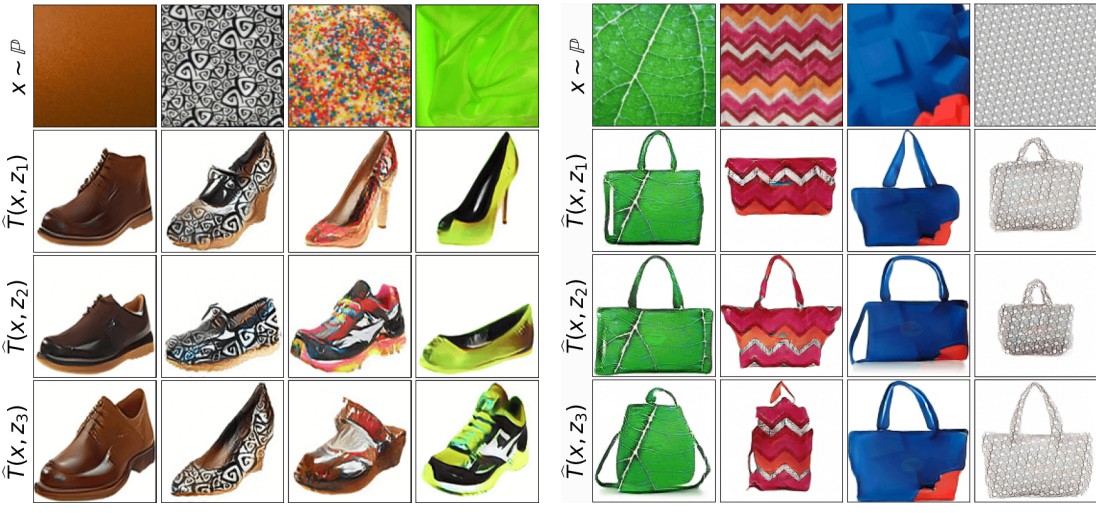

(a) Texture → shoes, $128 \times 128$.   (b) Texture → handbags, $128 \times 128$.

Figure 8: Unpaired one-to-many translation with kernel Neural Optimal Transport (NOT).

estimated in $O(|Z|)$ operations (Korotin et al., 2023, Equation 23), but NOT algorithm may suffer from fake solutions (§3.1).

**Kernels.** Consider the family of distance-induced kernels $k(x, y) = \frac{1}{2}\|x\|^\alpha + \frac{1}{2}\|y\|^\alpha - \frac{1}{2}\|x - y\|^\alpha$. For these kernels, we have $\|u(x) - u(x')\|_{\mathcal{H}}^2 = \|x - x'\|^\alpha$, i.e., (12), (13) can be expressed as

$$C_{k,\gamma}(x, \mu) = C_{u,\gamma}(x, \mu) = \frac{1}{2}\int_{\mathcal{Y}}\|x - y\|^\alpha d\mu(y) - \frac{\gamma}{2} \cdot \left[\frac{1}{2}\int_{\mathcal{Y}\times\mathcal{Y}}\|y - y'\|^\alpha d\mu(y)d\mu(y')\right]. \quad (16)$$

For $\alpha = 2$ the kernel is bilinear, i.e., $k(x, y) = \langle x, y \rangle$; it is PDS but not characteristic and (16) simply becomes the $\gamma$-weak quadratic cost (3). In the experiments (§5), we focus on the case $\alpha = 1$; it yields a PDS and characteristic kernel (Sejdinovic et al., 2013, Definition 13 & Proposition 14).

## 4 RELATED WORK

In deep learning, **OT costs** are primarily used as losses to train generative models. Such approaches are called Wasserstein GANs (Arjovsky & Bottou, 2017); they are **not related** to our paper since they only compute OT costs but not OT plans. Below we discuss methods to compute **OT plans**.

**Existing OT solvers.** NOT (Korotin et al., 2023) is the only parametric algorithm which is capable of computing OT plans for *weak* costs (2). Although NOT is generic, the authors tested it only with the $\gamma$-weak quadratic cost (3). The core of NOT is saddle point formulation (6) which subsumes analogs (Korotin et al., 2021b, Eq. 9), (Rout et al., 2022, Eq. 14), (Fan et al., 2022a, Eq. 11), (Henry-Labordere, 2019, Eq. 11), (Gazdieva et al., 2022, Eq. 10), (Korotin et al., 2022, Eq. 7) for *strong* costs (1). For the strong quadratic cost, (Makkuva et al., 2020), (Taghvaei & Jalali, 2019, Eq. 2.2), (Korotin et al., 2021a, Eq. 10) consider analogous to (9) formulations restricted to convex potentials; they use Input Convex Neural Networks (ICNNs (Amos et al., 2017)) to approximate the potentials. ICNNs are popular in OT (Korotin et al., 2021c; Mokrov et al., 2021; Huang et al., 2020; Alvarez-Melis et al., 2022; Bunne et al., 2022) but recent studies (Korotin et al., 2021b; 2022; Fan et al., 2022b) show that OT algorithms based on them underperform compared to unrestricted formulations such as NOT.

In (Genevay et al., 2016; Seguy et al., 2018; Daniels et al., 2021), the authors propose neural algorithms for $f$-divergence *regularized* costs (Genevay, 2019). The first two methods suffer from bias in high dimensions (Korotin et al., 2021b, §4.2). Algorithm (Daniels et al., 2021) alleviates the bias but is not end-to-end and is computationally expensive due to using the Langevin dynamics.

There also exist GAN-based (Goodfellow et al., 2014) methods (Lu et al., 2020; Xie et al., 2019; González-Sanz et al., 2022) to learn OT plans (or maps) for *strong* costs. However, they are harder to set up in practice due to the large amount of tunable hyperparameters.

**Kernels in OT.** In (Zhang et al., 2019; Oh et al., 2020), the authors propose a **strong** kernel $\mathbb{W}_2$ distance and an algorithm to approximate the transport map under the Gaussianity assumption on $\mathbb{P}, \mathbb{Q}$. In (Li et al., 2021), the authors generalize Sinkhorn divergences (Genevay et al., 2019) to Hilbert spaces. These papers consider discrete OT formulations and data-to-data matching tasks; they do not use neural networks to approximate the OT map.

## 5 EVALUATION

In Appendix A, we learn OT between *toy 1D distributions* and perform comparisons with discrete OT. In Appendix B, we conduct tests on *toy 2D distributions*. In this section, we test our algorithm on an *unpaired image-to-image translation* task. We perform *comparison* with principal translation methods in Appendix K. The code is written in `PyTorch` framework and is available at

```
https://github.com/iamalexkorotin/KernelNeuralOptimalTransport
```

**Image datasets.** We test the following datasets as $\mathbb{P}, \mathbb{Q}$: aligned anime faces[1], celebrity faces (Liu et al., 2015), shoes (Yu & Grauman, 2014), Amazon handbags, churches from LSUN dataset (Yu et al., 2015), outdoor images from the MIT places database (Zhou et al., 2014), describable textures (Cimpoi et al., 2014). The size of datasets varies from 5K to 500K images.

**Train-test split.** We pick 90% of each dataset for unpaired training. The rest 10% are considered as the test set. All the results presented here are *exclusively* for test images, i.e., *unseen data*.

**Transport costs.** We focus on the $\gamma$-weak cost for the kernel $k(x, y) = \frac{1}{2}\|x\| + \frac{1}{2}\|y\| - \frac{1}{2}\|x - y\|$. For completeness, we test *other popular PDS kernels* in Appendix E.

Other **training details** (optimizers, architectures, pre-processing, etc.) are given in Appendix I.

We learn stochastic OT maps between various pairs of datasets. We rescale images to $128 \times 128$ and use $\gamma = \frac{1}{3}$ in the experiments with the kernel cost. Additionally, in Appendix D we analyse how *varying parameter $\gamma$* affects the diversity of generated samples. We provide the qualitative results in Figures 1, 7 and 8; *extra results* are in Appendix L. Thanks to the first term in (16), our translation map $\hat{T}_x(z)$ tries to minimally change the image content $x$ in the pixel space. At the same time, the second term (kernel variance) in (16) enforces the map to produce diverse outputs for different $z \sim \mathbb{S}$.

We provide quantitative comparison with NOT with the $\gamma$-weak quadratic cost $C_{2,\gamma}$. We compute FID score (Heusel et al., 2017) between the mapped input test subset and the output test subset (Table 1). For $C_{2,\gamma}$, we use the pre-trained models provided by the authors of NOT (Korotin et al., 2023, §5).[2] We observe that FID of NOT with kernel cost $C_{k,\gamma}$ is better than that of NOT with cost $C_{2,\gamma}$. We show qualitative examples in Appendix J. In Appendix H, we perform a detailed comparison of NOT's *training stability* with the weak quadratic and kernel costs.

| **Datasets** ($128 \times 128$) | $C_{2,0}$ (strong) | $C_{2,\gamma}$ (weak) | $C_{k,\gamma}$ (weak) **Ours** |
|---|---|---|---|
| Handbags $\rightarrow$ shoes | 35.7 | $33.9 \pm 0.2$ | $\mathbf{26.7 \pm 0.06}$ |
| Shoes $\rightarrow$ handbags | 39.8 | – | $\mathbf{29.51 \pm 0.19}$ |
| Outdoor $\rightarrow$ church | 25.5 | $25.97 \pm 0.14$ | $\mathbf{15.16 \pm 0.03}$ |
| Celeba (f) $\rightarrow$ anime | 38.73 | $28.21 \pm 0.12$ | $\mathbf{21.96 \pm 0.07}$ |

Table 1: Test **FID**$\downarrow$ of NOT with various costs.

## 6 DISCUSSION

**Potential impact.** Neural OT methods and their usage in generative models constantly advance. We expect our proposed weak kernel quadratic costs to improve applications of OT to unpaired learning. In particular, we hope that our theoretical analysis provides better understanding of the performance.

**Limitations (theory).** In our Theorem 2, we implicitly assume the existence a maximizer $f^*$ of dual form (5) for kernel costs $C_{k,\gamma}$. Deriving precise conditions for existence of such maximizers is a challenging question. We hope that this issue will be addressed in the future theoretical OT research.

**Limitations (practice).** Applying kernel costs to domains of different nature (*RGB images $\rightarrow$ depth maps*, *infrared images $\rightarrow$ RGB images*) is not straightforward as it might require selecting meaningful *shared* features $u$ (or kernel $k$). Studying this question is a promising avenue for the future research.

---

[1]`kaggle.com/reitanaka/alignedanimefaces`
[2]`https://github.com/iamalexkorotin/NeuralOptimalTransport`

**Reproducibility.** We provide the source code for all experiments and release the checkpoints for all models of §5. The details are given in `README.MD` in the official repository.

`ACKNOWLEDGEMENTS.` The work was supported by the Analytical center under the RF Government (subsidy agreement 000000D730321P5Q0002, Grant No. 70-2021-00145 02.11.2021).

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

## A    TOY EXPERIMENTS IN 1D AND COMPARISON WITH DISCRETE OT

In this section, we learn transport plans between various pairs of toy 1D distributions and compare them with the discrete optimal transport (DOT) considered as the ground truth. We use the distance-induced kernel and consider $\gamma \in \{1, 10\}$. All the rest training details (fully-connected architectures, optimizers, etc.) match those of (Korotin et al., 2023, Appendix C). We consider *Gaussian $\mathcal{N}(0, 1)$ → Mixture of 2 Gaussians* and *Mixture of 3 Gaussians → Mixture of 2 Gaussians*.

In Figure 9, we visualize the pairs $\mathbb{P}, \mathbb{Q}$ (1st and 2nd columns), the plan $\hat{\pi}$ learned by Kernel NOT (3rd column) and the plan $\pi^*$ learned by DOT (4th column). To compute DOT, we sample $10^3$ random points $x \sim \mathbb{P}, y \sim \mathbb{Q}$ and compute a discrete plan by `ot.optim.cg` solver from Python OT (POT) library `https://pythonot.github.io/`. Our learned plan $\hat{\pi}$ and DOT's plan $\pi^*$ nearly match. Note also that, as one may expect, with the increase of $\gamma$ from 1 to 10, the conditional variance of the plan increases and for **very** high $\gamma = 10$ it becomes similar to the trivial plan $\mathbb{P} \times \mathbb{Q}$. This is analogous to the entropic optimal transport, see, e.g., (Peyré et al., 2019, Figure 4.2).

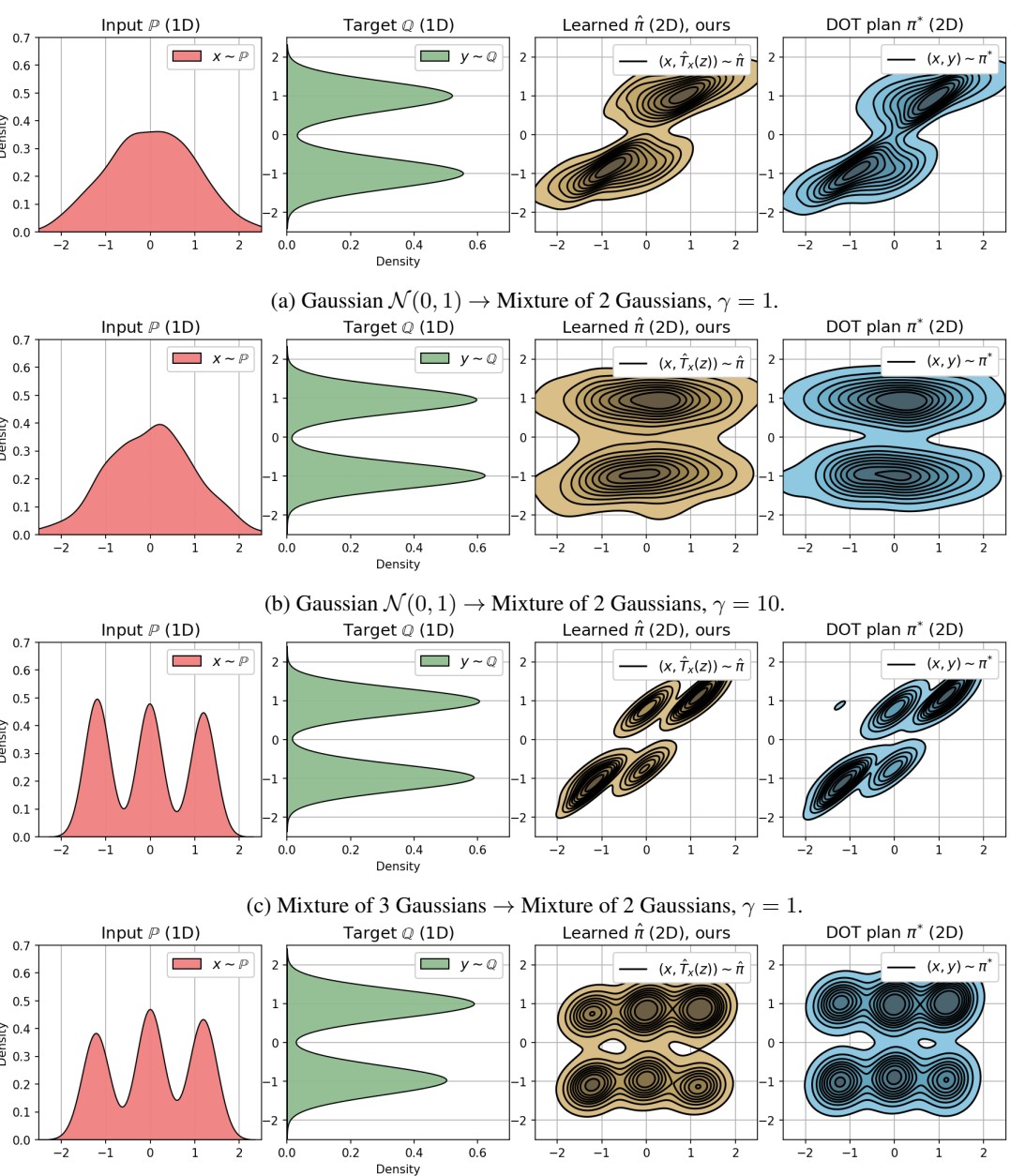

(a) Gaussian $\mathcal{N}(0, 1)$ → Mixture of 2 Gaussians, $\gamma = 1$.

(b) Gaussian $\mathcal{N}(0, 1)$ → Mixture of 2 Gaussians, $\gamma = 10$.

(c) Mixture of 3 Gaussians → Mixture of 2 Gaussians, $\gamma = 1$.

(d) Mixture of 3 Gaussians → Mixture of 2 Gaussians, $\gamma = 10$.

Figure 9: Stochastic plans (3rd and 4th columns) between toy 1D distributions (1st and 2nd columns) learned by our Kernel NOT (3rd column) and discrete OT with the weak kernel cost (4th column).

# B TOY EXPERIMENTS IN 2D

In this section, we learn transport maps between various common pairs of toy 2D distributions. We use the distance-induced kernel and $\gamma = 1$. All the rest training details (fully-connected architectures, optimizers, etc.) exactly match those of (Korotin et al., 2023, Appendix B). We consider *Gaussian* $\mathcal{N}(0, \frac{1}{2}I_2) \to$ *Gaussian* $\mathcal{N}(0, I_2)$ (the same experiment as in Figures 5d and 6), *Gaussian* $\mathcal{N}(0, [\frac{1}{2}]^2 I_2) \to$ *Mixture of 8 Gaussians* and *Gaussian* $\mathcal{N}(0, [\frac{1}{2}]^2 I_2) \to$ *Swiss roll* as $\mathbb{P}, \mathbb{Q}$ pairs. In Figure 10, we provide the learned stochastic (one-to-many) maps. Since the ground truth OT maps for kernel costs are not known, we provide only *qualitative results*.

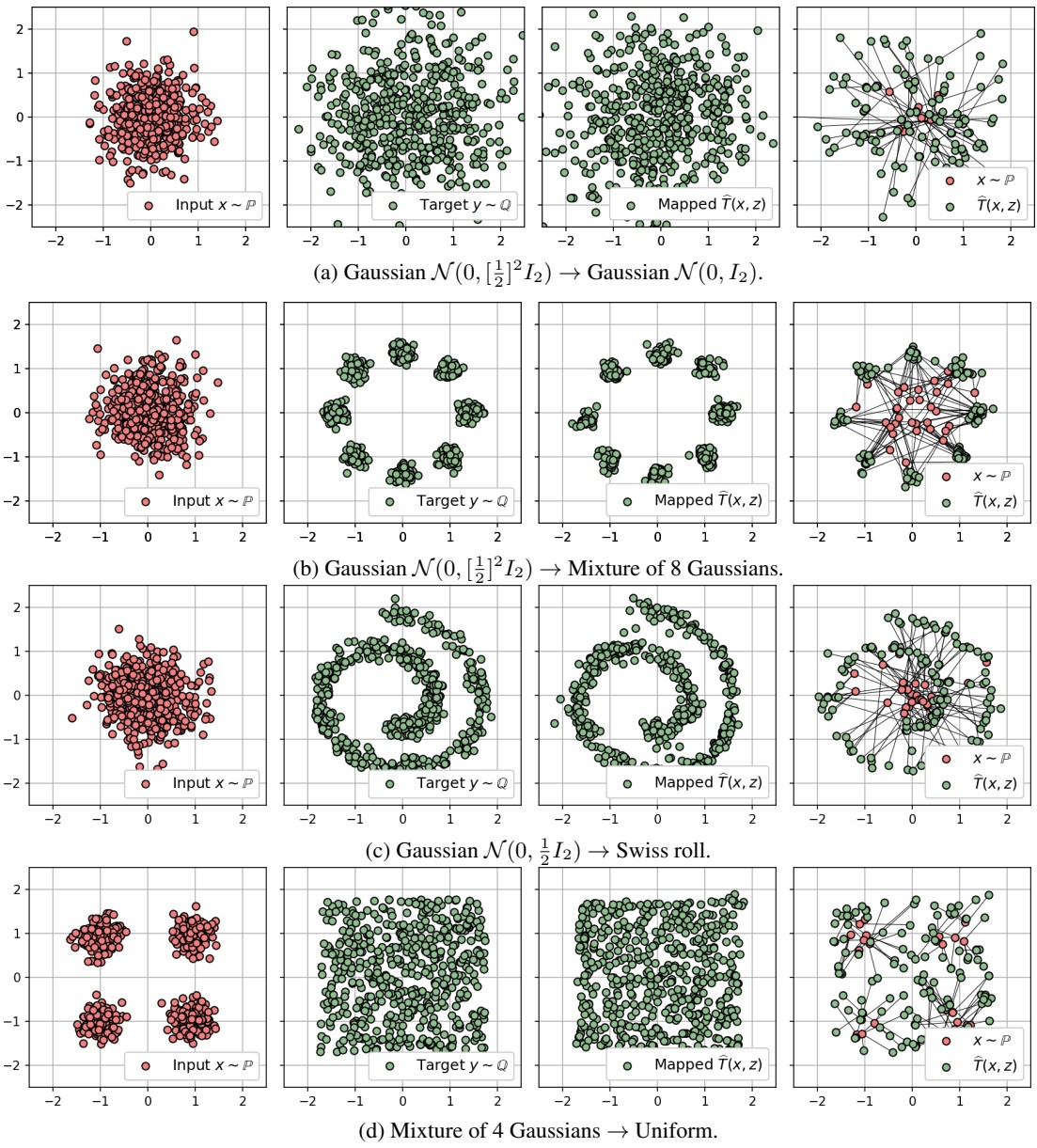

(a) Gaussian $\mathcal{N}(0, [\frac{1}{2}]^2 I_2) \to$ Gaussian $\mathcal{N}(0, I_2)$.

(b) Gaussian $\mathcal{N}(0, [\frac{1}{2}]^2 I_2) \to$ Mixture of 8 Gaussians.

(c) Gaussian $\mathcal{N}(0, \frac{1}{2}I_2) \to$ Swiss roll.

(d) Mixture of 4 Gaussians $\to$ Uniform.

Figure 10: Stochastic (one-to-many) maps learned between toy 2D distributions by Kernel NOT.

## C  ADDITIONAL TOY 2D EXAMPLE

In this section, we provide an additional toy 2D example demonstrating the issue with fake solutions for the weak quadratic cost. We consider a mixture of 4 Gaussians as $\mathbb{P}$ and the uniform distribution on a square as $\mathbb{Q}$. We train NOT with the $\gamma$-weak quadratic cost for $\gamma = 0, \frac{1}{2}, 1$ and report the results in Figure 11. For $\gamma = 0$ (Figure 11b), we see that NOT with the weak quadratic cost[3] learns the target distribution. However, when $\gamma = \frac{1}{2}$ (Figure 11c) and $\gamma = 1$ (Figure 11d), the method does not converge and yields fake solutions. In addition, in Figure 12, we show that for $\gamma = 1$ the method notably fluctuates between the fake solutions. This is analogous to the toy example with Gaussians in §3.1, see Figure 6. For completeness, we run NOT with our proposed kernel cost (16) on this pair $(\mathbb{P}, \mathbb{Q})$ and $\gamma = 1$ and show that it learns the distribution $\mathbb{Q}$, see Figure 10d.

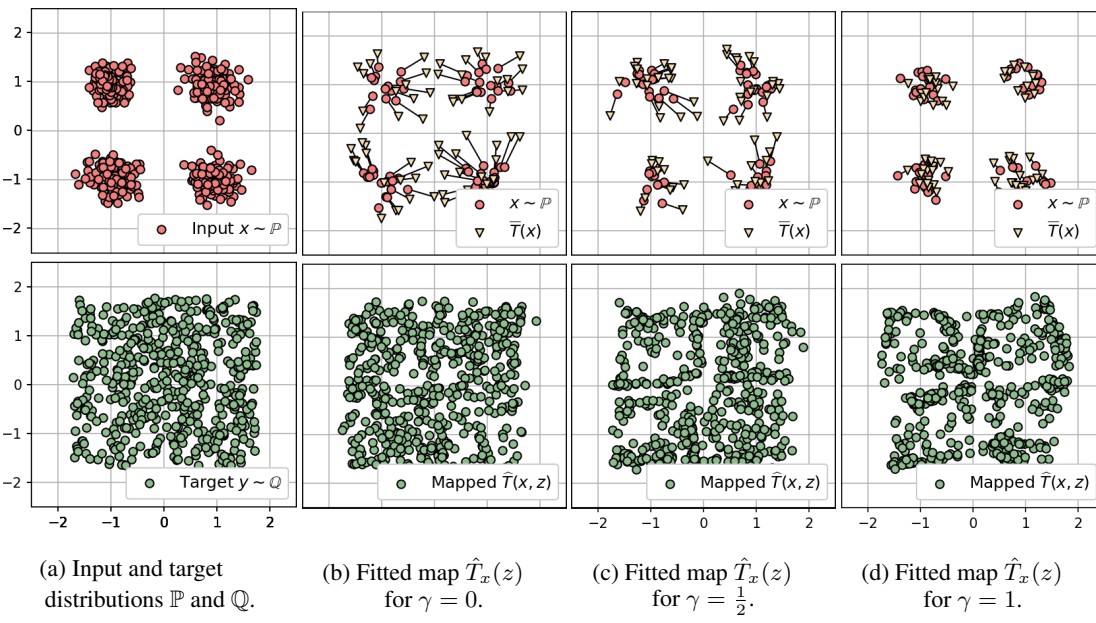

(a) Input and target distributions $\mathbb{P}$ and $\mathbb{Q}$.

(b) Fitted map $\hat{T}_x(z)$ for $\gamma = 0$.

(c) Fitted map $\hat{T}_x(z)$ for $\gamma = \frac{1}{2}$.

(d) Fitted map $\hat{T}_x(z)$ for $\gamma = 1$.

Figure 11: Stochastic maps $\widehat{T}$ between $\mathbb{P}$ and $\mathbb{Q}$ fitted by NOT with costs $C_{2,\gamma}$ for various $\gamma$.

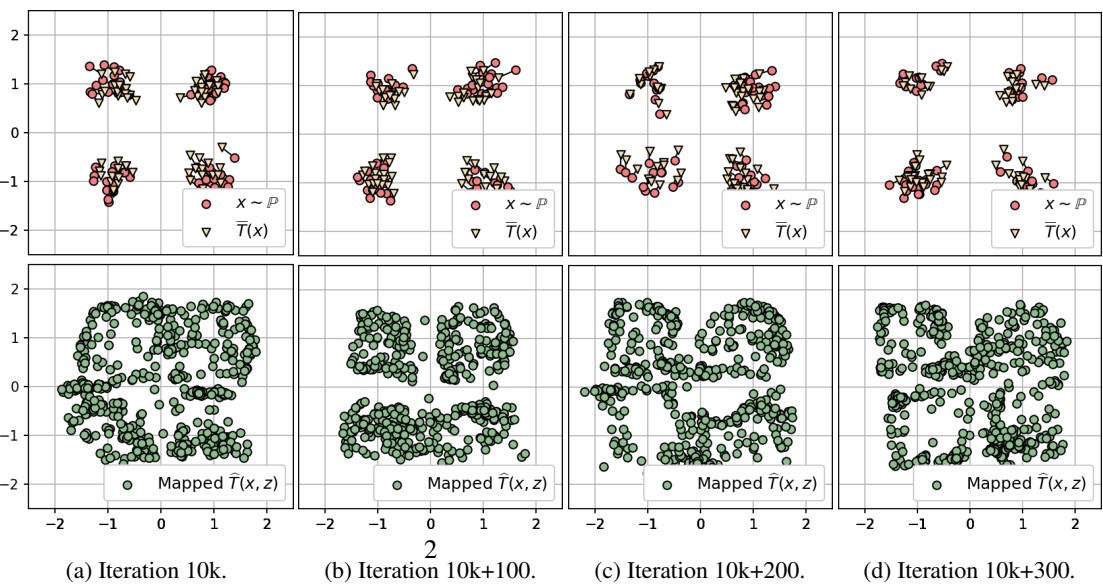

(a) Iteration 10k.

(b) Iteration 10k+100.

(c) Iteration 10k+200.

(d) Iteration 10k+300.

Figure 12: The evolution of the learned transport map $\widehat{T}$ during training on a toy 2D example ($\gamma = 1$).

---

[3]In this case, the cost is the **strong** quadratic cost.

# D  VARIANCE-SIMILARITY TRADE-OFF

In this section, we study how parameter $\gamma$ affects the resulting learned transport map. In (Korotin et al., 2023, Appendix A), the authors empirically show that for the $\gamma$-weak quadratic cost $C_{2,\gamma}$ the variety of samples $T_x(z)$ produced for a fixed $x$ and $z \sim \mathbb{S}$ increases with the increase of $\gamma$, but their similarity to $x$ decreases. We formalize this statement and generalize it for kernel costs $C_{k,\gamma}$.

For a plan $\pi$, we define its (feature) *conditional variance* and (the square of) input-output *distance* by

$$\mathrm{CVar}_u(\pi) \stackrel{def}{=} \int_{\mathcal{X}} \mathrm{Var}\left(u \sharp \pi(y|x)\right) d\pi(x) \quad \text{and} \quad \mathrm{Dist}_u^2(\pi) \stackrel{def}{=} \int_{\mathcal{X} \times \mathcal{Y}} \|u(x) - u(y)\|_{\mathcal{H}}^2 d\pi(x, y), \quad (17)$$

respectively. Recall that $\mathrm{Var}\left(u \sharp \pi(y|x)\right) = \int_{\mathcal{Y} \times \mathcal{Y}} \|u(y) - u(y')\|_{\mathcal{H}}^2 d\pi(y|x) d\pi(y'|x)$. We note that the $\gamma$-weak kernel cost of a plan $\pi \in \Pi(\mathbb{P}, \mathbb{Q})$ is given by

$$\mathrm{Cost}_{k,\gamma}(\pi) = \frac{1}{2} \mathrm{Dist}_u(\pi) - \frac{\gamma}{2} \mathrm{CVar}_u(\pi). \quad (18)$$

Our following proposition explains the behaviour of the above mentioned values for OT plans.

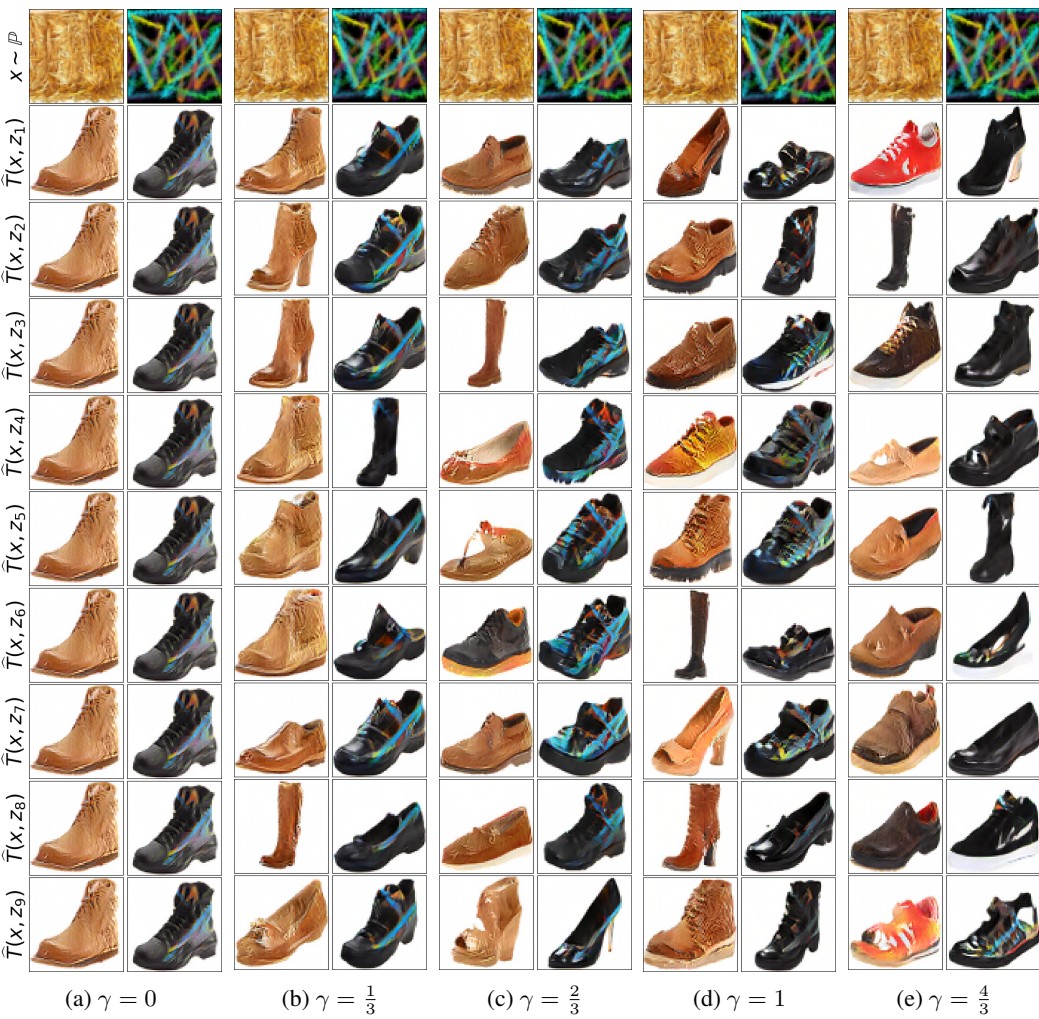

(a) $\gamma = 0$    (b) $\gamma = \frac{1}{3}$    (c) $\gamma = \frac{2}{3}$    (d) $\gamma = 1$    (e) $\gamma = \frac{4}{3}$

Figure 13: *Texture $\rightarrow$ shoes* ($64 \times 64$) translation with the $\gamma$-weak kernel cost for various values $\gamma$.

**Proposition 3** (Behavior of the conditional variance and input-output distance). *Let $\pi_\gamma^* \in \Pi(\mathbb{P}, \mathbb{Q})$ be an OT plan for $\gamma$-weak kernel cost. Then for $\gamma_2 > \gamma_1 \geq 0$ it holds true*

$$\mathrm{CVar}_u(\pi_{\gamma_1}^*) \leq \mathrm{CVar}_u(\pi_{\gamma_2}^*) \quad and \quad \mathrm{Dist}_u(\pi_{\gamma_1}^*) \leq \mathrm{Dist}_u(\pi_{\gamma_2}^*), \quad (19)$$

| | $\gamma = 0$ | $\gamma = \frac{1}{3}$ | $\gamma = \frac{2}{3}$ | $\gamma = 1$ | $\gamma = \frac{4}{3}$ |
|---|---|---|---|---|---|
| $\mathrm{CVar}_u(\widehat{\pi}_\gamma)$ | 0.72 | 15.2 | 16.28 | 17.86 | 18.26 |
| $\mathrm{Dist}_u^2(\widehat{\pi}_\gamma)$ | 46.6 | 48.13 | 48.75 | 49.87 | 51.24 |
| $\mathrm{Cost}_{k,\gamma}(\widehat{\pi}_\gamma)$ | 23.3 | 21.56 | 19.0 | 16.01 | 13.48 |

Table 2: The values of $\mathrm{CVar}(\widehat{\pi}_\gamma)$, $\mathrm{Dist}(\widehat{\pi}_\gamma)$ and $\mathrm{Cost}_{k,\gamma}(\widehat{\pi}_\gamma)$ of learned plans $\widehat{\pi}_\gamma \approx \pi_\gamma^*$ with different values of $\gamma$ on *texture $\to$ shoes* ($64 \times 64$) translation with the kernel quadratic cost.

*i.e., for larger $\gamma$, the OT plan $\pi_\gamma^*$ on average for each $x$ yields more conditionally diverse samples $\pi^*(y|x)$ but they are less close to $x$ in features w.r.t. $\| \cdot \|_{\mathcal{H}}^2$. The OT cost is non-increasing, i.e.,*

$$\mathrm{Cost}_{k,\gamma_1}(\pi_{\gamma_1}^*) \geq \mathrm{Cost}_{k,\gamma_1}(\pi_{\gamma_2}^*). \tag{20}$$

The proof is given in Appendix G. We empirically check the proposition by training OT maps for $C_{k,\gamma}$ for *texture $\to$ shoes* translation ($64 \times 64$), distance-induced $k$ (§5) and $\gamma \in \{0, \frac{1}{3}, \frac{2}{3}, 1, \frac{4}{3}\}$, see Figure 13 and Table 2. We observe the increase of the variety of samples with the increase of $\gamma$. At the same time, with the increase of $\gamma$, output samples become less similar to the inputs.

# E   EXPERIMENTS WITH DIFFERENT KERNELS

We empirically test several popular kernels on *texture $\to$ handbag* translation ($64 \times 64$), $\gamma = \frac{1}{3}$. We consider bilinear, distance-based, Gaussian and Laplacian kernels. The three latter kernels are characteristic. The quantitative and qualitative results are given in Figure 14.

For the kernels in view, the squared feature distance can be expressed as $\|u(x) - u(y)\|_{\mathcal{H}}^2 = h(\|x - y\|)$ for some increasing function $h : \mathbb{R}_+ \to \mathbb{R}_+$. Due to this, all the stochastic transport maps try to preserve the input content in the pixel space. According to FID, the distance-based kernel performs better than the bilinear one, which agrees with the results in §5. Interestingly, both Gaussian and Laplacian kernels are slightly outperformed by the bilinear kernel. We do not know why this happens, but we presume that this might be related to their boundness ($0 < k(x,y) \leq 1$) and $\exp$ operation.

# F   RESTRICTED DUALITY FOR THE WEAK QUADRATIC COST

In this section, we derive duality formula (9) for the $\gamma$-weak quadratic cost. First, we derive formula (9) for $\gamma = 1$ by using (Gozlan & Juillet, 2020, Theorems 1.1, 1.2). Next, following the discussion in (Gozlan & Juillet, 2020, §5.2), we generalize duality formula (9) to arbitrary $\gamma > 0$. We note that the constants in our derivations differ from those in (Gozlan & Juillet, 2020) since our quadratic cost $C_{2,\gamma}$ differs by a constant multiplicative factor.

**Part 1** ($\gamma = 1$). From (Gozlan & Juillet, 2020, Theorems 1.1, 1.2) it follows that there exists a lower-semi-continuous and convex function $v^* : \mathbb{R}^D \to \mathbb{R} \cup \{\infty\}$ which maximizes the following expression:

$$\mathcal{W}_{2,1}^2(\mathbb{P}, \mathbb{Q}) = \max_{v \in \text{l.s.c. Cvx}} \left[ \int_{\mathcal{X}} \inf_{y \in \mathbb{R}^D} \left[ v(y) + \frac{1}{2}\|x - y\|^2 \right] d\mathbb{P}(x) - \int_{\mathcal{Y}} v(y) d\mathbb{Q}(y) \right]. \tag{21}$$

Importantly, $\phi^*(x) \stackrel{def}{=} \overline{(\frac{\|\cdot\|^2}{2} + v^*)}(x)$ is the optimal restricted potential, i.e., $\nabla \phi^*$ implements the projection of $\mathbb{P}$ to $\mathbb{Q}$ and is a 1-smooth convex function.

We consider the change of variables $\phi(x) = \overline{(\frac{\|\cdot\|^2}{2} + v)}(x)$ in (21). Since $\frac{\|y\|^2}{2} + v(y)$ is 1-strongly convex, its conjugate $\phi(x)$ is 1-smooth. From the lower-semi-continuity of $v$ it follows that

$$\overline{\phi}(y) = \overline{\overline{(\frac{\|\cdot\|^2}{2} + v)}}(y) = (\frac{\|\cdot\|^2}{2} + v)(y) = \frac{1}{2}\|y\|^2 + v(y) \implies -v(y) = \frac{1}{2}\|y\|^2 - \overline{\phi}(y). \tag{22}$$

We derive

$$\inf_{y \in \mathbb{R}^D} \left[ v(y) + \frac{1}{2}\|x - y\|^2 \right] = \frac{1}{2}\|x\|^2 + \inf_{y \in \mathbb{R}^D} \left[ \phi(y) - \langle x, y \rangle \right] =$$

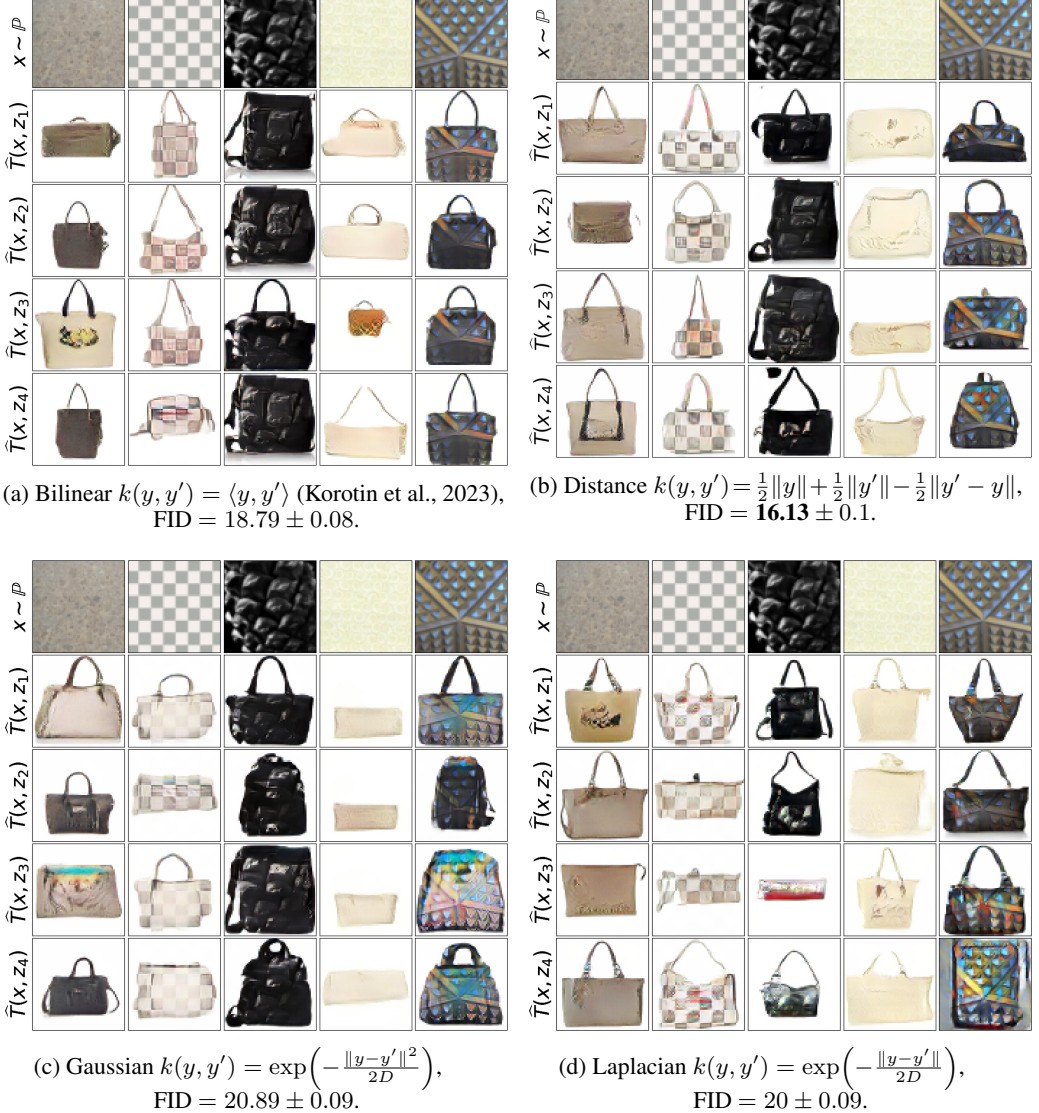

(a) Bilinear $k(y, y') = \langle y, y' \rangle$ (Korotin et al., 2023), FID $= 18.79 \pm 0.08$.

(b) Distance $k(y, y') = \frac{1}{2}\|y\| + \frac{1}{2}\|y'\| - \frac{1}{2}\|y' - y\|$, FID $= \mathbf{16.13} \pm 0.1$.

(c) Gaussian $k(y, y') = \exp\left(-\frac{\|y - y'\|^2}{2D}\right)$, FID $= 20.89 \pm 0.09$.

(d) Laplacian $k(y, y') = \exp\left(-\frac{\|y - y'\|}{2D}\right)$, FID $= 20 \pm 0.09$.

Figure 14: *Texture → handbags* ($64 \times 64$) translation with the different $\gamma$-weak kernel costs.

$$\frac{1}{2}\|x\|^2 - \sup_{y \in \mathbb{R}^D}\left[\langle x, y \rangle - \phi(y)\right] = \frac{1}{2}\|x\|^2 - \overline{\phi}(x). \tag{23}$$

We substitute (22) and (23) to (21) and obtain

$$\mathcal{W}_{2,1}^2(\mathbb{P}, \mathbb{Q}) = \max_{\phi \in \mathrm{Cvx}(1)}\left[\int_{\mathcal{X}}\frac{\|x\|^2}{2}d\mathbb{P}(x) + \int_{\mathcal{Y}}\frac{\|y\|^2}{2}d\mathbb{Q}(x) - \left(\int_{\mathcal{X}}\phi(x)d\mathbb{P}(x) + \int_{\mathcal{Y}}\overline{\phi}(y)d\mathbb{Q}(y)\right)\right]. \tag{24}$$

We only need to note that (24) exactly matches the desired (9) for $\gamma = 1$.

**Part 2** (arbitrary $\gamma > 0$). In (Gozlan & Juillet, 2020, §5.2), the authors show that the OT problem between $\mathbb{P}$ and $\mathbb{Q}$ for the $\gamma$-weak cost becomes the OT problem between $\frac{1}{\gamma}\mathbb{P}$ and $\mathbb{Q}$ for the 1-weak cost. It holds

$$\mathcal{W}_{2,\gamma}^2(\mathbb{P}, \mathbb{Q}) = \Big[\gamma \cdot \underbrace{\inf_{\mathbb{P}' \preceq \mathbb{Q}}\mathbb{W}_2^2(\frac{1}{\gamma}\sharp\mathbb{P}, \mathbb{P}')}_{=\mathcal{W}_{2,1}^2(\frac{1}{\gamma}\sharp\mathbb{P}, \mathbb{Q}), \text{ see (Backhoff-Veraguas et al., 2019, Thm. 1.4)}}\Big] + \frac{1-\gamma}{2\gamma}\int_{\mathcal{X}}\|x\|^2 d\mathbb{P}(x) + \frac{1-\gamma}{2}\int_{\mathcal{Y}}\|y\|^2 d\mathbb{Q}(x). \tag{25}$$

Moreover, $\phi^*(x)$ is the optimal restricted potential for $\gamma$-weak cost between $\mathbb{P}, \mathbb{Q}$ if and only if $\phi_1^*(x) = \gamma\phi^*(\gamma^{-1}x)$ is the optimal restricted potential between $\frac{1}{\gamma}\sharp\mathbb{P}$ and $\mathbb{Q}$ for the 1-weak quadratic

cost. Note that $\nabla\phi^*(x) = \nabla\phi_1^*(\frac{1}{\gamma}x)$, i.e., $\nabla\phi^*(x)$ first scales $x \sim \mathbb{P}$ by $\frac{1}{\gamma}$ and then implements projection $\nabla\phi_1^*$ of $\frac{1}{\gamma}\sharp\mathbb{P}$ to $\mathbb{Q}$. In particular, the function $\phi^*$ is $\frac{1}{\gamma}$-Lipschitz.

We reparameterize the duality formula (24) for distributions $\frac{1}{\gamma}\sharp\mathbb{P}$ and $\mathbb{Q}$ as follows:

$$\mathcal{W}_{2,1}^2(\frac{1}{\gamma}\sharp\mathbb{P}, \mathbb{Q}) = \max_{\phi_1 \in \text{Cvx}(1)} \left[ \int_{\mathcal{X}} \frac{\|x\|^2}{2} d(\frac{1}{\gamma}\sharp\mathbb{P})(x) + \int_{\mathcal{Y}} \frac{\|y\|^2}{2} d\mathbb{Q}(x) - \right.$$
$$\left( \int_{\mathcal{X}} \phi_1(x) d(\frac{1}{\gamma}\sharp\mathbb{P})(x) + \int_{\mathcal{Y}} \overline{\phi_1}(y) d\mathbb{Q}(y) \right) \right] =$$
$$\max_{\phi_1 \in \text{Cvx}(1)} \left[ \frac{1}{\gamma^2} \int_{\mathcal{X}} \frac{\|x\|^2}{2} d\mathbb{P}(x) + \int_{\mathcal{Y}} \frac{\|y\|^2}{2} d\mathbb{Q}(x) - \left( \int_{\mathcal{X}} \phi_1(\frac{1}{\gamma}x) d\mathbb{P}(x) + \int_{\mathcal{Y}} \overline{\phi_1}(y) d\mathbb{Q}(y) \right) \right] = \quad (26)$$
$$\max_{\phi \in \text{Cvx}(\frac{1}{\gamma})} \left[ \frac{1}{\gamma^2} \int_{\mathcal{X}} \frac{\|x\|^2}{2} d\mathbb{P}(x) + \int_{\mathcal{Y}} \frac{\|y\|^2}{2} d\mathbb{Q}(x) - \frac{1}{\gamma} \left( \int_{\mathcal{X}} \phi(x) d\mathbb{P}(x) + \int_{\mathcal{Y}} \overline{\phi}(y) d\mathbb{Q}(y) \right) \right]. \quad (27)$$

In transition from (26) to (27), we use the change of variables for $\phi(x) = \gamma\phi_1(\frac{1}{\gamma}x)$ known as the *right scalar multiplication*. It yields $\overline{\phi}(y) = \gamma\overline{\phi_1}(y)$.

To finish the derivation of dual form (9), we simply substitute (27) to (25).

## G PROOFS

*Proof of Lemma 1.* Assume the opposite, i.e., $T^{\dagger}\sharp(\mathbb{P}\times\mathbb{S}) = \mathbb{Q}$. Then $T^{\dagger}$ implicitly represents some transport plan $\pi^{\dagger} \in \Pi(\mathbb{P}, \mathbb{Q})$ between $\mathbb{P}$ and $\mathbb{Q}$. By the definition of $f^*$, $T^{\dagger}$ and (6), it holds

$$\text{Cost}(\mathbb{P}, \mathbb{Q}) = \mathcal{L}(f^*, T^{\dagger}) =$$
$$\int_{\mathcal{X}} C(x, T_x^{\dagger}\sharp\mathbb{S}) d\mathbb{P}(x) - \int_{\mathcal{X}} \int_{\mathcal{Z}} f^*(T_x^{\dagger}(z)) d\mathbb{S}(z) d\mathbb{P}(x) + \int_{\mathcal{Y}} f^*(y) d\mathbb{Q}(y) = \quad (28)$$
$$\int_{\mathcal{X}} C(x, T_x^{\dagger}\sharp\mathbb{S}) d\mathbb{P}(x) - \int_{\mathcal{Y}} f^*(y) d\mathbb{Q}(y) + \int_{\mathcal{Y}} f^*(y) d\mathbb{Q}(y) = \quad (29)$$
$$\int_{\mathcal{X}} C(x, T_x^{\dagger}\sharp\mathbb{S}) d\mathbb{P}(x) = \int_{\mathcal{X}} C(x, \pi^{\dagger}(\cdot|x)) d\pi^{\dagger}(x), \quad (30)$$

where in transition between lines (28) and (29), we use the change of variables formula for $y = T_x^{\dagger}(z)$ and the equality $T^{\dagger}\sharp(\mathbb{P}\times\mathbb{S}) = \mathbb{Q}$. From (30) we see that the cost of the plan $\pi^{\dagger} \in \Pi(\mathbb{P}, \mathbb{Q})$ equals the optimal cost $\text{Cost}(\mathbb{P}, \mathbb{Q})$. As a result, it is an optimal plan by definition. In turn, $T^{\dagger}$ is an stochastic OT map but not a fake solution. This is a contradiction. Thus, it holds that $T^{\dagger}\sharp(\mathbb{P}\times\mathbb{S}) \neq \mathbb{Q}$. $\square$

*Proof of Lemma 2.* We compute the $C$-transform of $f^*(y) = \frac{\|y\|^2}{2} - \overline{\phi^*}(x)$. We have

$$(f^*)^C(x) = \inf_{\mu \in \mathcal{P}(\mathcal{Y})} \left[ C_{2,\gamma}(x, \mu) - \int_{\mathcal{Y}} f^*(y) d\mu(y) \right] =$$
$$\inf_{\mu \in \mathcal{P}(\mathcal{Y})} \left[ \frac{1}{2} \int_{\mathcal{Y}} \|x - y\|^2 d\mu(y) - \frac{\gamma}{2} \text{Var}(\mu) - \int_{\mathcal{Y}} \left[ \frac{\|y\|^2}{2} - \overline{\phi^*}(y) \right] d\mu(y) \right] =$$
$$\inf_{\mu \in \mathcal{P}(\mathcal{Y})} \left[ \frac{1}{2}\|x\|^2 - \langle x, \int_{\mathcal{Y}} y \, d\mu(y) \rangle + \frac{1}{2} \int_{\mathcal{Y}} \|y\|^2 d\mu(y) - \frac{\gamma}{2} \text{Var}(\mu) - \int_{\mathcal{Y}} \left[ \frac{\|y\|^2}{2} - \overline{\phi^*}(y) \right] d\mu(y) \right] =$$
$$\inf_{\mu \in \mathcal{P}(\mathcal{Y})} \left[ \frac{1}{2}\|x\|^2 - \langle x, \int_{\mathcal{Y}} y \, d\mu(y) \rangle - \frac{\gamma}{2} \text{Var}(\mu) + \int_{\mathcal{Y}} \overline{\phi^*}(y) d\mu(y) \right] =$$
$$\frac{1}{2}\|x\|^2 + \inf_{\mu \in \mathcal{P}(\mathcal{Y})} \left[ \int_{\mathcal{Y}} (\overline{\phi^*}(y) - \langle x, y \rangle) d\mu(y) - \frac{\gamma}{2} \text{Var}(\mu) \right] =$$
$$\frac{1}{2}\|x\|^2 + \inf_{\mu \in \mathcal{P}(\mathcal{Y})} \left[ \int_{\mathcal{Y}} (\overline{\phi^*}(y) - \frac{\gamma}{2}\|y\|^2 - \langle x, y \rangle) d\mu(y) + \frac{\gamma}{2} \left\| \int_{\mathcal{Y}} y \, d\mu(y) \right\|^2 \right]. \quad (31)$$

Pick any $\mu \in \mathcal{P}(\mathcal{Y})$ and denote its expectation by $m = \int_{\mathcal{Y}} y \, d\mu(y)$. Since $\phi^*$ is $\frac{1}{\gamma}$-smooth, it holds that $\overline{\phi^*}$ is $\gamma$-strongly convex, i.e., $\overline{\phi^*}(y) - \frac{\gamma}{2}\|y\|^2 - \langle x, y\rangle$ is convex. We use the Jensen's inequality:

$$\int_{\mathcal{Y}} \left(\overline{\phi^*}(y) - \frac{\gamma}{2}\|y\|^2 - \langle x, y\rangle\right) d\mu(y) \geq \overline{\phi^*}(m) - \frac{\gamma}{2}\|m\|^2 - \langle x, m\rangle. \tag{32}$$

Therefore, we may restrict the feasible set of inf in (31) to Dirac distributions $\delta_m$, $m \in \mathcal{Y}$. That is,

$$(f^*)^C(x) = \frac{1}{2}\|x\|^2 + \inf_{m \in \mathcal{Y}} \left[\overline{\phi^*}(m) - \frac{\gamma}{2}\|m\|^2 - \langle x, m\rangle + \frac{\gamma}{2}\|m\|^2\right] =$$

$$\frac{1}{2}\|x\|^2 + \inf_{m \in \mathcal{Y}} \left[\overline{\phi^*}(m) - \langle x, m\rangle\right] = \frac{1}{2}\|x\|^2 - \underbrace{\sup_{m \in \mathcal{Y}} \left[\langle x, m\rangle - \overline{\phi^*}(m)\right]}_{=\overline{\overline{\phi^*}}(x)} = \frac{1}{2}\|x\|^2 - \phi^*(x), \tag{33}$$

where $\overline{\overline{\phi^*}}(x) = \phi^*(x)$ holds since $\phi^*$ is continuous. We substitute (33) to (5) and obtain

$$\int_{\mathcal{X}} (f^*)^C(x) d\mathbb{P}(x) + \int_{\mathcal{Y}} f^*(y) d\mathbb{Q}(y) =$$

$$\int_{\mathcal{X}} \left[\frac{\|x\|^2}{2} - \phi^*(x)\right] d\mathbb{P}(x) + \int_{\mathcal{Y}} \left[\frac{\|y\|^2}{2} - \overline{\phi^*}(y)\right] d\mathbb{Q}(y) = \mathcal{W}_{2,\gamma}^2(\mathbb{P}, \mathbb{Q}), \tag{34}$$

where in line (34), we use the optimality of $\phi^*$ and (9). We conclude that $f^*$ maximizes (5), (6). $\quad\square$

*Proof of Proposition 1.* For all $y', y'' \in U_\psi(y)$, $\alpha \in [0,1]$, from the convexity of $\psi$ it follows that

$$\psi(\alpha y' + (1-\alpha)y'') \leq \alpha\psi(y') + (1-\alpha)\psi(y''). \tag{35}$$

By the definition of the subgradient of a convex function it also holds

$$\psi(\alpha y' + (1-\alpha)y'') \geq \psi(y) + \langle x, \alpha y' + (1-\alpha)y'' - y\rangle =$$

$$\alpha\left(\psi(y) + \langle x, y' - y\rangle\right) + (1-\alpha) \cdot \left(\psi(y) + \langle x, y'' - y\rangle\right) = \alpha\psi(y') + (1-\alpha)\psi(y'') \tag{36}$$

for all $x \in \partial_y \psi(y)$. Therefore, (35) and (36) are equalities, and $\alpha y' + (1-\alpha)y'' \in U_\psi(y)$. $\quad\square$

*Proof of Theorem 1.* It holds that

$$T^\dagger \in \arg\inf_T \mathcal{L}(f^*, T) \Leftrightarrow T^\dagger \in \arg\inf_T \int_{\mathcal{X}} \left(C\left(x, T_x \sharp \mathbb{S}\right) - \int_{\mathcal{Z}} f^*\left(T_x(z)\right) d\mathbb{S}(z)\right) d\mathbb{P}(x).$$

The latter condition holds if and only if $\mathbb{P}$-almost surely for all $x \in \mathcal{X}$ we have

$$T_x^\dagger \in \arg\inf_{T_x : \mathcal{Z} \to \mathcal{Y}} \left[C\left(x, T_x^\dagger \sharp \mathbb{S}\right) - \int_{\mathcal{Z}} f^*\left(T_x^\dagger(z)\right) d\mathbb{S}(z)\right]. \tag{37}$$

We substitute $f^* = \frac{\|\cdot\|^2}{2} - \overline{\phi^*}$ and $C = C_{2,\gamma}$. As a result, we derive

$$C_{2,\gamma}\left(x, T_x^\dagger \sharp \mathbb{S}\right) - \int f^*\left(T_x^\dagger(z)\right) d\mathbb{S}(z) =$$

$$\frac{1}{2}\int_{\mathcal{Z}} \|x - T_x^\dagger(z)\|^2 d\mathbb{S}(z) - \frac{\gamma}{2}\mathrm{Var}(T_x^\dagger \sharp \mathbb{S}) - \frac{1}{2}\int_{\mathcal{Z}} \|T_x^\dagger(z)\|^2 d\mathbb{S}(z) + \int_{\mathcal{Z}} \overline{\phi^*}\left(T_x^\dagger(z)\right) d\mathbb{S}(z) =$$

$$\frac{1}{2}\|x\|^2 - \langle x, \int_{\mathcal{Z}} T_x^\dagger(z) d\mathbb{S}(z)\rangle - \frac{\gamma}{2}\mathrm{Var}(T_x^\dagger \sharp \mathbb{S}) + \int_{\mathcal{Z}} \overline{\phi^*}\left(T_x^\dagger(z)\right) d\mathbb{S}(z) =$$

$$\frac{1}{2}\|x\|^2 - \langle x, \int_{\mathcal{Z}} T_x^\dagger(z) d\mathbb{S}(z)\rangle + \frac{\gamma}{2}\left\|\int_{\mathcal{Z}} T_x^\dagger(z) d\mathbb{S}(z)\right\|^2 + \int_{\mathcal{Z}} \left[\overline{\phi^*}\left(T_x^\dagger(z)\right) - \frac{\gamma}{2}\|T_x^\dagger(z)\|^2\right] d\mathbb{S}(z) =$$

$$\frac{1}{2}\|x\|^2 - \langle x, \int_{\mathcal{Z}} T_x^\dagger(z) d\mathbb{S}(z)\rangle + \frac{\gamma}{2}\left\|\int_{\mathcal{Z}} T_x^\dagger(z) d\mathbb{S}(z)\right\|^2 + \int_{\mathcal{Z}} \psi\left(T_x^\dagger(z)\right) d\mathbb{S}(z) \geq \tag{38}$$

$$\frac{1}{2}\|x\|^2 - \langle x, \int_{\mathcal{Z}} T_x^\dagger(z) d\mathbb{S}(z)\rangle + \frac{\gamma}{2}\left\|\int_{\mathcal{Z}} T_x^\dagger(z) d\mathbb{S}(z)\right\|^2 + \psi\left(\int_{\mathcal{Z}} T_x^\dagger(z) d\mathbb{S}(z)\right), \tag{39}$$

where we substitute $\psi = \overline{\phi^*} - \frac{\gamma}{2}\|\cdot\|^2$. Recall that $\psi$ is convex since $\phi^*$ is $\frac{1}{\gamma}$-smooth (Kakade et al., 2009). In transition from (38) to (39), we use the convexity of $\psi$ and the Jensen's inequality.

**Part 1.** To begin with, we prove implication "$\Longrightarrow$" in (11).

If (37) holds true, then (39) is the equality. If it is not, one may pick $T_x'(z) \overset{def}{=} \int_{\mathcal{Z}} T_x^\dagger(z')d\mathbb{S}(z')$ which provides smaller value (39) for (37) than $T_x^\dagger$. Let $T^*$ be any stochastic OT map and $\pi^*$ be its respective OT plan. We know that $T^* \in \arg\inf_T \mathcal{L}(f^*, T)$, i.e., the optimal value (39) of (37) equals

$$\frac{1}{2}\|x\|^2 - \langle x, \int_{\mathcal{Z}} T_x^*(z)d\mathbb{S}(z)\rangle + \frac{\gamma}{2}\left\|\int_{\mathcal{Z}} T_x^*(z)d\mathbb{S}(z)\right\|^2 + \psi\left(\int_{\mathcal{Z}} T_x^*(z)d\mathbb{S}(z)\right) = \qquad (40)$$

$$\frac{1}{2}\|x\|^2 - \langle x, \nabla\phi^*(x)\rangle + \frac{\gamma}{2}\|\nabla\phi^*(x)\|^2 + \psi(\nabla\phi^*(x)), \qquad (41)$$

where we use the equalities $\nabla\phi^*(x) = \int_{\mathcal{Y}} y\,d\pi^*(y|x) = \int_{\mathcal{Z}} T_x^*(z)d\mathbb{S}(z)$, see §2. The function $m \mapsto \frac{1}{2}\|x\|^2 - \langle x, m\rangle + \frac{\gamma}{2}\|m\|^2 + \psi(m)$ is $\gamma$-strongly convex in $m$, therefore, the minimizer $m^* = \nabla\phi^*(x)$ is unique. This yields that $\int_{\mathcal{Z}} T_x^\dagger(z)d\mathbb{S}(z) = m^* = \nabla\phi^*(x)$.

We know that (38) equals (39), i.e.,

$$\int_{\mathcal{Z}} \psi(T_x^\dagger(z))d\mathbb{S}(z) - \psi\left(\int_{\mathcal{Z}} T_x^\dagger(z)d\mathbb{S}(z)\right) = \int_{\mathcal{Z}} \psi(T_x^\dagger(z))d\mathbb{S}(z) - \psi(\nabla\phi^*(x)) = 0, \qquad (42)$$

and the Jensen's gap vanishes. Now we are going to prove that $\mathrm{Supp}(T_x\sharp\mathbb{S}) \subset U_\psi(\nabla\phi^*(x))$. We need to show that for every $x' \in \partial_y(\nabla\phi^*(x))$ the following inequality

$$\psi(y) \geq \psi(\nabla\phi^*(x)) + \langle x', y - \nabla\phi^*(x)\rangle \qquad (43)$$

is the equality $T_x^\dagger\sharp\mathbb{S}$-almost surely for all $y \in \mathrm{Supp}(T_x^\dagger\sharp\mathbb{S})$. Assume the opposite, i.e., for some $x'$, (43) holds true not $T_x^\dagger\sharp\mathbb{S}$-almost surely. We integrate (43) w.r.t. $y \sim T_x^\dagger\sharp\mathbb{S}$ and get the strict inequality

$$\int_{\mathcal{Y}} \psi(y)d(T_x^\dagger\sharp\mathbb{S})(y) > \psi(\nabla\phi^*(x)) + \langle x', \int_{\mathcal{Y}} y\,d(T_x^\dagger\sharp\mathbb{S})(y) - \nabla\phi^*(x)\rangle.$$

We use the change of variables for $y = T_x^\dagger(z)$ and obtain

$$\int_{\mathcal{Z}} \psi(T_x^\dagger(z))d\mathbb{S}(z) > \psi(\nabla\phi^*(x)) + \langle y', \underbrace{\int_{\mathcal{Z}} T_x^\dagger(z)d\mathbb{S}(z) - \nabla\phi^*(x)}_{=0}\rangle = \psi(\nabla\phi^*(x)),$$

which contradicts (42). This finishes the proof of implication "$\Longrightarrow$".

**Part 2.** Now we prove implication "$\Longleftarrow$" in (11).

For every $T^\dagger$ satisfying the conditions on the right-hand side of (11), the Jensen's gap (42) is zero since $\psi$ is linear in $U_\psi(\nabla\phi^*(x))$. Therefore, (38) equals (39). Due to $\int_{\mathcal{Z}} T_x^\dagger(z)d\mathbb{S}(z) = \nabla\phi^*(x)$, we have that (39) attains the optimal (minimal) values. Consequently, (37) holds.

Finally, we note that convexity of $\arg\inf_T \mathcal{L}(f^*, T)$ follows from the convexity of sets $U_\psi(\cdot)$. $\qquad \square$

*Proof of Lemma 3.* The plan $\pi \in \Pi(\mathbb{P}, \mathbb{Q})$ is optimal if and only if $\int_{\mathcal{Y}} y\,d\pi(y|x) = \nabla\phi^*(x)$. That is, a stochastic map $T : \mathcal{X} \times \mathcal{Z} \to \mathcal{Y}$ is optimal if and only if it pushes $\mathbb{P}$ to $\mathbb{Q}$ (represents some plan $\pi \in \Pi(\mathbb{P}, \mathbb{Q})$), i.e., $T\sharp(\mathbb{P}\times\mathbb{S}) = \mathbb{Q}$, and $\int_{\mathcal{Z}} T_x(z)d\mathbb{S} = \nabla\phi^*(x)$. The optimal barycentric projection $\overline{T^*}$ satisfies the second condition by the definition.

Consider implication "$\overset{(a)}{\Longrightarrow}$". Since $\overline{T^*}$ is a stochastic OT map, the first condition $\overline{T^*}\sharp(\mathbb{P}\times\mathbb{S}) = \mathbb{Q}$ holds true. Recall that by the definition of $\phi^*$ and $\overline{T^*}$ we have $\overline{T^*}\sharp(\mathbb{P}\times\mathbb{S}) = \nabla\phi^*\sharp\mathbb{P} = (\frac{1}{\gamma}\sharp\mathbb{P})$.

Thus, $\mathbb{Q} = \mathrm{Proj}_{\preceq\mathbb{Q}}(\frac{1}{\gamma}\sharp\mathbb{P})$. Consider implication "$\overset{(a)}{\Longleftarrow}$". Since $\mathrm{Proj}_{\preceq\mathbb{Q}}(\frac{1}{\gamma}\sharp\mathbb{P}) = \mathbb{Q}$, the first condition $\overline{T^*}\sharp(\mathbb{P}\times\mathbb{S}) = \mathbb{Q}$ holds true. Therefore, $\overline{T^*}$ is a stochastic OT map.

Now we prove implication "$\overset{(b)}{\Longrightarrow}$". Let $T^*$ be any stochastic OT map. We compute the second moment of $T_x^*\sharp(\mathbb{P}\times\mathbb{S}) = \mathbb{Q}$ below:

$$\int_{\mathcal{Y}} \|y\|^2 d\mathbb{Q}(y) = \int_{\mathcal{X}}\int_{\mathcal{Z}} \|T_x^*(z)\|^2 d\mathbb{S}(z)d\mathbb{P}(x) = \qquad (44)$$

$$\int_{\mathcal{X}} \left( \text{Var}(T_x^* \sharp \mathbb{S}) + \left\| \int_{\mathcal{Z}} T_x^*(z) d\mathbb{S}(z) \right\|^2 \right) d\mathbb{P}(x) =$$

$$\int_{\mathcal{X}} \text{Var}(T_x^* \sharp \mathbb{S}) d\mathbb{P}(x) + \int_{\mathcal{X}} \|\nabla \phi^*(x)\|^2 d\mathbb{P}(x) = \tag{45}$$

$$\int_{\mathcal{X}} \text{Var}(T_x^* \sharp \mathbb{S}) d\mathbb{P}(x) + \int_{\mathcal{Y}} \|y\|^2 d\mathbb{Q}(y), \tag{46}$$

where in transition to (45), we use the equality $\int_{\mathcal{Z}} T_x^*(z) d\mathbb{S}(z) = \nabla \phi^*(x)$; in transition to (46), we use the change of variables formula for $y = \nabla \phi^*(x)$ and $\nabla \phi^*(x) \sharp \mathbb{P} = \mathbb{Q}$. Finally, by comparing (44) and (46), we obtain that $\text{Var}(T_x \sharp \mathbb{S}) = 0$ holds $\mathbb{P}$-almost surely. That is, $T^*$ is deterministic (does not depend on $z$) and $(\mathbb{P} \times \mathbb{S})$-almost surely matches the optimal barycentric projection $\overline{T^*}$. $\qquad \square$

*Proof of Proposition 2.* We are going to show that the second moment of $T^\alpha \sharp (\mathbb{P} \times \mathbb{S})$ is less than that of $\mathbb{Q}$. Consequently, $T^\alpha \sharp (\mathbb{P} \times \mathbb{S}) \neq \mathbb{Q}$, and $T^\alpha$ can not be a stochastic OT map. First, for $T^\dagger$ we have

$$\int_{\mathcal{X}} \int_{\mathcal{Z}} \|T_x^\dagger(z)\|^2 d\mathbb{S}(z) d\mathbb{P}(x) = \underbrace{\text{Var} \left( T^\dagger \sharp (\mathbb{P} \times \mathbb{S}) \right)}_{\leq \text{Var}(\mathbb{Q})} - \left\| \int_{\mathcal{X}} \int_{\mathcal{Z}} T_x^\dagger(z) d\mathbb{S}(z) d\mathbb{P}(x) \right\|^2 \leq$$

$$\text{Var}(\mathbb{Q}) - \left\| \int_{\mathcal{X}} \nabla \phi^*(x) d\mathbb{P}(x) \right\|^2 = \text{Var}(\mathbb{Q}) - \|m_{\mathbb{Q}}\|^2 = \int_{\mathcal{X}} \int_{\mathcal{Z}} \|T_x^*(z)\|^2 d\mathbb{S}(z) d\mathbb{P}(x), \tag{47}$$

where in the last equality we use $T^* \sharp (\mathbb{P} \times \mathbb{S}) = \mathbb{Q}$. Finally, we derive

$$\int_{\mathcal{X}} \int_{\mathcal{Z}} \|T_x^\alpha(z)\|^2 d\mathbb{S}(z) d\mathbb{P}(x) = \int_{\mathcal{X}} \int_{\mathcal{Z}} \|\alpha T_x^*(z) + (1-\alpha) T_x^\dagger(z)\|^2 d\mathbb{S}(z) d\mathbb{P}(x) <$$

$$\int_{\mathcal{X}} \int_{\mathcal{Z}} \left[ \alpha \|T_x^*(z)\|^2 + (1-\alpha) \|T_x^\dagger(z)\|^2 \right] d\mathbb{S}(z) d\mathbb{P}(x) = \tag{48}$$

$$\alpha \int_{\mathcal{X}} \int_{\mathcal{Z}} \|T_x^*(z)\|^2 d\mathbb{S}(z) d\mathbb{P}(x) + (1-\alpha) \int_{\mathcal{X}} \int_{\mathcal{Z}} \|T_x^\dagger(z)\|^2 d\mathbb{S}(z) d\mathbb{P}(x) \leq$$

$$\int_{\mathcal{X}} \int_{\mathcal{Z}} \|T_x^*(z)\|^2 d\mathbb{S}(z) d\mathbb{P}(x) = [\text{second moment of } \mathbb{Q}]. \tag{49}$$

In transition to (48), we use the Jensen's inequality for $\| \cdot \|^2$. The inequality is strict since $\| \cdot \|^2$ is strictly convex and $T^\dagger \neq T^*$ ($\mathbb{P} \times \mathbb{S}$-almost surely). In transition to (49), we use (47). That is, $T^\alpha \sharp (\mathbb{P} \times \mathbb{S}) \neq \mathbb{Q}$ as its second moment is smaller. $\qquad \square$

**Remark**. The assumption $\text{Proj}_{\preceq \mathbb{Q}}(\frac{1}{\gamma} \sharp \mathbb{P}) \neq \mathbb{Q}$ in Proposition 2 is needed to guarantee the existence of a function $T^\dagger \in \arg \inf_T \mathcal{L}(f^*, T)$ which differs from the given stochastic OT map $T^*$. Due to our Lemma 3, the optimal barycenteric projection $\overline{T^*} \in \arg \inf_T \mathcal{L}(f^*, T)$ is not an OT map. Thus, $T^\dagger = \overline{T^*}$ is a suitable example. Since $\overline{T^*} \sharp (\mathbb{P} \times \mathbb{S}) \preceq \mathbb{Q}$, we also have $\text{Var} \left( \overline{T^*} \sharp (\mathbb{P} \times \mathbb{S}) \right) \leq \text{Var}(\mathbb{Q})$.

*Proof of Lemma 4.* The lower-semi-continuity of $C_{k,\gamma}(x, \mu) = C_{u,\gamma}(x, \mu)$ in $(x, \mu)$ follows from the continuity of $k$, compactness of $\mathcal{X} = \mathcal{Y} \subset \mathbb{R}^D$ and (Santambrogio, 2015, Lemma 7.3). [4]

The term $\frac{1}{2} \int_{\mathcal{Y}} \|u(x) - u(y)\|_{\mathcal{H}}^2 d\mu(y)$ in (12) is linear in $\mu$ and, consequently, convex. The second term in (12) equals to $-\frac{\gamma}{2} \text{Var}(u \sharp \mu)$. The pushforward operator $u \sharp$ is linear and $\text{Var}(\cdot)$ is concave. Therefore, the second term is convex in $\mu$ ($\gamma \geq 0$). As a result, $C_{k,\gamma}(x, \mu)$ is convex in $\mu$.

To prove that $C_{k,\gamma}$ is lower-bounded, we rewrite (12) analogously to (3), i.e.,

$$C_{k,\gamma}(x, \mu) = \frac{1}{2} \|u(x) - \int_{\mathcal{Y}} u(y) d\mu(y)\|_{\mathcal{H}}^2 + \frac{1-\gamma}{2} \cdot \left[ \frac{1}{2} \int_{\mathcal{Y} \times \mathcal{Y}} \|u(y) - u(y')\|_{\mathcal{H}}^2 d\mu(y) d\mu(y') \right] =$$

---

[4] We use the lower semi-continuity (in $\mu$) of $C(x, \mu)$ w.r.t. the weak convergence of distributions in $\mathcal{P}(\mathcal{Y})$. In contrast, Backhoff-Veraguas et al. (2019) work with $\mu \in \mathcal{P}_p(\mathcal{Y}) \subset \mathcal{P}(\mathcal{Y})$, i.e., with the distributions which have a finite $p$-th moment. They prove the existence and duality results for weak OT (2) assuming that $C(x, \mu)$ is lower semi-continuous w.r.t. the convergence in the Wasserstein-$p$ sence in $\mathcal{P}_p(\mathcal{Y})$. Since we consider compact $\mathcal{Y}$, it holds that $\mathcal{P}_p(\mathcal{Y}) = \mathcal{P}(\mathcal{Y})$ and these notions of convergence coincide (Villani, 2008, Def. 6.8).

$$\frac{1}{2}\|u(x) - \int_{\mathcal{Y}} u(y)d\mu(y)\|_{\mathcal{H}}^2 + \frac{1-\gamma}{2} \cdot \mathrm{Var}(u\sharp\mu). \quad (50)$$

Both terms are non-negative ($\gamma \in [0,1]$), i.e., $C_{k,\gamma}(x,\mu)$ is lower bounded by 0. $\qquad\square$

*Proof of Lemma 5.* To begin with, we prove that if $k$ is characteristic, then $C_{k,\gamma}(x,\mu) = C_{u,\gamma}(x,\mu)$ is *strictly* convex in $\mu$. The term $\int_{\mathcal{X}} \|u(x) - u(y)\|_{\mathcal{H}}^2 d\mu(y)$ in (12) is linear in $\mu$, so we focus on the second (variance) term $-\frac{\gamma}{2}\mathrm{Var}(u\sharp\mu)$. We prove that $\mathrm{Var}(u\sharp\mu)$ is strictly concave. We derive

$$\mathrm{Var}(u\sharp\mu) = \frac{1}{2}\int_{\mathcal{Y}\times\mathcal{Y}} \|u(y) - u(y')\|_{\mathcal{H}}^2 d\mu(y)d\mu(y') =$$

$$\frac{1}{2}\int_{\mathcal{Y}} \|u(y)\|_{\mathcal{H}}^2 d\mu(y) - \int_{\mathcal{Y}\times\mathcal{Y}} \langle u(y), u(y')\rangle_{\mathcal{H}} d\mu(y)d\mu(y') + \frac{1}{2}\int_{\mathcal{Y}} \|u(y')\|_{\mathcal{H}}^2 d\mu(y') =$$

$$\int_{\mathcal{Y}} \|u(y)\|_{\mathcal{H}}^2 d\mu(y) - \langle \int_{\mathcal{Y}} u(y)d\mu(y), \int_{\mathcal{Y}} u(y')d\mu(y')\rangle_{\mathcal{H}} =$$

$$\int_{\mathcal{Y}} \|u(y)\|_{\mathcal{H}}^2 d\mu(y) - \|\int_{\mathcal{Y}} u(y)d\mu(y)\|_{\mathcal{H}}^2. \quad (51)$$

The first term in (51) is linear in $\mu$, so it suffices to prove that $\|\int_{\mathcal{Y}} u(y)d\mu(y)\|_{\mathcal{H}}^2 = \|u(\mu)\|_{\mathcal{H}}^2$ is strictly convex. To do this, we pick any $\mu_1 \neq \mu_2 \in \mathcal{P}(\mathcal{X})$, $\alpha \in (0,1)$. Since the kernel $k$ is characteristic, it holds that $u(\mu_1) \neq u(\mu_2)$. The squared norm function $\|\cdot\|_{\mathcal{H}}^2$ is strictly convex. As a result, the following strict inequality holds

$$\alpha\|u(\mu_1)\|_{\mathcal{H}}^2 + (1-\alpha)\|u(\mu_2)\|_{\mathcal{H}}^2 > \|u(\alpha\mu_1 + (1-\alpha)\mu_2)\|_{\mathcal{H}}^2,$$

which yields strict convexity of $\|u(\mu)\|_{\mathcal{H}}^2$. Consequently, $C_{k,\gamma}(x,\mu)$ is strictly convex in $\mu$. In this case, the weak OT functional $\pi \mapsto \int_{\mathcal{X}} C_{k,\gamma}(x, \pi(\cdot|x))d\mathbb{P}(x)$ is also strictly convex and yields the unique minimizer $\pi^* \in \Pi(\mathbb{P}, \mathbb{Q})$ which is the OT plan (Backhoff-Veraguas et al., 2019, §1.3.1). $\qquad\square$

*Proof of Theorem 2.* To begin with, we expand the functional $\mathcal{L}$:

$$T^* \in \arg\inf_T \mathcal{L}(f^*, T) \iff T^* \in \arg\inf_T \int_{\mathcal{X}} \left( C(x, T_x\sharp\mathbb{S}) - \int_{\mathcal{Z}} f^*(T_x(z))d\mathbb{S}(z) \right) d\mathbb{P}(x). \quad (52)$$

Define $\mu_x^* \overset{def}{=} T_x^*\sharp\mathbb{S}$. We are going to prove that $\mu_x^* \equiv \pi^*(y|x)$, where $\pi^* \in \Pi(\mathbb{P}, \mathbb{Q})$ is the (unique) optimal plan; this yields that $T^*$ is a stochastic OT map. Since the optimization over functions in NOT equals to the optimization over distributions that they generate (Korotin et al., 2023, §4.1), we have

$$\{\mu_x^*\} \in \arg\inf_{\{\mu_x\}} \int_{\mathcal{X}} \left( C(x, \mu_x) - \int_{\mathcal{Y}} f^*(y)d\mu_x(y) \right) d\mathbb{P}(x), \quad (53)$$

where the inf is taken over collections of distributions $\mu_x \in \mathcal{P}(\mathcal{Y})$ indexed by $\mathcal{X}$. Importantly, (53) can be split into $x \in \mathcal{X}$ independent problems, i.e., we have that

$$\mu_x^* \in \arg\inf_\mu \left[ C(x, \mu_x) - \int_{\mathcal{Y}} f^*(y)d\mu_x(y) \right] \quad (54)$$

holds true $\mathbb{P}$-almost surely for all $x \in \mathcal{X}$. Note that the functional $\mu \mapsto C(x,\mu) - \int_{\mathcal{Y}} f^*(y)d\mu(y)$ consists of a strictly convex term $C(x,\mu)$, which follows from the proof of Lemma 5, and a linear term (integral over $\mu$). Therefore, the functional itself is strictly convex. Since $\pi^*(y|x)$ minimizes this functional, it is the unique solution due to the strict convexity. Therefore, $\mu^*(x) = \pi^*(y|x)$ holds true $\mathbb{P}$-almost surely for $x \in \mathcal{X}$ and $T_x^*\sharp\mathbb{S} = \mu_x^* = \pi^*(y|x)$, i.e., $T^*$ is a stochastic OT map. $\qquad\square$

*Proof of Proposition 3.* Since $\pi_{\gamma_1}^*$ is optimal for the $\gamma_1$-weak cost, for all $\pi \in \Pi(\mathbb{P}, \mathbb{Q})$ it holds $\mathrm{Cost}_{k,\gamma_1}(\pi_{\gamma_1}^*) \leq \mathrm{Cost}_{k,\gamma_1}(\pi)$. In particular, for $\pi = \pi_{\gamma_2}^*$ it holds true that

$$\mathrm{Dist}_u^2(\pi_{\gamma_1}^*) - \gamma_1 \mathrm{CVar}_u(\pi_{\gamma_1}^*) \leq \mathrm{Dist}_u^2(\pi_{\gamma_2}^*) - \gamma_1 \mathrm{CVar}_u(\pi_{\gamma_2}^*). \quad (55)$$

Analogously, $\pi_{\gamma_2}^*$ is optimal for the $\gamma_2$-weak cost; the following holds:

$$\text{Dist}_u^2(\pi_{\gamma_2}^*) - \gamma_2 \, \text{CVar}_u(\pi_{\gamma_2}^*) \le \text{Dist}_u^2(\pi_{\gamma_1}^*) - \gamma_2 \, \text{CVar}_u(\pi_{\gamma_1}^*). \tag{56}$$

We sum (55) and (56) and obtain

$$-\gamma_1 \, \text{CVar}_u(\pi_{\gamma_1}^*) - \gamma_2 \, \text{CVar}_u(\pi_{\gamma_2}^*) \le -\gamma_1 \, \text{CVar}_u(\pi_{\gamma_2}^*) - \gamma_2 \, \text{CVar}_u(\pi_{\gamma_1}^*),$$

or, equivalently,

$$(\gamma_2 - \gamma_1) \, \text{CVar}_u(\pi_{\gamma_1}^*) \le (\gamma_2 - \gamma_1) \, \text{CVar}_u(\pi_{\gamma_2}^*),$$

which is equivalent to $\text{CVar}_u(\pi_{\gamma_1}^*) \le \text{CVar}_u(\pi_{\gamma_2}^*)$ since $\gamma_2 - \gamma_1 > 0$. Now we multiply (55) and (56) by $\gamma_2$ and $\gamma_1$, respectively, and sum the resulting inequalities. We obtain

$$\gamma_2 \, \text{Dist}_u^2(\pi_{\gamma_1}^*) + \gamma_1 \, \text{Dist}_u^2(\pi_{\gamma_2}^*) \le \gamma_2 \, \text{Dist}_u^2(\pi_{\gamma_2}^*) + \gamma_1 \, \text{Dist}_u^2(\pi_{\gamma_1}^*),$$

or, equivalently,

$$(\gamma_2 - \gamma_1) \, \text{Dist}_u^2(\pi_{\gamma_1}^*) \le (\gamma_2 - \gamma_1) \, \text{Dist}_u^2(\pi_{\gamma_2}^*),$$

which provides $\text{Dist}_u(\pi_{\gamma_1}^*) \le \text{Dist}_u(\pi_{\gamma_2}^*)$. Finally, we note that

$$\text{Cost}_{k,\gamma_2}(\pi_{\gamma_2}^*) = \inf_{\pi \in \Pi(\mathbb{P}, \mathbb{Q})} \text{Cost}_{k,\gamma_2}(\pi) \le \text{Cost}_{k,\gamma_2}(\pi_{\gamma_1}^*) =$$

$$\text{Cost}_{k,\gamma_1}(\pi_{\gamma_1}^*) - \underbrace{(\gamma_2 - \gamma_1)}_{\ge 0} \underbrace{\text{CVar}_u(\pi_{\gamma_1}^*)}_{\ge 0} \ge \text{Cost}_{k,\gamma_1}(\pi_{\gamma_1}^*), \tag{57}$$

which concludes the proof. $\qquad\square$

## H  WEAK QUADRATIC VS. KERNEL COSTS ON REAL DATA

The goal of this section is to demonstrate that our proposed kernel cost (12) consistently outperforms the weak quadratic cost (3) in the downstream task of unpaired image translation. We consider the distance-induced kernel $k(x, y) = \frac{1}{2}\|x\| + \frac{1}{2}\|y\| - \frac{1}{2}\|x - y\|$. In short, we run NOT (Korotin et al., 2023, Algorithm 1) multiple times (with various random seeds) with the same hyperparameters (Appendix I) for weak and kernel costs, and then we compare the obtained FID ($\mu \pm \sigma$).

**Datasets.** We consider *shoes → handbags* and *celeba (female) → anime* translation. We work only with small $32 \times 32$ images to speed up the training and be able to run many experiments.

**EXPERIMENT 1.** We consider the $\gamma$-**weak** quadratic cost for each $\gamma \in \{\frac{1}{2}, 1, \frac{3}{2}, 2\}$ we run NOT with 5 different random seeds and train it for 60k iterations of $f_\omega$. In this experiment, during training, we evaluate *test* FID every 1k iteration and for each experiment we report 3 FID values: the **best** FID value of iterations 25-60k[5] indicating *what the model can achieve best*, the **max** FID value of iterations 25-60k indicating *what the model achieves worst* (because of potential training instabilities) and the **last** FID value at the end of training (60k) showing *what the model actually achieved*.

The experimental results of Tables 3, 4 provide several important insights. First, we see that with the increase of parameter $\gamma$, the *best* FID stably increases. Second, the overall training becomes less stable: *max* FID becomes extremely large, which indicates severe fluctuations of the model. In particular, both the mean and standard deviation of *last* FID (as well as *max* FID) drastically increase.

Why does this happen? We think that with the increase of $\gamma$ sets $U_\psi(\cdot)$ (11) become large. These sets determine how much a fake solution may vary (Theorem 1). Thus, this naturally leads to high ambiguity of the solutions and results in unstable and unpredictable behaviour.

**EXPERIMENT 2.** We pick the **highest** considered value $\gamma = 2$ and show that our $\gamma$-weak kernel cost performs better than the weak quadratic cost from the previous experiments.

We run NOT with the kernel cost 5 times and report the results in the same Tables 3, 4. The results show that even for high $\gamma$ the issues with the fluctuation are notably softened. This is seen from the fact that for $\gamma = 2$ the gap between the *last* FID (or *max* FID) and *best* much smaller for the kernel cost than for the quadratic cost. In particular, this gap is comparable to the gap for $\gamma = \frac{1}{2}$-weak quadratic cost for which sets $U_\psi(\cdot)$ are presumably small and provide less ambiguity to the solutions.

**Conclusion.** Our empirical evaluation shows that NOT with our proposed kernel costs yields more stable behaviour than NOT with the weak quadratic cost. This agrees with our theory which suggests that one of the reasons for unstable behaviour and severe fluctuations might be the existence of the fake solutions (§3.1). Our weak kernel cost removes all the fake solutions (§3.2).

---

[5]We choose 25k iterations as the starting point because at this time point FID roughly stabilizes at a small level. This indicates the model has nearly converged and starts fluctuating around the optimum.

| Method | Weak quadratic, $C_{2,\gamma}$ | | | | Weak kernel $C_{k,\gamma}$ [Ours] |
|--------|--------|--------|--------|--------|--------|
| Setting | $\gamma = 0.5$ | $\gamma = 1$ | $\gamma = 1.5$ | $\gamma = 2$ | $\gamma = 2$ |
| Best FID | $16.08 \pm 0.37$ | $19.87 \pm 0.69$ | $23.01 \pm 5.16$ | $25.36 \pm 1.26$ | $16.50 \pm 0.88$ |
| Last FID | $21.98 \pm 3.04$ | $33.56 \pm 10.49$ | $30.35 \pm 8.85$ | $37.18 \pm 8.58$ | $20.19 \pm 2.36$ |
| Max FID | $28.40 \pm 3.18$ | $52.03 \pm 16.00$ | $46.59 \pm 9.53$ | $67.50 \pm 30.76$ | $29.85 \pm 3.75$ |

Table 3: Test FID↓ ($\mu \pm \sigma$) on *shoes → handbags*, $32 \times 32$ of $C_{2,\gamma}$ and $C_{k,\gamma}$ for different $\gamma$. Red color highlights $\mu \geq 30$, orange color highlights $\sigma \geq 5$.

| Method | Weak quadratic, $C_{2,\gamma}$ | | | | Weak kernel $C_{k,\gamma}$ [Ours] |
|--------|--------|--------|--------|--------|--------|
| Setting | $\gamma = 0.5$ | $\gamma = 1$ | $\gamma = 1.5$ | $\gamma = 2$ | $\gamma = 2$ |
| Best FID | $18.88 \pm 0.57$ | $34.71 \pm 6.48$ | $24.49 \pm 0.64$ | $40.49 \pm 5.57$ | $19.39 \pm 1.06$ |
| Last FID | $20.26 \pm 1.98$ | $36.89 \pm 8.19$ | $28.49 \pm 1.66$ | $49.66 \pm 5.45$ | $20.62 \pm 1.66$ |
| Max FID | $29.51 \pm 1.45$ | $81.78 \pm 25.93$ | $48.44 \pm 5.14$ | $76.25 \pm 4.26$ | $36.86 \pm 0.45$ |

Table 4: Test FID↓ ($\mu \pm \sigma$) on *celeba (female) → anime*, $32 \times 32$ of $C_{2,\gamma}$ and $C_{k,\gamma}$ for different $\gamma$. Red color highlights $\mu \geq 40$, orange color highlights $\sigma \geq 4$.

# I  ADDITIONAL TRAINING DETAILS

**Pre-processing.** In all the cases, we rescale RGB channels of images from $[0, 1]$ to $[-1, 1]$. As in (Korotin et al., 2023), we beforehand rescale anime face images to $512 \times 512$, and do $256 \times 256$ crop with the center located 14 pixels above the image center to get the face. Next, for all the datasets except for the describable textures, we resize images to the required size ($64 \times 64$ or $128 \times 128$). Specifically for the describable textures dataset ($\approx$5K textures), we augment the samples. We rescale input textures to minimal border size of 300, do the random resized crop (from 128 to 300 pixels) and random horizontal & vertical flips. Then we resize images to the required size ($64 \times 64$ or $128 \times 128$).

**Neural networks.** We use WGAN-QC discriminator's ResNet architecture (Liu et al., 2019) for potential $f$. We use UNet[6] (Ronneberger et al., 2015) as the stochastic transport map $T(x, z) = T_x(z)$. To condition it on $z$, we insert conditional instance normalization (CondIN) layers after each UNet's upscaling block[7]. We use CondIN from AugCycleGAN[8] (Almahairi et al., 2018). In experiments, $z$ is the 128-dimensional standard Gaussian noise.

**Optimization.** To learn stochastic OT maps, we use NOT algorithm (Korotin et al., 2023, Algorithm 1). We use the Adam optimizer (Kingma & Ba, 2014) with the default betas for both $T_\theta$ and $f_\omega$. The learning rate is $lr = 1 \cdot 10^{-4}$. We use the MultiStepLR scheduler which decreases $lr$ by 2 after [15k, 25k, 40k, 55k, 70k] (iterations of $f_\omega$). The batch size is $|X| = 64$, $|Z_x| = 4$. The number of inner iterations is $k_T = 10$. In toy experiments, we do 10K total iterations of $f_\omega$ update. In the image-to-image translation experiments, we observe convergence in $\approx$ 70k iterations for $128 \times 128$ datasets, in $\approx$ 40k iterations for $64 \times 64$ datasets. In image-to-image translation, we gradually change $\gamma$. Starting from $\gamma = 0$, we linearly increase it to the desired value (mostly $\frac{1}{3}$) during 25K first iterations of $f_\omega$.

**Computational complexity.** NOT with kernel costs for $128 \times 128$ images converges in 3-4 days on a $4\times$ Tesla V100 GPUs (16 GB). This is slightly bigger than the respective time for NOT (Korotin et al., 2023, §6) with the quadratic cost due to the reasons discussed in §3.3.

---

[6] github.com/milesial/Pytorch-UNet
[7] github.com/kgkgzrtk/cUNet-Pytorch
[8] github.com/ErfanMN/Augmented_CycleGAN_Pytorch

## J COMPARISON WITH NOT WITH THE QUADRATIC COST

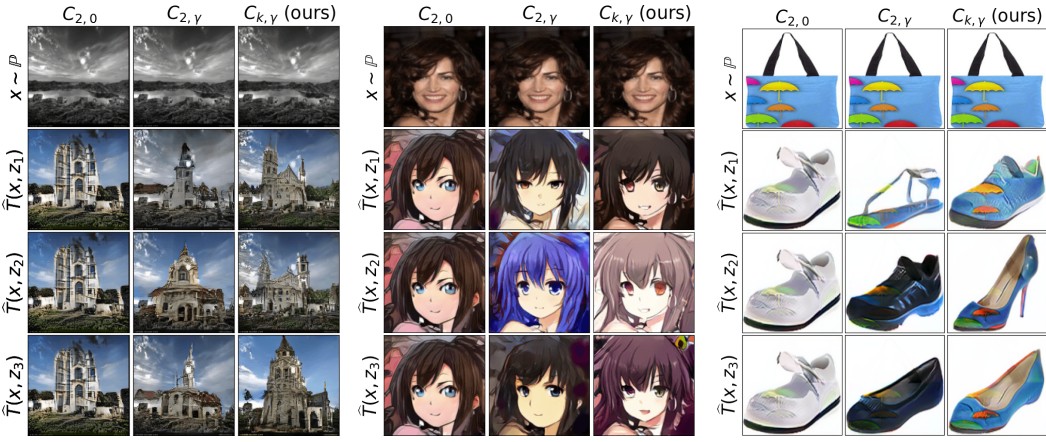

(a) Outdoor → church, $128 \times 128$.    (b) Celeba (f) → anime, $128 \times 128$.    (c) Handbags → shoes, $128 \times 128$.

Figure 15: Unpaired translation with Neural Optimal Transport with various costs ($C_{2,0}, C_{2,\gamma}, C_{k,\gamma}$).

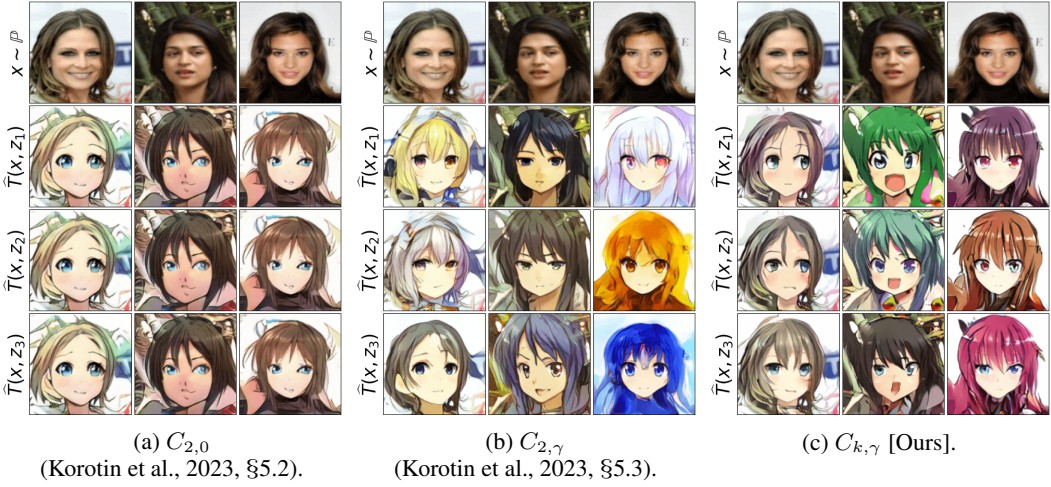

(a) $C_{2,0}$
(Korotin et al., 2023, §5.2).

(b) $C_{2,\gamma}$
(Korotin et al., 2023, §5.3).

(c) $C_{k,\gamma}$ [Ours].

Figure 16: Celeba (f) → anime ($128 \times 128$) translation with NOT with costs $C_{2,0}, C_{2,\gamma}, C_{k,\gamma}$.

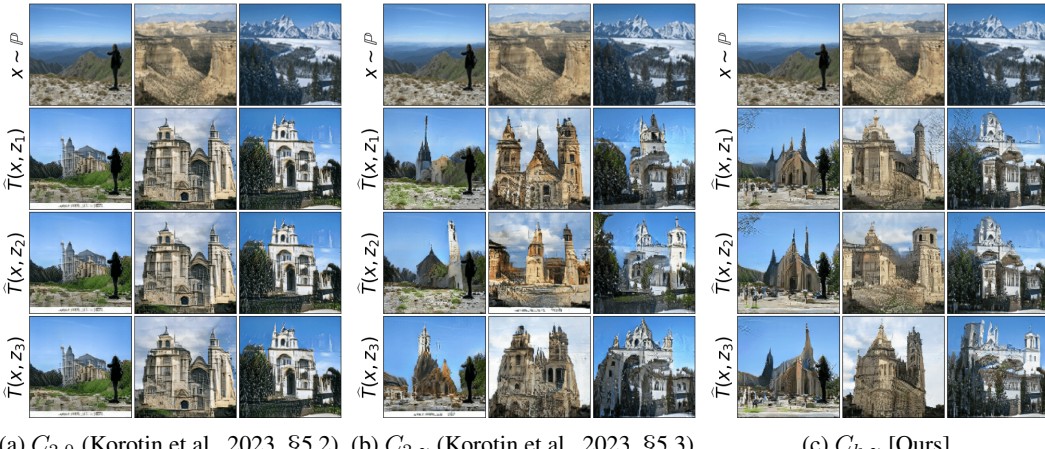

(a) $C_{2,0}$ (Korotin et al., 2023, §5.2). (b) $C_{2,\gamma}$ (Korotin et al., 2023, §5.3).    (c) $C_{k,\gamma}$ [Ours].

Figure 17: Outdoor → church ($128 \times 128$) translation with NOT with costs $C_{2,0}, C_{2,\gamma}, C_{k,\gamma}$.

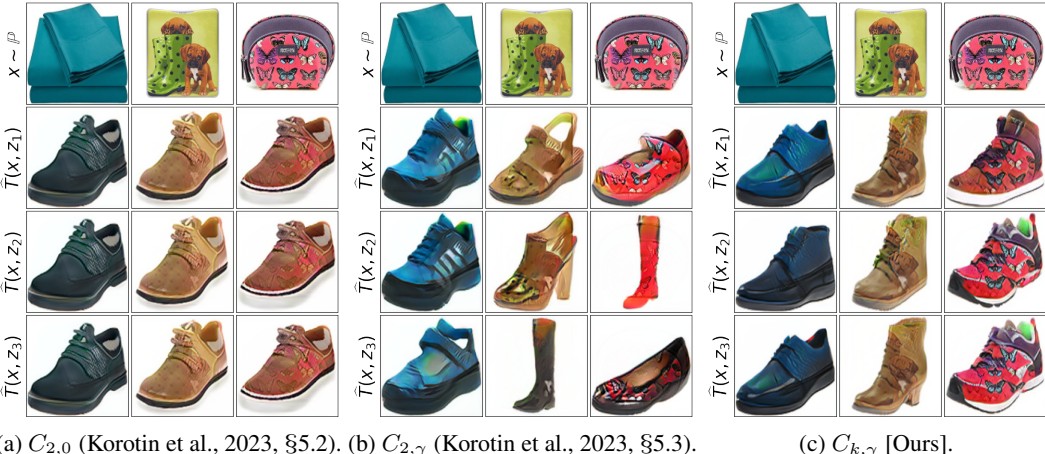

(a) $C_{2,0}$ (Korotin et al., 2023, §5.2). (b) $C_{2,\gamma}$ (Korotin et al., 2023, §5.3).  (c) $C_{k,\gamma}$ [Ours].

Figure 18: Handbags → shoes ($128 \times 128$) translation with NOT with costs $C_{2,0}$, $C_{2,\gamma}$, $C_{k,\gamma}$.

## K  COMPARISON WITH IMAGE-TO-IMAGE TRANSLATION METHODS

We compare NOT with our kernel costs with *principal* models (one-to-one and one-to-many) for unpaired image-to-image translation. We consider CycleGAN [9](Zhu et al., 2017), AugCycleGAN[10] (Almahairi et al., 2018) and MUNIT[11] (Huang et al., 2018) for comparison. We use the official or community implementations with the hyperparameters from the respective papers. We consider *outdoor → church* and *texture → shoes* dataset pairs ($128 \times 128$). The FID scores are given in Table 5. Qualitative examples are shown in Figures 20, 19.

| Method | One-to-one | | One-to-many | | |
|:---:|:---:|:---:|:---:|:---:|:---:|
| **Datasets** ($128 \times 128$) | Cycle GAN (with $\mathcal{L}_1$ loss) | Cycle GAN (no $\mathcal{L}_1$ loss) | AugCycleGAN | MUNIT | NOT with $C_{k,\gamma}$ (**Ours**) |
| *Outdoor → church* | 43.74 | 36.16 | $51.15 \pm 0.19$ | $32.14 \pm 0.18$ | $\mathbf{15.16} \pm 0.03$ |
| *Texture → shoes* | $34.65 \pm 0.12$ | $50.95 \pm 0.12$ | N/A | $43.74 \pm 0.16$ | $\mathbf{24.84} \pm 0.09$ |

Table 5: Test FID↓ of the considered image-to-image translation methods.

We do not include the results of AugCycleGAN on *texture→shoes* as it did not converge on these datasets (FID $\gg$ 100). We tried tuning its hyperparameters, but this did not yield improvement.

---

[9]github.com/eriklindernoren/PyTorch-GAN/tree/master/implementations/cyclegan
[10]github.com/aalmah/augmented_cyclegan
[11]github.com/NVlabs/MUNIT

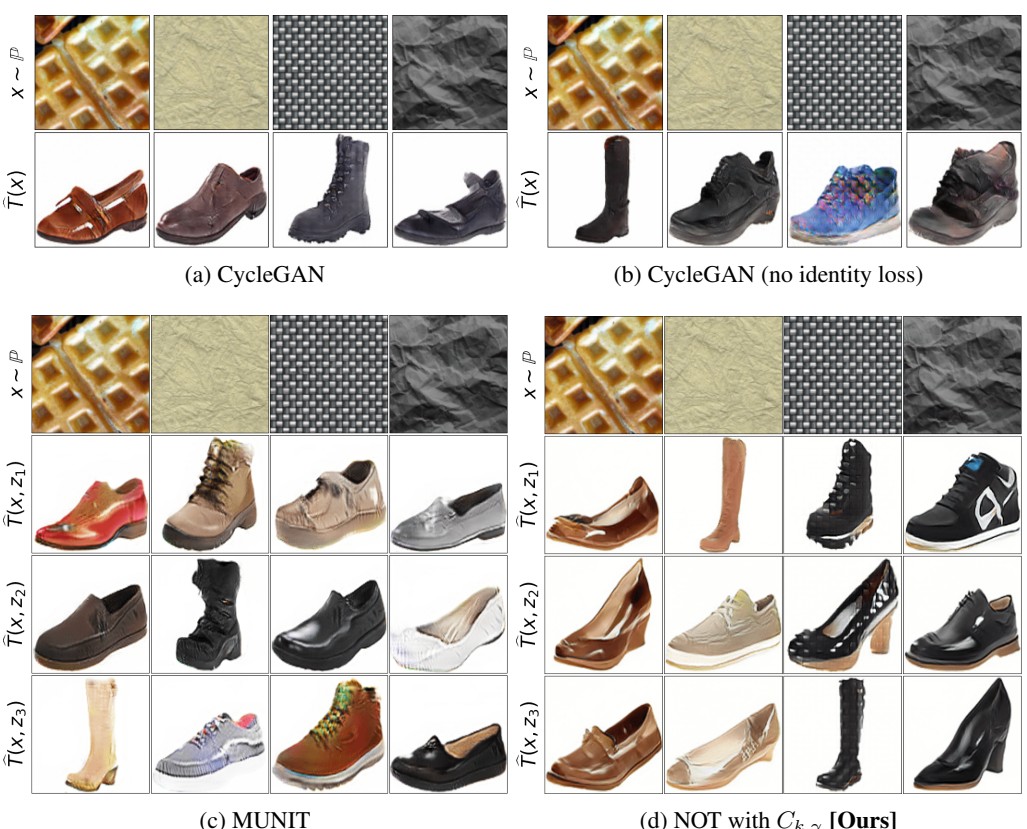

Figure 19: Texture $\rightarrow$ shoes ($128 \times 128$) translation with various methods.

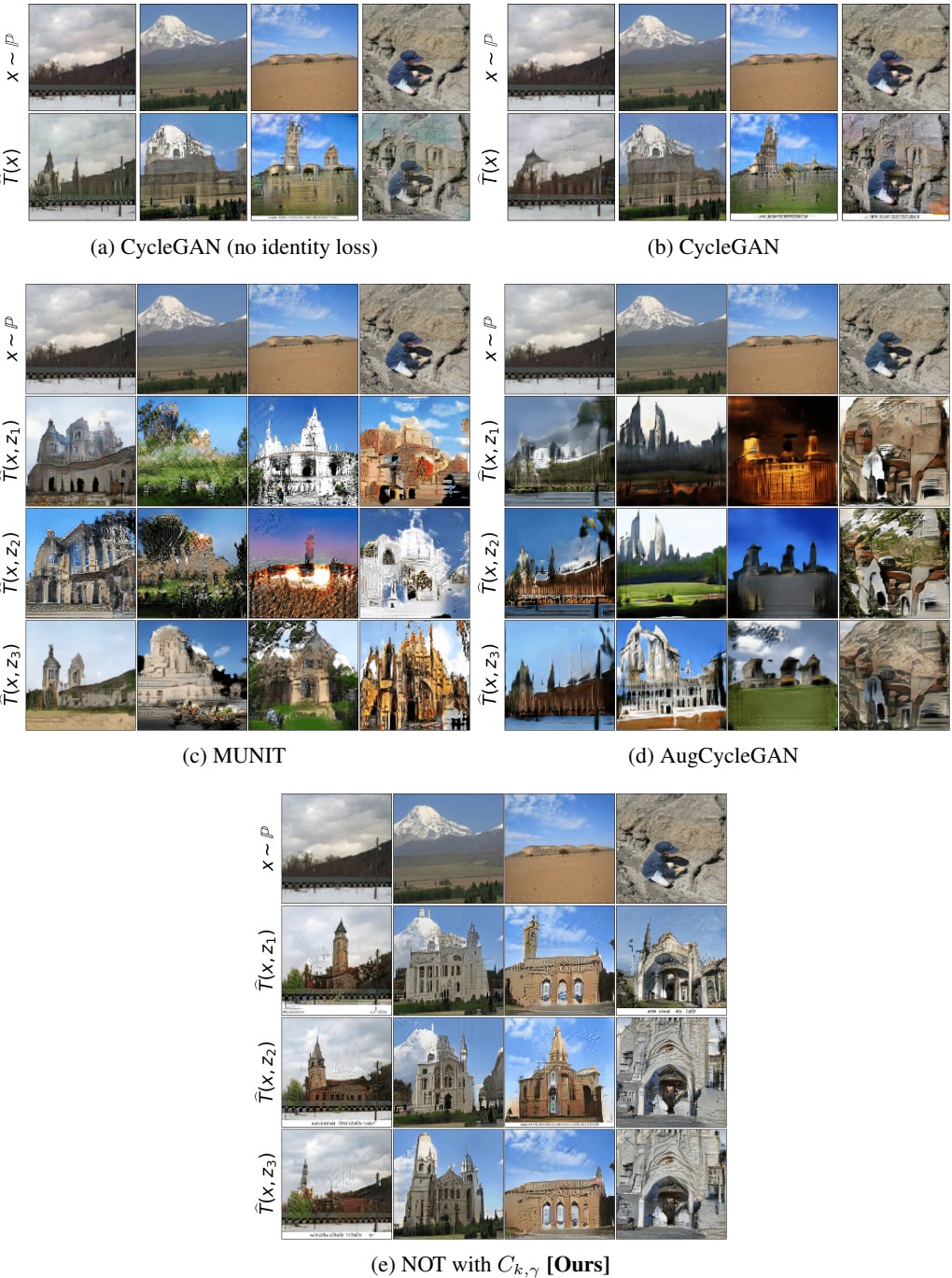

Figure 20: Outdoor → church (128 × 128) translation with various methods.

## L  ADDITIONAL EXPERIMENTAL RESULTS

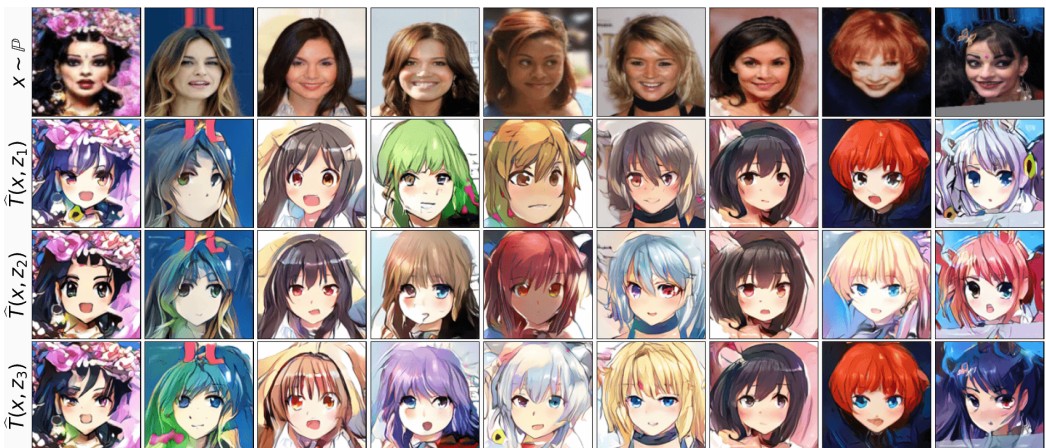

(a) Input images $x$ and random translated examples $T_x(z)$.

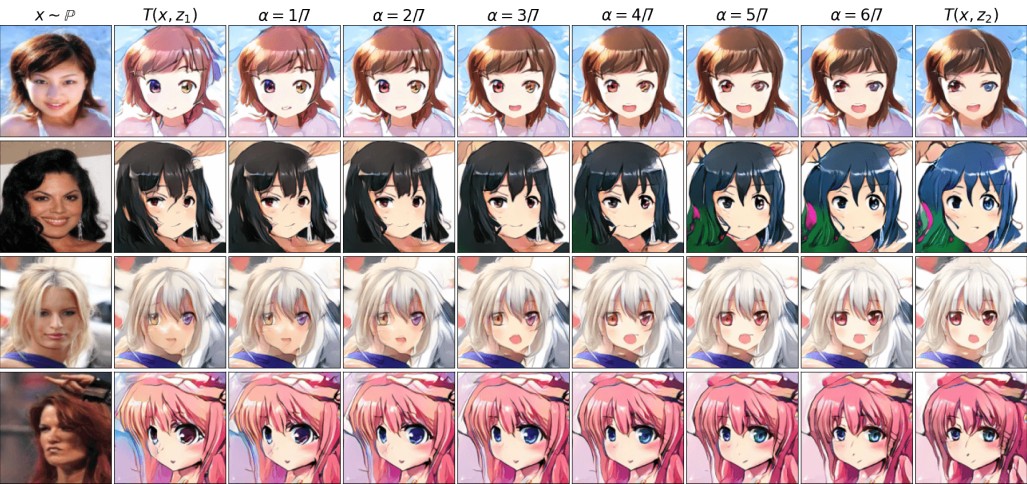

(b) Interpolation in the conditional latent space, $z = (1 - \alpha)z_1 + \alpha z_2$.

Figure 21: Celeba (female) $\to$ anime translation, $128 \times 128$. Additional examples.

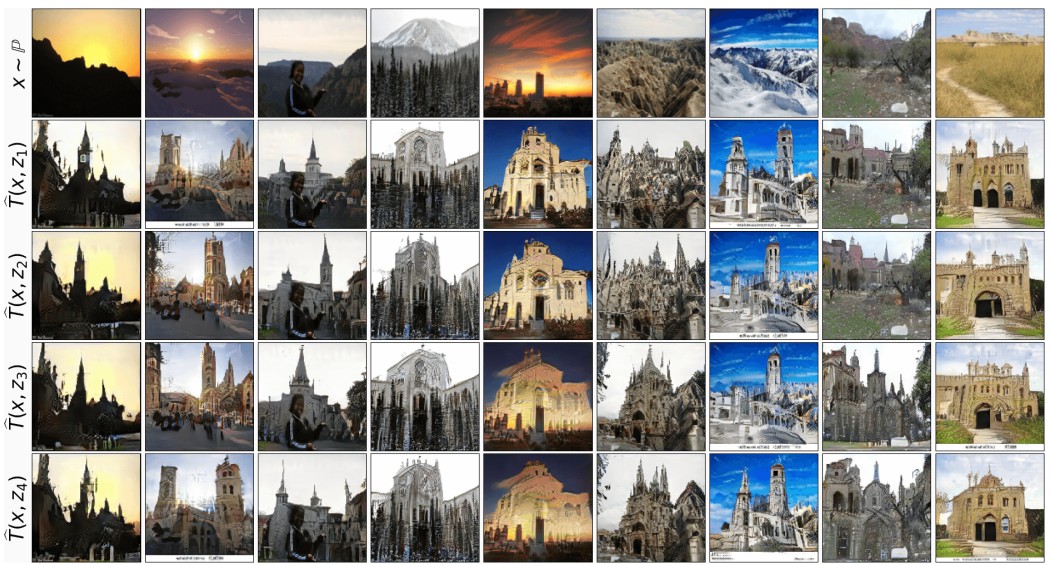

(a) Input images $x$ and random translated examples $T_x(z)$.

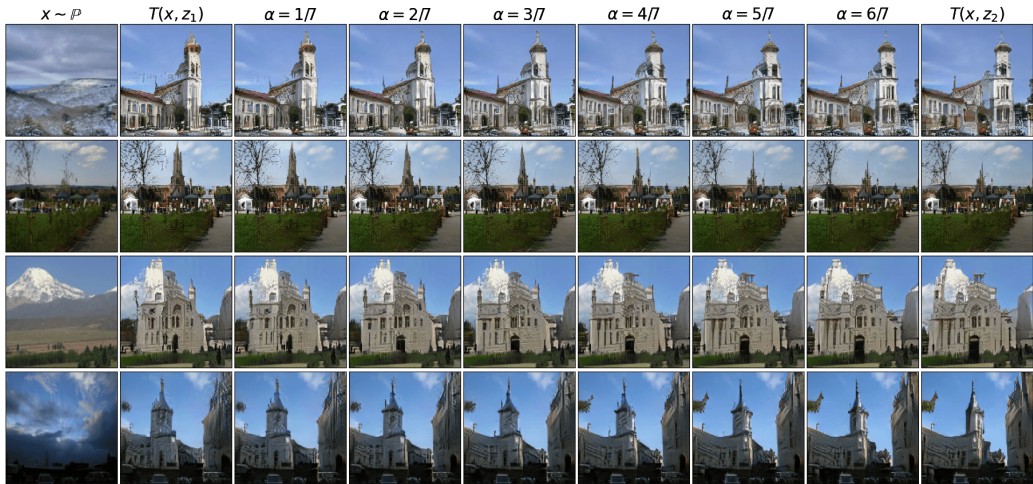

(b) Interpolation in the conditional latent space, $z = (1 - \alpha)z_1 + \alpha z_2$.

Figure 22: Outdoor $\rightarrow$ church, $128 \times 128$. Additional examples.

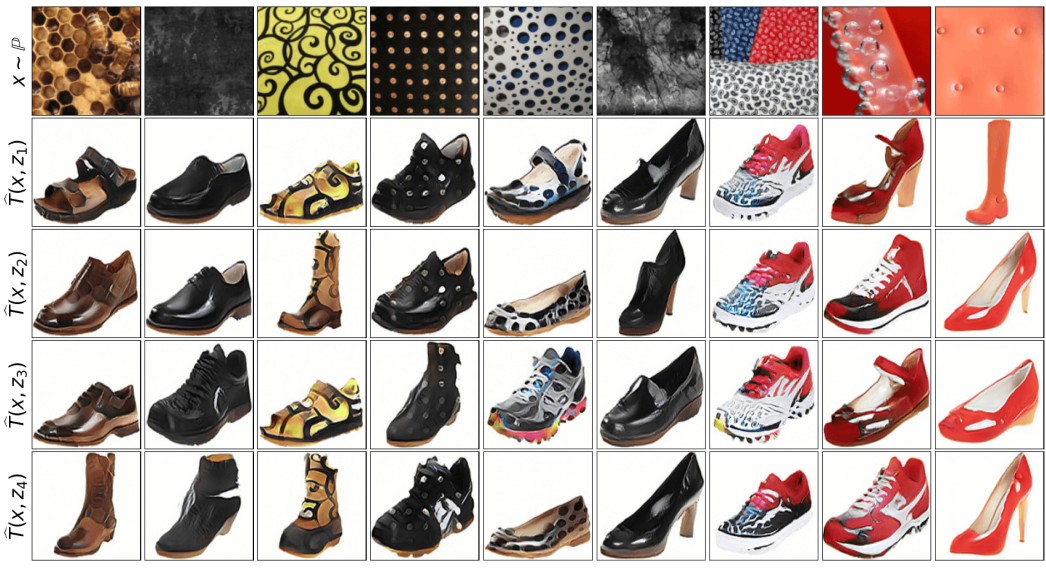

(a) Input images $x$ and random translated examples $T_x(z)$.

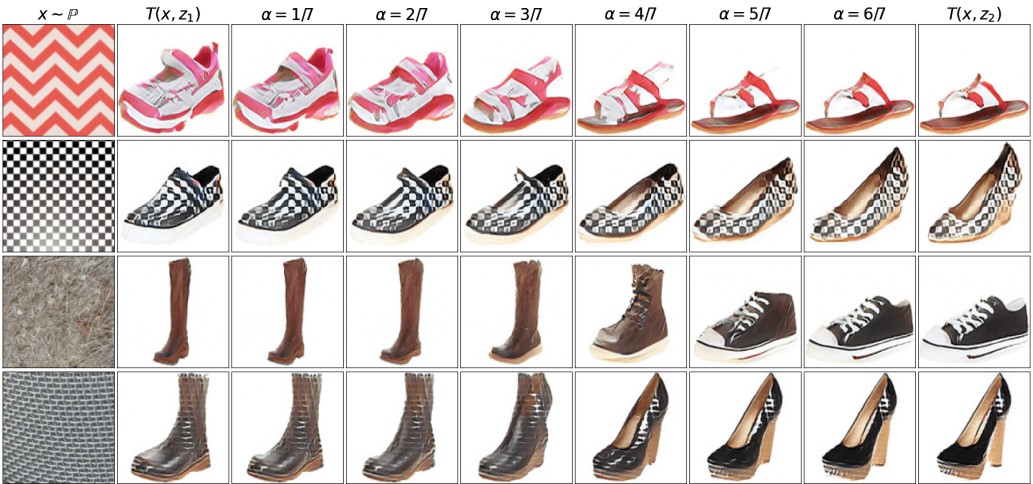

(b) Interpolation in the conditional latent space, $z = (1 - \alpha)z_1 + \alpha z_2$.

Figure 23: Texture $\to$ shoes translation, $128 \times 128$. Additional examples.

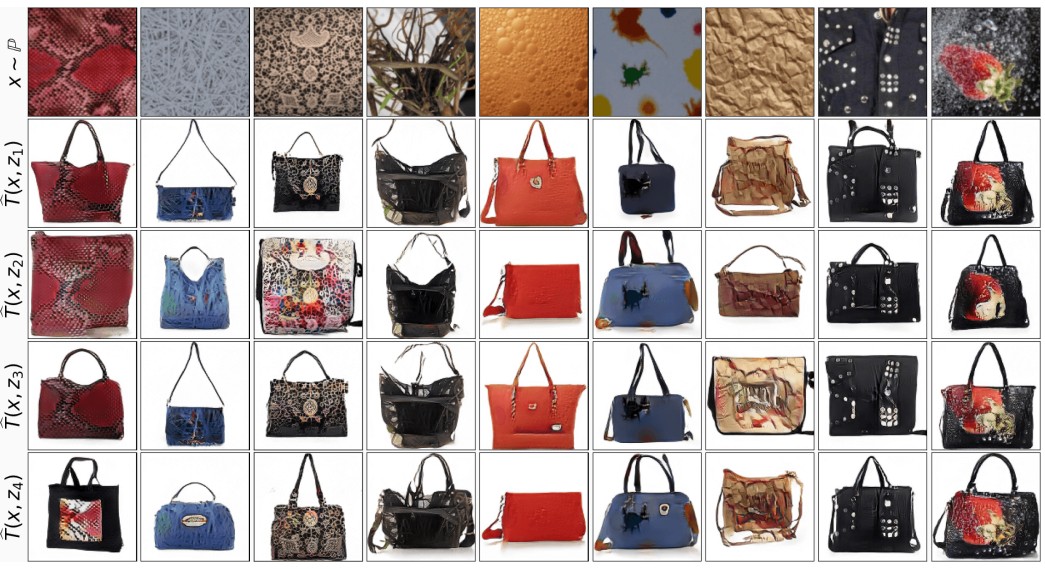

(a) Input images $x$ and random translated examples $T_x(z)$.

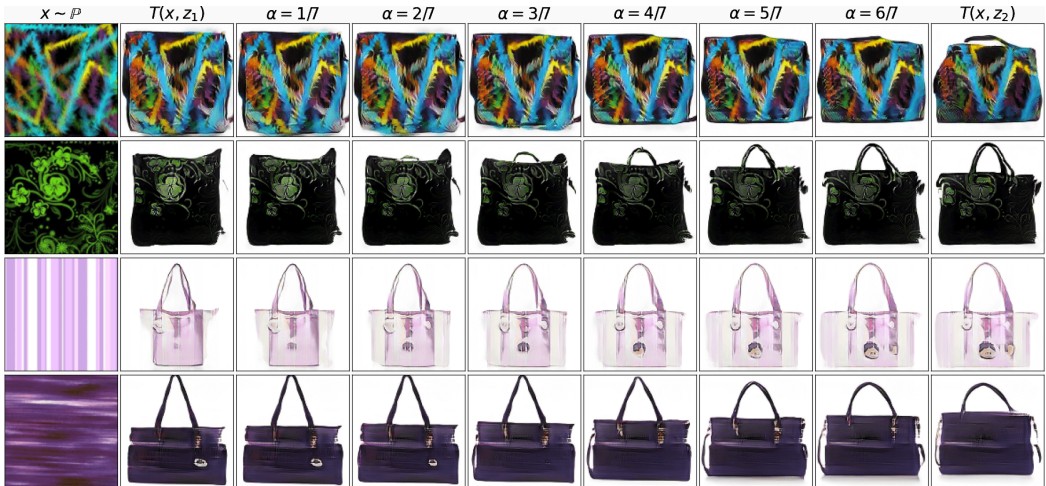

(b) Interpolation in the conditional latent space, $z = (1 - \alpha)z_1 + \alpha z_2$.

Figure 24: Texture $\rightarrow$ handbags translation, $128 \times 128$. Additional examples.

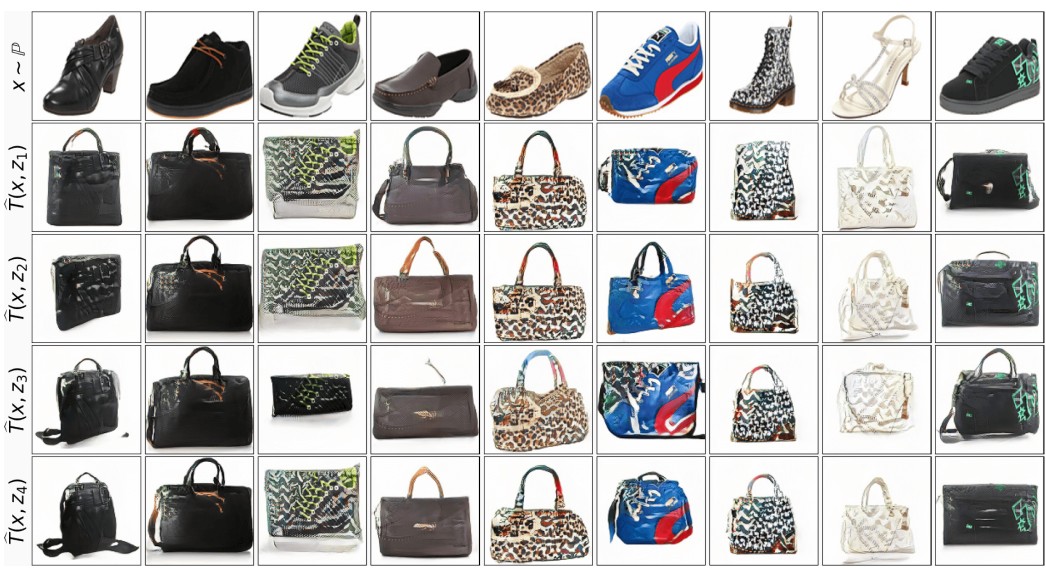

(a) Input images $x$ and random translated examples $T_x(z)$.

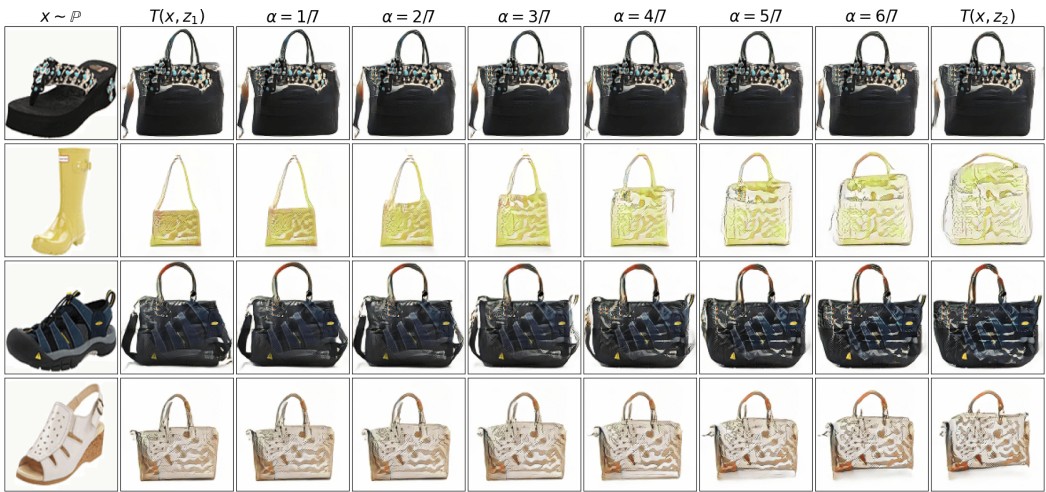

(b) Interpolation in the conditional latent space, $z = (1 - \alpha)z_1 + \alpha z_2$.

Figure 25: Shoes $\rightarrow$ handbags translation, $128 \times 128$. Additional examples.

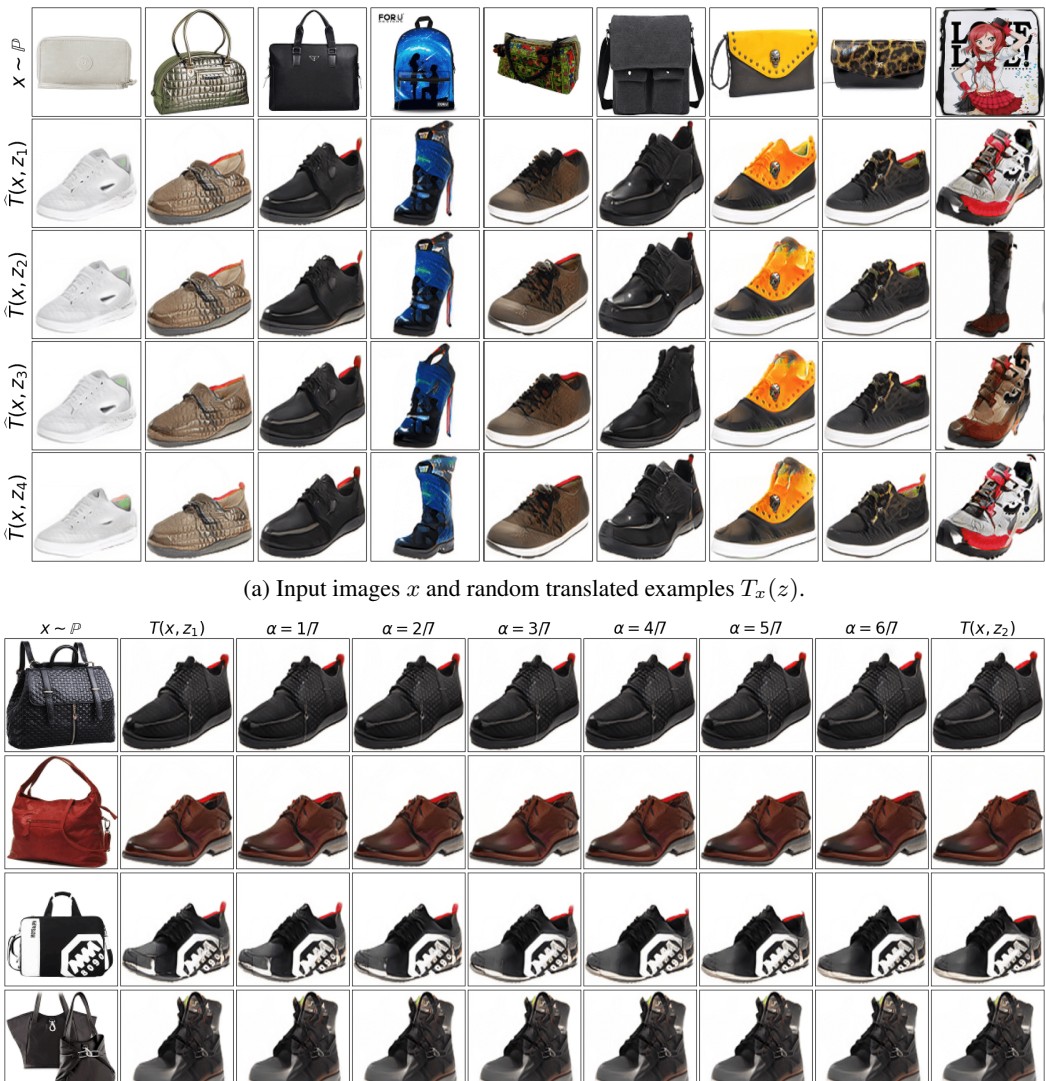

(a) Input images $x$ and random translated examples $T_x(z)$.

(b) Interpolation in the conditional latent space, $z = (1 - \alpha)z_1 + \alpha z_2$.

Figure 26: Handbags $\rightarrow$ shoes translation, $128 \times 128$. Additional examples.

