# OpenReview forum: "Kernel Neural Optimal Transport"
_ICLR.cc/2023/Conference — ICLR 2023 poster_

### Official Review · Reviewer_QYaw · 2022-10-20

**Confidence:** 3
**Clarity, Quality, Novelty And Reproducibility:** Exposition is clear and easy to read.…
**Correctness:** 4
**Technical Novelty And Significance:** 3
**Empirical Novelty And Significance:** 3
**Recommendation:** 8

**Strength And Weaknesses:**

First, it looks like existence of fake plans is a result of some kind of degeneracy in the problem formulation (like the multicollinearity problem in linear regression). Can it be tackled by some form of regularization? Interesting question was asked on page 6: "Should we really care about solutions being fake?" and noted that "the map T may fluctuate between fake solutions rather than converge." The question is: does it fluctuate between fake solutions, or, in fact, it fluctuates as some convex combination of stochastic OT plans? The Proposition 2 hints that it could be the second case.
Another issue is that existence of fake plans are demonstrated on toy experiments. A natural question is: are fake plans common when the technique is applied to real-world data, e.g. to unpaired image-to-image translation task?

Minor comments:
Page 1: (a) Celeba (female) → shoes  ->   (a) Celeba (female) → anime
Page 2:  fomulations ->  formulations
Page 8: under perform -> underperform

**Summary Of The Paper:**

Authors propose a novel technique to compute weak optimal transport plan between two distributions, given in possibly two different spaces. Based on an earlier work of Gozlan et al, the computation of OT plan is formulated as a certain minimax task. The major issue that the paper addresses is that some saddle points of the latter task can correspond to fake plans. Existence of fake plans is proven theoretically and demonstrated on toy experiments.

**Summary Of The Review:**

Overall, a quantity and a quality of experiments convinces me in correctness of major points of the paper. Though, for me it is hard to judge experiments with unpaired image-to-image translation task. Some discussions of  unpaired image-to-image translation task, or a citation where one can learn details of that problem is needed.

---

> ### Author Response · Authors · 2022-11-16
> **Response to Reviewer QYaw**
>
> Thank you for your valuable feedback and high appreciation of our contribution. Please below find below our answers to your questions.
>
> **(1) First, it looks like existence of fake plans is a result of some kind of degeneracy in the problem formulation (like the multicollinearity problem in linear regression). Can it be tackled by some form of regularization?**
>
> We think that this degeneracy in the NOT formulation comes from the **not-strict** convexity of the *weak quadratic cost* in $\mu$. Indeed, in Appendix E, when we prove our **main Theorem 2** (the non-existence of fake solutions for *our proposed kernel costs*), we rely on the **strict** convexity our proposed negative kernel variance. In contrast, the vanilla variance term $-\mbox{Var}(\cdot)$ is not strictly convex and the same analysis can not be applied. Consider two observations:
>
> - For any two distributions $\mu,\nu\in\mathcal{P}(\mathcal{Y})$ with the same **expectation**, the variance is linear on $[\mu,\nu]$, i.e., for $\alpha\in [0,1]$:
> $$-\alpha\mbox{Var}(\mu)-(1-\alpha)\mbox{Var}(\nu)=-\mbox{Var}(\alpha\mu+(1-\alpha)\nu).$$
>
> - In our key **Theorem 1**, we prove that all the fake solutions $T_{x}(z)$ share the same **expectation** (for each $x$).
>
> These two observations seem to be interconnected. A reasonable hypothesis here is that the **degeneracy** (existence of fake solutions) for weak costs $C(x,\mu)$ may appear if there are some areas in $\mathcal{P}(\mathcal{Y})$ where $C(x,\mu)$ is not strictly convex in $\mu$. Technically, the degeneracy issue might be resolved by adding strictly convex (in $\mu$) terms $R(\mu)$ (**regularizers**) to the weak cost $C(x,\mu)$, or just by considering the costs which is strictly convex, e.g., our proposed kernel costs.
>
> **(2) The question is: does it fluctuate between fake solutions, or, in fact, it fluctuates as some convex combination of stochastic OT plans?**
>
> We suppose that NOT with the weak quadratic cost might fluctuate between a large set of fake solutions which is **bigger** than just the convex combination of OT plans. For example, in Figures 12 and 13 of Appendix C, we see that some of the transported samples fall outside the convex support of the target distribution $\mathbb{Q}$ (square). Therefore, such learned $T_{x}(z)$ are **not** convex combinations of OT plans.
>
> **(3) A natural question is: are fake plans common when the technique is applied to real-world data, e.g. to unpaired image-to-image translation task?**
>
> Our experiments suggest that the answer to your question is *positive*. Please consider Appendix H. There we provide the assessment of stability of NOT with the weak quadratic cost. We show that **on the real data** with the increase of parameter $\gamma$, the overall training becomes less stable. We think that with the increase of $\gamma$ sets $U_{\psi}(\cdot)$ in our Theorem 1 become larger. According to our theoretical insights, these sets determine how much a fake solution may vary. Thus, this naturally results in unstable and unpredictable behaviour **as the model starts fluctuating between fake solutions.**
>
> **(4) Minor comments.**
>
> Thanks for pointing! We fixed them in the text.
>
> **Concluding remarks**. Please respond to our post to let us know if the clarifications above suitably address your concerns about our work. We are happy to address any remaining points during the discussion phase; if the responses above are sufficient, we kindly ask that you consider raising your review confidence.

---

### Official Review · Reviewer_84ix · 2022-10-24

**Confidence:** 5
**Correctness:** 2
**Technical Novelty And Significance:** 2
**Empirical Novelty And Significance:** 2
**Recommendation:** 5

**Clarity, Quality, Novelty And Reproducibility:**

The overall structure of this paper is clear, the shortcomings of NOT are analyzed theoretically,  and the KNOT algorithm is proposed, but the experimental comparison of the two algorithms is not given in the experiment, which lacks the support of the experiment.
Only an improvement of kernel cost function is given, which lacks some novelty. Can not be reproduced.

**Strength And Weaknesses:**

Strengths: This paper conducts the theoretical and empirical analysis of the saddle point optimization problem of NOT algorithm for the weak quadratic cost, this show it may appear fake solutions.
This paper introduces kernel quadratic costs to solve above problem, and it proves the feasibility of the proposed algorithm via theoretical analysis and examples.

Weaknesses: The cost function with kernel is too complex in practice. Comparison with other neural network-based algorithms for solving OT mappings is lacking.
This paper does not explain the advantages of the proposed algorithm compared with the classical algorithm, and there is no experimental support.

**Summary Of The Paper:**

This paper introduced kernel weak quadratic costs to solve the problem that NOT with the weak quadratic cost might learn fake plans which are not optimal.
And it conducts the theoretical and empirical analysis of the saddle point optimization problem of NOT algorithm for the weak quadratic cost.
Finally, they analyze from theory and examples to show that NOT generates fake solutions which do not provide an OT plan.

**Summary Of The Review:**

In this paper, a new neural optimal transport algorithm based on kernel cost function is proposed, and the deficiency of NOT is analyzed theoretically.
However, the lack of experimental support, the key is the lack of experimental comparison of NOT, does not reflect the advantages of improving the cost function.
Moreover, there is a lack of comparison of classical neural OT algorithms, such as ICNN.

---

> ### Author Response · Authors · 2022-11-16
> **Response to Reviewer 84ix + ICNN method results**
>
> Thank you for your valuable feedback. Please find above (in our reply to all the Reviewers) the answers to your comments common with other reviews. Please below find below our answers to your questions that do not overlap with those of other Reviewers.
>
> **(1) The cost function with the kernel is too complex in practice.**
>
> Our proposed weak kernel cost is a sum of several integrals and admits straightforward unbiased Monte-Carlo estimation from random batches. *Could you please further elaborate, why do you consider the proposed cost complex?*
>
> **(2) Comparison with other neural network-based algorithms for solving OT mappings is lacking. This paper does not explain the advantages of the proposed algorithm compared with the classical algorithm, and there is no experimental support. [...] However, the lack of experimental support, the key is the lack of experimental comparison of NOT, does not reflect the advantages of improving the cost function. Moreover, there is a lack of comparison of classical neural OT algorithms, such as ICNN.**
>
> To the best of our knowledge, NOT is the **only neural algorithm for weak OT**, so there is *nothing else to compare with*. Moreover, as we explicitly discuss in Section 4, the algorithm also **subsumes** various prior neural OT methods (Rout et al., 2022), (Fan et al., 2021a), (Korotin et al., 2021b) for **strong** OT. These methods are **state-of-the-art** methods in recovering strong OT maps according the Wasserstein-2 benchmark (Korotin et al., 2021b). Thus, formally, **we compare with state-of-the-art methods in OT** by comparing with base NOT for stong and weak quadratic costs.
>
> **ICNN-based methods**. We did not perform any comparison with ICNN-based algorithm because they are known to be not suitable for large-scale problems due to poor expressiveness of ICNN architectures (Korotin et al., 2021b). The two other recent papers (Fan et al., 2021b) and (Korotin 2022b) also show that replacing the ICNN-based architectures with general architectures significantly boosts the performance of OT methods in image-based applications. We emphasize that among the authors of these two recent papers there are authors of original ICNN-based methods (Makkuva et al., 2019), (Taghvaei et al., 2019), (Korotin et at., 2021a).
>
> Following your comment, we took ICNN-based method and applied it to the unpaired translation on *celeba*$\rightarrow$*anime*, *outdoor*$\rightarrow$*church* and *handbags*$\rightarrow$*shoes* dataset pairs. We used the publicly available ConvICNN64 architecture and code from the Wasserstein-2 benchmark repo (Korotin et al., 2021b) and considered $64\times 64$ RBG images for brevity. The results are given in the supplementary material (**answers/reviewer\_84ix** folder). In each image, the 1st line shows inputs $x\sim\mathbb{P}$, 2nd line -- learned translated images $T(x)$, 3rd line -- random images from $y\sim\mathbb{Q}$. One may clearly see that the performance (even on $64\times 64$ images) is very poor. The ICNN-based method tries to learn something meaningful (e.g., in *celeba*$\rightarrow$*anime* one may clearly see the appearance of anime eyes) but fails. This expectidely happens because of the poor expressiveness of ICNN architectures which are not suitable for large scale tasks.
>
> **(3)** ...and the KNOT algorithm is proposed, but the experimental comparison of the two algorithms is not given in the experiment, which lacks the support of the experiment.}
>
> We think there might be **several crucial misunderstandings**. First, we do not propose a new algorithm. We study a particular NOT algorithm  (Korotin et al., 2022b) for weak OT and propose the kernel transport costs for it. Second, the dominant majority of experiments in our paper are comparisons of the performance of NOT with the weak quadratic and our proposed kernel costs.
>
> **(a)** We do both qualitative and quantitative experimental comparisons in the main text (Section 5) and Appendix J.
>
> **(b)** We provide a **detailed quantitative comparison of the performance and stability** in Appendix H, where we report the results of multiple random restarts to justify our observations from the statistical point of view.
>
> **(c)** We provide **explanatory** toy 2D examples (Figures 5,6 in Section 3.1 and Figures 12, 13 in Appendix C) illustrating the issues of the quadratic cost and toy examples of performance of our proposed kernel cost (Appendices A, B).
>
> Additionally, our observations are supported by **rigorous theoretical analysis** of weak quadratic and proposed kernel costs. Thus, we think that our main claims are sufficiently well supported.
>
> *Could you please clarify which extra support of our main claims would you like to see?*

---

> > ### Author Response · Authors · 2022-11-16
> > **Response to Reviewer 84ix + ICNN method results (Part 2)**
> >
> > **(4) Only an improvement of kernel cost function is given, which lacks some novelty.**
> >
> > To the best of our knowledge, **weak** kernel costs have never been considered in the OT literature, especially in that related to machine learning. Thus, our presented results are entirely novel. According to our detailed theoretical analysis and practical evaluation, such costs provide a notable improvement (both theoretical and practical) w.r.t. the weak quadratic costs used in the prior works.
> >
> > **(5) Can not be reproduced.**
> >
> > In the initial submission, we provide the entire source code for all the experiments presented in the paper.
> >
> > *Could you please clarify, which part can not be reproduced?*
> >
> > **Concluding remarks**. Please respond to our post to let us know if the clarifications above suitably address your concerns about our work. We are happy to address any remaining points during the discussion phase; if the responses above are sufficient, we kindly ask that you consider raising your score.
> >
> > **References**
> > - Rout, L., Korotin, A., & Burnaev, E. (2021, September). Generative Modeling with Optimal Transport Maps. In International Conference on Learning Representations.
> > - Fan, J., Liu, S., Ma, S., Chen, Y., & Zhou, H. (2021a). Scalable computation of monge maps with general costs. arXiv preprint arXiv:2106.03812.
> > - Korotin, A., Li, L., Genevay, A., Solomon, J. M., Filippov, A., & Burnaev, E. (2021b). Do neural optimal transport solvers work? a continuous wasserstein-2 benchmark. Advances in Neural Information Processing Systems, 34, 14593-14605.
> > - Makkuva, A., Taghvaei, A., Oh, S., & Lee, J. (2020, November). Optimal transport mapping via input convex neural networks. In International Conference on Machine Learning (pp. 6672-6681). PMLR.
> > - Fan, J., Taghvaei, A., & Chen, Y. (2021b, July). Scalable Computations of Wasserstein Barycenter via Input Convex Neural Networks. In International Conference on Machine Learning (pp. 1571-1581). PMLR.
> > - Korotin, A., Egiazarian, V., Li, L., & Burnaev, E. (2022b). Wasserstein iterative networks for barycenter estimation. arXiv preprint arXiv:2201.12245. (NeurIPS 2022)
> > - Taghvaei, A., & Jalali, A. (2019). 2-wasserstein approximation via restricted convex potentials with application to improved training for gans. arXiv preprint arXiv:1902.07197.
> > - Korotin, A., Egiazarian, V., Asadulaev, A., Safin, A., & Burnaev, E. (2020a, September). Wasserstein-2 Generative Networks. In International Conference on Learning Representations.

---

### Official Review · Reviewer_9WD8 · 2022-10-25

**Confidence:** 3
**Correctness:** 3
**Technical Novelty And Significance:** 3
**Empirical Novelty And Significance:** 2
**Recommendation:** 6

**Clarity, Quality, Novelty And Reproducibility:**

I am not an expert in neural optimal transport.
My correctness checks for all equations and notations are limited, and I may have missed something that needs to be corrected.
Therefore, I do not discuss this point in my review.

The motivation and problem that this paper address is clear and clearly understandable.
Moreover, to my knowledge, the proposed method seems novel and provides new findings to the community.

This paper promises to release all the source code and checkpoints to reproduce the experiments in this paper.
Therefore, reproducibility should be no problem.

**Details Of Ethics Concerns:**

I found no ethical concerns.

**Strength And Weaknesses:**

Strength:
* The proposed method seems reasonable, and motivation is clear.
* This paper provides the theoretical justification of superiority of the proposed method.
* The figures help to intuitively understand the methods.


Weaknesses:
* The most of the experimental results shown in this paper are qualitative comparisons with small number of evaluation data. Therefore, it may exist the doubt of cherry picking.
* Similarly, it is hard to say that the proposed method really provides the better performance by solving the fake plan issue from the results shown in this paper. Therefore, I feel that the experimental results do not much support the main claim of this paper.


**Summary Of The Paper:**

This paper proposes an extended method for neural optimal transport.
The authors first point out that the theoretical flaws of neural optimal transport with using the weak quadratic cost in terms of potentially learning face plans.
Then, they proposed a method of using kernel costs of neural optimal transport to overcome learning fake plans.
This paper also show the theoretical justification that the proposed method can solve the lack of the previous method.
The experiments are conducted on the image-to-image translation task, and demonstrate that the proposed method produces better results.


**Summary Of The Review:**

I think this paper (potentially) has sufficient contributions to be accepted to the conference.
However, as I wrote above, I am not a specialist in neural optimal transport, so that it is hard for me to check the correctness of the theoretical justification of this paper.
My recommendation is based on the fact that this paper shows the correct justification.

---

> ### Author Response · Authors · 2022-11-16
> **Response to Reviewer 9WD8**
>
> Thank you for your valuable feedback and positive assessment of our work. Please consider our general answer to all the reviewers where we summarize the experiments and explain how they support the main claim of the paper. Below we answer your questions.
>
> **(1) The most of the experimental results shown in this paper are qualitative comparisons with small number of evaluation data. Therefore, it may exist the doubt of cherry picking.**
>
> In the experiments with the unpaired translation, the qualitative results are not very explanatory as both the results for weak quadratic and our proposed kernel results are perceptually good. Thus, we also do the **quantitative evaluation** as well. In addition to Section 5, please consider **Appendix H**, where we provided a detailed quantitative evaluation of the weak quadratic and kernel costs with multiple random restarts.
>
> **(2) Similarly, it is hard to say that the proposed method really provides the better performance by solving the fake plan issue from the results shown in this paper. Therefore, I feel that the experimental results do not much support the main claim of this paper.**
>
> We provide a detailed theoretical justification of all our claims (**Section 3**), explanatory toy examples (**Section 3** and **Appendix C**) and extensive evaluation on real data (**Section 5** and **Appendix H**). *Could you please suggest, what additional evidence may be useful to support the main claims?*
>
> **Concluding remarks**. Please respond to our post to let us know if the clarifications above suitably address your concerns about our work. We are happy to address any remaining points during the discussion phase; if the responses above are sufficient, we kindly ask that you consider raising your score.

---

### Official Review · Reviewer_goNj · 2022-10-25

**Confidence:** 4
**Clarity, Quality, Novelty And Reproducibility:** The paper is generally clear, of high…
**Correctness:** 3
**Technical Novelty And Significance:** 3
**Empirical Novelty And Significance:** 3
**Recommendation:** 6

**Strength And Weaknesses:**

Strengths:
- Neural OT and related methods are of increasing interest across the community.
- The paper is generally well written. The math is clear and appropriate. Proofs are thorough, complete, and in my checking correct.
- The theoretical results are accompanied by detailed simulations.

Weaknesses:
- The claimed limitations of weak costs were not detailed as comprehensively as I would have liked. Specifically, how far off are the "fake" plans? Also, how bad is the instability? These seem like important points, given that they motivate the paper in the first place.
- I found the exposition difficult. The introduction is quite short, has jargon that is not explained, and hence doesn't completely set up the problem. After reading the paper through, I appreciate the contribution but it think it would help to try to get more of the details earlier in the paper.
- There are a large number of figures with image to image translation. I don't see what the unique contribution is for each. It seems like one would suffice in the main text.

Other detailed comments
- I didn't really understand what figure 1 was supposed to show. How would I know if the result is "good"?
- Please define strong and weak transport costs in the introduction. What is the significance of the difference aside from the fact that people use them?
- Please define fake solutions. It would be helpful to be specific here so readers understand the gravity of the situation.
- Please define a weak kernel cost when one uses it the first time.
- It is not clear why imagine to image translation is a good way of testing performance.
- "may be not an optimal stochastic map. In this paper, we show that for the γ-weak quadratic cost (3) this may be problematic: the arg infT sets might contain fake
solutions T" I don't quite understand whether this is the same point, i.e. it is problematic because it is not an optimal stochastic map, or whether there are other aspects that are problematic. If it is problematic because it is not an optimal stochastic map, and the fake solutions are of this type, it would help to use simpler language (fewer different names) and adopt a label more descriptive than "fake solutions".
- I appreciated the inclusion of a derivation of (9) in the appendix.
- "Should we really care about solutions being fake?" It would be great for the answer to this question to appear early in the paper.
- It would be nice to have some sense of how bad fake solutions are. Are these truly pathological? (In the sense of being far from Q?) Also, can we demonstrate the stability issues more concretely?
- I don't see the added value of having the example figures in the paper (beyond the first one).

**Summary Of The Paper:**

The paper introduces kernel neural optimal transport. Prior work on neural optimal transport makes use of a weak cost. The paper proves that weak costs admit non-solutions in the sense that the resulting plans do not satisfy the marginal constraints. The authors also propose that the weak costs are related to optimization instability. As a solution, weak kernel costs are introduced. Extensive testing demonstrates the effectiveness of weak kernel costs including via unpaired image to image translation tasks.

**Summary Of The Review:**

The paper introduces kernel NOT, which uses weak kernel costs, to address issues in earlier neural OT work. There are both theoretical and extensive simulation results. The simulations do not document the limitations of and solution to the prior work as clearly as I would have liked. Overall, this is a nice contribution.

---

> ### Author Response · Authors · 2022-11-16
> **Response to Reviewer goNj**
>
> Thank you for your positive assessment of our paper, detailed analysis and valuable feedback. We took your feedback onboard and uploaded a revision of the paper. Please find our answers to your questions below.
>
> **(1) I didn't really understand what figure 1 was supposed to show. How would I know if the result is "good"? [...] I don't see the added value of having the example figures in the paper (beyond the first one).**
>
> Figure 1 plays the role of the teaser for the paper. The other qualitative experimental results (Figures 7, 8) are included to demonstrate that our proposed kernel costs are suitable for a wide range of pairs of datasets.
>
> **(2) It is not clear why image to image translation is a good way of testing performance.**
>
> Please see our general answer to all the reviewers.
>
> **(3) What is the significance of the difference aside from the fact that people use them? [weak vs. strong costs]**
>
> As the authors of NOT noted in their paper (Korotin et al., 2022b, Section 5.1), the usage of strong costs leads to the **conditional collapse**, i.e., the learned transport part $T_{x}(z)$ becomes independent of $z$, i.e., $T_{x}(z)=T(x,z)=T(x)$. From the theoretical side, this prevents from learning the actual OT plan when it is stochastic because $T_{x}(z)$ degenerates. In the practical tasks, this removes the capability to generate diverse output samples $T_{x}(z)$ for a given $x$. In turn, weak quadratic or our proposed weak kernel costs are helpful to solve this issue.
>
> **(4) Please define fake solutions. [...] Please define a weak kernel cost when one uses it the first time.**
>
> Following your suggestion, we defined fake solutions at the beginning of **Section 3.1** and weak kernel costs -- in **Section 3.2**. Defining them earlier seems not possible as they both require a large amount of mathematical preliminaries.
>
> **(5) [...] it is problematic because it is not an optimal stochastic map, or whether there are other aspects that are problematic [...] "Should we really care about solutions being fake?" It would be great for the answer to this question to appear early in the paper.**
>
> To address your comment, we provided the early discussion about why fake solutions are problematic to the beginning of **Section 3.1** in the reviser paper. Specifically, we added new motivating **Lemma 1** which demonstrates that all the fake solutions are simply not measure-preserving, i.e., they do not transform input distribution $\mathbb{P}$ to the target $\mathbb{Q}$. The proof is given in **Appendix G**.
>
> **(7)  would be nice to have some sense of how bad fake solutions are. Are these truly pathological? (In the sense of being far from Q?)**
>
> Our Theorem 1 provides an **explicit characterization** of the fake solutions via the sets $U_{\psi}$ of local linearity of $\psi=\overline{\phi^{\star}}(y)-\frac{\gamma}{2}\|y\|^{2}$, where $\phi^{\star}$ is an optimal restricted potential for the pair $\mathbb{P},\mathbb{Q}$. The result states that for each $x$ the fake solution $T$ may spread the mass of input $x$ nearly arbitrarily inside the set $U_{\psi}(\nabla\phi^{\star}(x))$. Thus, the badness of the fake solution depends on the **sizes** of the linearity sets of the convex $\psi$ which depends on the optimal potential $\phi^{*}$.
>
> Our toy 2D example in Section 3.1 shows that such sets might be extremely large (each set is the entire $\mathbb{R}^{D}$!) which might lead to **truly pathological** solutions. In the additional example in Appendix C, the sets $U_{\psi}(\cdot)$ seem to be not very large as the fluctuations are not as big as in Section 3.1, but still they lead to unstable behavior (Figures 12, 13).
>
> **(8) Also, can we demonstrate the stability issues more concretely?**
>
> Please consider **Appendix H**. There we provide a **concrete comparison of the stability** of NOT with the weak quadratic and our proposed kernel cost. On the real image data, we do multiple restarts and report the results with the mean and central tendency. Our empirical evaluation shows that NOT with our proposed kernel costs yields more stable behaviour than NOT with the weak quadratic cost. This agrees with our theory which suggests that one of the reasons for unstable behaviour and severe fluctuations might be the existence of the fake solutions.
>
> **(9) The simulations do not document the limitations of and solution to the prior work as clearly as I would have liked.**
>
> *Could you please suggest which additional limitations would you like to see the corresponding section?*
>
> **Concluding remarks**. Please respond to our post to let us know if the clarifications above suitably address your concerns about our work. We are happy to address any remaining points during the discussion phase; if the responses above are sufficient, we kindly ask that you consider raising your score.

---

### Author Response · Authors · 2022-11-16
**General Response**

Dear reviewers, thanks for your insightful comments! We highly appreciate that the reviewers find our theoretical part complete and accompanied by detailed simulations (**goNj**), our proposed method clearly motivated (**9WD8**) and the major points of the paper correct (**QYaw**). Reviewer **84ix** mentions ICNN-based OT methods  in context of comparisons -- we adressed this question in the corresponding reply to the reviewer.

As Reviewers **9WD8**, **84ix** and **QYaw** ask to clarify certain details of the experiments, in this general reply we decided to do a concise summary of the experimental evaluation presented in our paper.

**Summary of the experimental evaluation (9WD8, 84ix and QYaw).**

We emphasize that optimal transport is a tool which can be used in machine learning problems. In practice, solving the OT problem is rarely a self-goal. Usually, it is needed to compute the OT map (or cost) and use it in some downstream task. Therefore, experiments in our paper test the two following aspects of NOT with **our proposed kernel costs**:

**Aspect 1. Performance in the downstream task (real data).** Here we choose the unpaired image-to-image translation as the testbed for several reasons. First, NOT with the weak quadratic cost (our main competitor) considers the same application. Second, there are a lot of principal (non-OT) baseline models specifically for this task with reproducible code (CycleGAN, MUNIT, etc.). Third, there are well-known and commonly used metrics for this task, e.g., FID.

- We qualitatively and quantitatively compare with NOT with the weak quadratic cost (**Section 5, Appendix J**). Additionally, in **Appendix H**, we provide a detailed comparison of performance and stability of these costs. Our evaluation shows that NOT with our proposed kernel costs is more stable than NOT with the weak quadratic cost. In particular, the evaluation confirms that **empirical instabilities** of NOT with the weak quadratic cost which we reveal theoretically and demonstrate on toy examples (**Section 3**, **Appendix C**) *appear in real data* as well.

- We qualitatively and quantitatively compare with principal (non OT) unpaired translation methods (**Appendix K**).

**Aspect 2. Performance in optimal transport**. We show that NOT with the weak kernel costs actually recovers the weak kernel OT plan. Unfortunately, the evaluation here is limited by the fact that the ground truth kernel plan $\pi^{*}$ is *unknown*. Therefore,

- We provide **direct** evidence that our method recovers the OT plan in small-dimensional spaces. In **Appendix A**, we consider 1D distributions and use the solution of the discrete OT (with our weak kernel cost) as a fine *discrete* approximation of the actual OT plan. We compare our recovered plan with the discrete OT plan and see that they match.

- We provide **indirect** evidence that our method recovers the OT plan in high-dimensional spaces. In **Appendix D**, we study the behaviour of the **conditional variance** and **input-output distance** (which are parts of the cost) for varying parameter $\gamma$ in the cost. With the increase of $\gamma$, these statistics behave (Table 2) as suggested by our theory (Proposition 3), namely input-output distance and the conditional variance increase.

- All the qualitative results (e.g., Figures 1, 7, 8) serve as the evidence that our method recovers the OT plan. Indeed, by the formulation of OT problem with the kernel costs, one expects the output images $y=T_{x}(z)\sim \pi^{\star}(x)$ to be similar in $\ell^{2}$ norm to $x$, while having some diversity. This behavior is observed in the experiments with the unpaired translation.

*To conclude, we think that the provided evaluation is sufficient as it demonstrates the superioty of the weak kernel costs w.r.t. the weak quadratic cost in a series of toy and real data experiments (the key claim of our paper). Importantly, in the paper, we justify this superioty not only experimentally but also via rigorous mathematical analysis (Section 3).*

---

> ### Author Response · Authors · 2022-11-16
> **Paper revision**
>
> We have revised the paper according to the reviewers' suggestions. The changes are highlighted with the **blue** color. The changes are:
>
> - **[goNj]** In the beginning of **Section 3.1**, we provided the early discussion about why fake solutions are problematic (Lemma 1; proof is in **Appendix G**).
>
> - **[goNj]** We highlighted the formal introduction of *fake solutions* (Section 3.1) and *weak kernel costs* (Section 3.2).
>
> - **[QYaw]** We fixed minor typos mentioned by the reviewer.
>
> Please consider the updated paper. We are happy to address any remaining points during the discussion phase.

---

### Decision · Program_Chairs · 2023-01-20

**Decision:**

Accept: poster

**Justification For Why Not Higher Score:**

This paper is an interesting formulation of OT problems. However, the contribution of this paper is relatively focused to small number of researchers. Thus, presenting it as a poster is reasonable.

**Justification For Why Not Lower Score:**

This paper can be potentially rejected since many reviewers are not that excited about the result. However, the proposed method is solid and it is good to have the paper at the conference if there is enough space to present.

**Metareview: Summary, Strengths And Weaknesses:**

In this paper, the authors propose a kernel version of neural optimal transport method. The authors provide the theoretical and empirical analysis of the saddle point optimization problem of the neural OT algorithm for the weak quadratic cost. Overall the authors proposed an interesting OT method with theoretical guarantee, and many of reviewers puts positive reviews. Thus, I also vote for an acceptance.

**Note From Pc:**

if the above contains the word "oral" or "spotlight" please see: "oral" presentation means -> notable-top-5% and "spotlight" means -> notable-top-25%. As stated in our emails, we are disassociating presentation type from AC recommendations